# Transplantation of hPSC-derived pericyte-like cells promotes functional recovery in ischemic stroke mice

Jiaqi Sun [1,2,3,12], Yinong Huang [4,5,12], Jin Gong [1,12], Jiancheng Wang [2], Yubao Fan [2], Jianye Cai [6], Yi Wang [2], Yuan Qiu [2], Yili Wei [2], Chuanfeng Xiong [2], Jierui Chen[2], Bin Wang [2], Yuanchen Ma [2], Lihua Huang [7], Xiaoyong Chen [2], Shuwei Zheng [2], Weijun Huang [2], Qiong Ke [2,8], Tao Wang [2], Xiaoping Li[2], Wei Zhang [9], Andy Peng Xiang [2,10,11✉] & Weiqiang Li [2,10,11✉]

Pericytes play essential roles in blood–brain barrier (BBB) integrity and dysfunction or degeneration of pericytes is implicated in a set of neurological disorders although the underlying mechanism remains largely unknown. However, the scarcity of material sources hinders the application of BBB models in vitro for pathophysiological studies. Additionally, whether pericytes can be used to treat neurological disorders remains to be elucidated. Here, we generate pericyte-like cells (PCs) from human pluripotent stem cells (hPSCs) through the intermediate stage of the cranial neural crest (CNC) and reveal that the cranial neural crest-derived pericyte-like cells (hPSC-CNC PCs) express typical pericyte markers including PDGFRβ, CD146, NG2, CD13, Caldesmon, and Vimentin, and display distinct contractile properties, vasculogenic potential and endothelial barrier function. More importantly, when transplanted into a murine model of transient middle cerebral artery occlusion (tMCAO) with BBB disruption, hPSC-CNC PCs efficiently promote neurological functional recovery in tMCAO mice by reconstructing the BBB integrity and preventing of neuronal apoptosis. Our results indicate that hPSC-CNC PCs may represent an ideal cell source for the treatment of BBB dysfunction-related disorders and help to model the human BBB in vitro for the study of the pathogenesis of such neurological diseases.

[1] Department of Neurosurgery, The Third Affiliated Hospital, Sun Yat-Sen University, Guangzhou, China. [2] Center for Stem Cell Biology and Tissue Engineering, Key Laboratory for Stem Cells and Tissue Engineering, Ministry of Education, Sun Yat-Sen University, Guangzhou, Guangdong, China. [3] Bioland Laboratory (Guangzhou Regenerative Medicine and Health Guangdong Laboratory), Guangzhou, Guangdong, China. [4] Department of Neurology, The Third Affiliated Hospital, Sun Yat-Sen University, Guangzhou, China. [5] Department of Endocrinology, The First Affiliated Hospital of Sun Yat-Sen University, Guangzhou, Guangdong, China. [6] Department of Hepatic Surgery and Liver Transplantation Center, The Third Affiliated Hospital, Organ Transplantation Institute, Sun Yat-Sen University, Guangzhou, China. [7] Department of Pediatric Surgery, Guangzhou Women and Children's Medical Center, Guangzhou Medical University, Guangzhou, China. [8] Department of Cell Biology, Zhongshan Medical School, Sun Yat-Sen University, Guangzhou, Guangdong, China. [9] Guangdong Provincial Engineering and Technology Research Center of Stem Cell Therapy for Pituitary Disease, Department of Neurosurgery, The First Affiliated Hospital of Guangdong Pharmaceutics University, Guangzhou, Guangdong, China. [10] Department of Biochemistry, Zhongshan Medical School, Sun Yat-Sen University, Guangzhou, Guangdong, China. [11] Guangdong Key Laboratory of Reproductive Medicine, Guangzhou, Guangdong, China. [12] These authors contributed equally: Jiaqi Sun, Yinong Huang, Jin Gong. ✉email: xiangp@mail.sysu.edu.cn; liweiq6@mail.sysu.edu.cn

The blood–brain barrier (BBB) is a neurovascular unit (NVU) that serves as a physical and chemical barrier against plasma components, blood cells, and pathogens for protecting the central nervous system (CNS). The BBB also controls the exchange and movement of nutrients, hormones and other molecules into and out of the brain for proper functioning of the CNS[1]. Previous studies have demonstrated that the BBB consists of brain microvascular endothelial cells (BMECs), astrocytes, neurons, pericytes, and extracellular matrix around the vessels composed mainly of type IV collagen, fibronectin, laminin, heparan sulfate, and perlecan[2–4]. The dysfunction of the BBB is found to be related to numerous acute and chronic neurological deficits and/or cerebrovascular diseases, including Alzheimer's disease (AD), Parkinson's disease (PD), Huntington's disease, ischemic stroke, and so on[1].

It has long been recognized that BMECs play an essential role in the maintenance of the BBB[3] and the tight junctions (TJs), which are composed of claudin-5, Occludin, and zonula occludens-1 (ZO-1) formed by BMECs are critical for modulating the structural integrity of the BBB[5,6]. However, the major contribution of pericytes to the BBB was reported in 2010 by two studies using pericyte-deficient mouse models. They revealed that pericyte–endothelial cell interactions were critical in the establishment and maintenance of the BBB, and pericytes could influence BBB-specific gene expression patterns in BMECs and induce the polarization of astrocyte end-feet surrounding CNS blood vessels[7,8]. Pericytes are contractile cells[9] that express basement membrane proteins, including laminins[10] and vitronectin[11] involved in the construction of the basal lamina at the BBB and play important roles in vascular tube remodeling, maturation and stabilization[12]. Furthermore, recent transcriptomic studies suggest that pericytes express multiple transporters, receptors, and ion channels[1,11,12]. These results indicate that pericytes play a fundamental role in regulating the chemical composition of the extracellular fluid surrounding the brain cells and cerebral blood flow[1].

Given the critical roles of pericytes in BBB function, it is unsurprising that the dysfunction or degeneration of the pericytes contributes to the pathogenesis of diverse CNS disorders. Early death of capillary pericytes and the disruption of the BBB during ischemic stroke were detected in animal models and in human samples[13]. In AD, a loss of pericytes from the capillary wall and the derangement of the BBB also appeared and seemed to promote amyloid β accumulation, tau pathology and early neuronal death[12]. Most recently, Nortley et al.[14] found that amyloid β oligomers could lead to the dysfunction of contractile pericytes and constrict brain capillaries through the ROS-EDN1-EDNRA pathway. The above evidence suggests that pericyte replacement therapies may help to restore normal pericyte function and BBB integrity. However, the scarcity of material sources greatly limits the application of primary pericytes in disease modeling and cell transplantation studies.

Accordingly, human pluripotent stem cells with the properties of self-renewal and pluripotency present an ideal cell model for the isolation of pericytes to study their development or the therapeutic effect in BBB-related diseases. Previous studies demonstrated that pericytes and smooth muscle cells in all blood vessels of the face and forebrain are derived from the cephalic (cranial) neural crest[15,16]. Most recently, two studies also verified that forebrain pericyte-like cells (PCs) could be readily differentiated from human pluripotent stem cells (hPSCs) through the cranial neural crest (CNC) stage[17,18]. However, the in vivo functional characteristics and therapeutic potential of hPSC-derived pericyte-like cells still need to be clarified.

Here, we successfully derive pericyte-like cells with cranial neural crest origin from hPSCs (designated as hPSC-CNC PCs),

and show that hPSC-CNC PCs express typical pericyte markers and display similar contractile properties, vasculogenic ability and endothelial barrier function as primary human brain vascular pericytes (HBVPs). More importantly, we demonstrate that intravenous injection of hPSC-CNC PCs into a murine model of transient middle cerebral artery occlusion (tMCAO) with BBB disruption could efficiently promote neurological functional recovery by restoring of BBB properties and preventing neuron death.

## Results

**Generation of cranial neural crest from hPSCs.** To develop a stable, chemically defined and highly efficient protocol for cranial neural crest differentiation from hPSCs, we modified the protocols based on dual SMAD inhibition and WNT activation described previously[19–21] (Supplementary Fig. 1a). We found that differentiated cells proliferated continuously and became confluent at days 6–7 in our differentiation system (Supplementary Fig. 1b). Immunostaining assays showed that sustained culture in NCN2 for 6 days resulted in large proportion of cells expressing the neural crest surface markers p75 and HNK1 and the transcription factors SOX10 and AP2α, while only a small number of differentiated cells were PAX6 positive. Moreover, the intermediate filament Nestin was widely expressed by these cells (Supplementary Fig. 1c). The quantitative PCR (qPCR) analysis further showed that messenger RNAs (mRNAs) for the neural crest-specific markers SOX10, AP2α, and p75 were highly upregulated, whereas the expression of endogenous pluripotency markers OCT4 (POU5F1), NANOG and KLF4 decreased rapidly in a time-dependent manner during the neural crest differentiation of hiPSCs (Supplementary Fig. 1d).

HNK1 and p75 are widely used to enrich neural crest from human pluripotent stem cells by fluorescent-activated cell sorting (FACS)[20,22]. However, several recent studies demonstrated that p75bright cells that expressed high levels of the neural crest marker AP2α or SOX10 were bona fide neural crest cells[19,21]. Therefore, we intended to isolate the HNK1+p75bright cell population by FACS and further characterized these cells. FACS analysis showed that the neural crest differentiation efficiency (the percentage of HNK1+p75bright cells) of human-induced pluripotent stem cells (hiPSCs) was over 80% (Fig. 1a). Moreover, the neural crest cells could also be readily generated from human embryonic stem cell lines H1 and H9, indicating the universality and robustness of this protocol (Supplementary Fig. 2a–f). The freshly isolated neural crest cells were then cultured on poly-L-ornithine/fibronectin (PO/FN)-coated dishes in NCCM for adherent culture and attached cells maintained their typical cellular morphology and the expression of neural crest-specific markers during long-term in vitro culture, as illustrated by immunostaining and qPCR (Fig. 1b–d and Supplementary Fig. 3). We also found that HOXA1, which played important roles in the patterning of the cranial neural crest, was expressed in most HNK1+p75bright cells (Fig. 1c and Supplementary Fig. 3). We then examined the transcripts associated with cranial neural crest[23,24], and the results showed that mRNAs for HOXA1, HOXB1, LHX5 and OTX2 were highly upregulated in HNK1+p75bright cells (thus termed cranial neural crest cells, CNCs) (n = 3; Fig. 1d).

**Multilineage differentiation of hPSC-derived cranial neural crest cells.** The neural crest is a multipotent cell population arising from neural ectoderm in vertebrate embryos, capable of forming a wide array of derivatives, including neurons and glia of the peripheral nervous system, bone, cartilage and smooth muscle[22]. We thus explored the potential of hiPSC-derived CNCs (hiPSC-CNCs) to give rise to various cell types

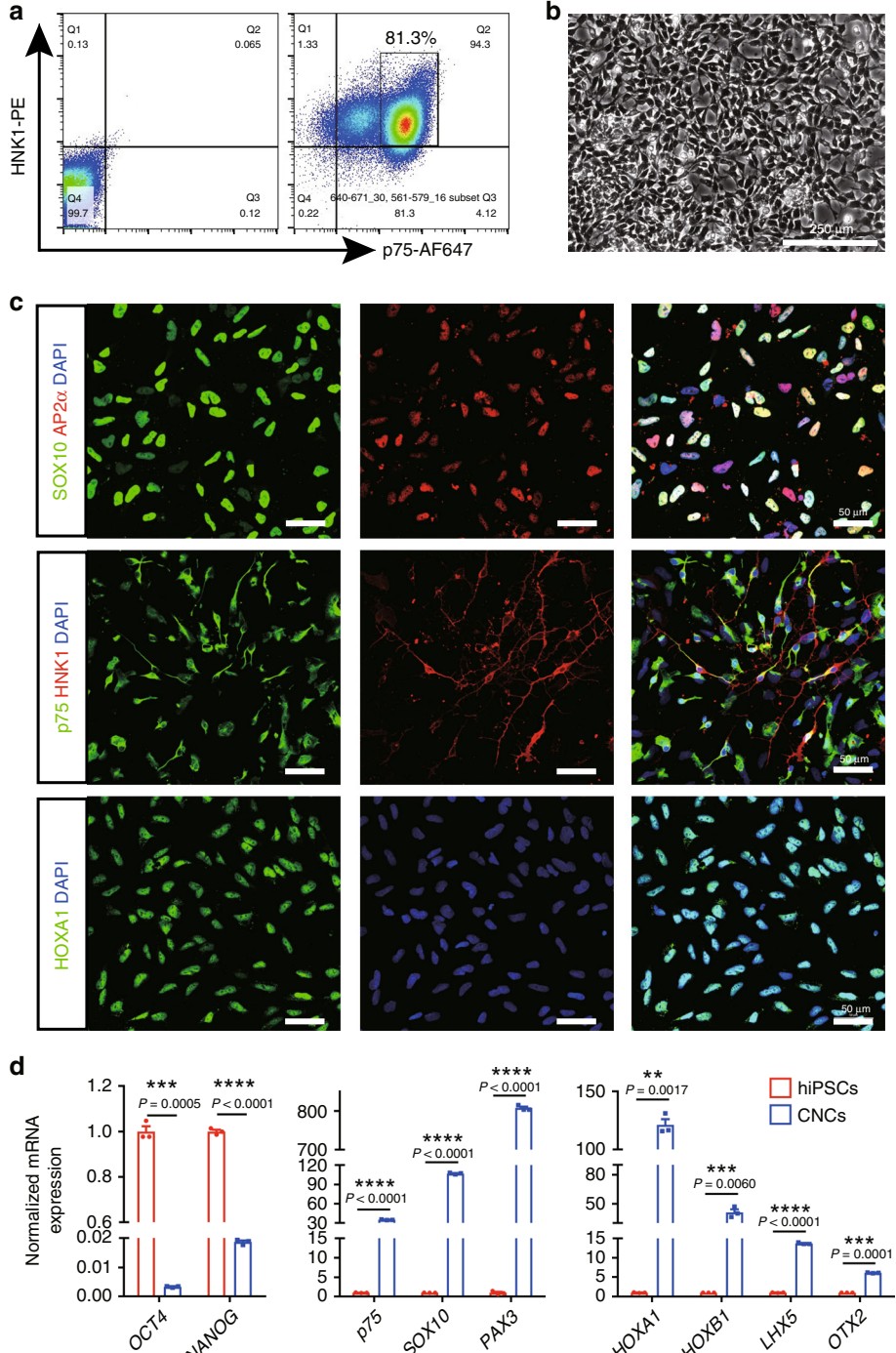

**Fig. 1 Enrichment and characterization of cranial neural crest cells (CNCs) derived from hiPSCs. a** CNCs (p75[bright]HNK1[+] cells) were isolated by FACS. **b** Enriched CNCs were cultivated in adherent monoculture and maintained typical neural crest morphology during in vitro culture. Scale bar: 250 μm. **c** The expression of cranial neural crest-specific markers (SOX10, AP2α, p75, HNK1, HOXA1) in isolated cell was detected by immunostaining. Scale bar: 50 μm. **d** The expression of pluripotency genes (*OCT4*, *NANOG*), CNC markers (*p75*, *SOX10*, *PAX3*), and cranial-specific genes (*HOXA1*, *HOXB1*, *LHX5*, *OTX2*) in undifferentiated hiPSCs and CNCs was detected by qPCR. Graphs represent the individual data points, the mean ± SEM of three independent experiments. FACS analysis, Confocal and bright field images are representative of $n = 3$ biological replicates. $P$-value (*$P < 0.05$, **$P < 0.01$, ***$P < 0.001$, ****$P < 0.0001$) was calculated by two-tailed unpaired Student's $t$-test. Source data are provided as a Source Data file.

(Supplementary Fig. 4a). We confirmed that when cultured in neural induction medium for 3–4 weeks, most of the cells coexpressed peripherin and β3-tubulin (TUBB3), indicative of peripheral neuron identity (Supplementary Fig. 4b). Furthermore, glial derivatives, as defined by the coexpression of the Schwann cell markers S100b and GFAP or myelin basic protein (MBP), were observed when CNCs were differentiated in Schwann cell medium for 2–3 weeks (Supplementary Fig. 4c). To examine the differentiation potential toward mesenchymal lineages, CNCs were cultured in fetal bovine serum (FBS)-containing medium for ~2–3 weeks. Under these conditions, cells with mesenchymal morphologies emerged and

proliferated actively. In addition, mesenchymal cells generated from CNCs were capable of smooth muscle cell, osteogenic, adipogenic, and chondrogenic differentiation as illustrated by anti-αSMA immunostaining, Alizarin Red S staining, Oil Red O staining, and Toluidine blue staining, respectively (Supplementary Fig. 4d, e).

**Differentiation and characterization of hPSC-CNC-derived pericyte-like cells**. Quail-chick chimerization experiments demonstrate that pericytes found in the forebrain are derived from the neural crest[16]. Thus, we then verified whether hiPSC-CNCs have the potential to generate forebrain pericyte-like cells. CNCs were differentiated in pericyte medium supplemented with platelet-derived growth factor (PDGF)-BB for ~2 weeks (Fig. 2a), since PDGF-B/PDGFRβ signaling has been shown to play a pivotal role in pericyte development in vivo[25,26]. The CNCs adopted elongated mesenchymal morphology gradually and exhibited a parallel or spiral arrangement 14 days after induction (Fig. 2b). To analyze the characteristics of differentiated CNCs, a qPCR assay was performed and the results indicated that the mRNA levels of pericyte-specific markers such as *Caldesmon*, *CSPG4* (*NG2*) and *PDGFRβ* (*CD140b*) were highly upregulated, whereas the expression of neural crest-specific markers *p75*, *SOX10* and *PAX3* decreased rapidly in a time-dependent manner during the pericyte differentiation of CNCs (Fig. 2c). Flow cytometry analysis showed that most CNC-derived pericyte-like cells (hPSC-CNC PCs) expressed markers of pericytes or mesenchymal stromal cells, including PDGFRβ (CD140b), CD146, CD13, CD248, NG2, and PDGFRα (CD140a), at day 14 but were nearly negative for p75, HNK1, or CD45 (Fig. 2d and Supplementary Fig. 5a), similar to human brain vascular pericytes (HBVPs; isolated from human embryonic brain tissue). Notch3, an important regulator of brain vascular integrity and pericyte expansion[27], was detected in at least 50% of hPSC-CNC PCs and HBVPs (Supplementary Fig. 5a). The above results demonstrate that pericyte-like cells could be readily induced from hPSC-CNCs in PDGF-BB-containing medium.

Multiple assays were carried out to analyze the similarities in phenotype between hPSC-CNC PCs and HBVPs in detail. hPSC-CNC PCs exhibited similar spindle-shaped morphology as HBVPs and homogeneously expressed pericyte markers including NG2 and PDGFRβ, while Calponin was present in only a small population of hPSC-CNC PCs and HBVPs, as assessed by immunofluorescence staining. Significantly, most of the hPSC-CNC PCs and HBVPs lacked the expression of the smooth muscle cell marker αSMA (Fig. 3a), which is consistent with a previous study showing that brain capillary pericytes generally did not express αSMA in vivo[28]. The qPCR results further revealed that markers of neural crest stem cells (*p75*, *PAX3* and *SOX10*) were absent in hPSC-CNC PCs and HBVPs (Fig. 3b), while hPSC-CNC PCs and HBVPs expressed the mRNA transcripts of *Vimentin*, *PDGFRβ*, *NG2*, *Caldesmon*, *ABCC9*, *ANPEP* (*CD13*), *DLK1*, *IFITM1*[11], and others at similar or different levels (Fig. 3c and Supplementary Fig. 5b). In addition, western blotting assays identified that different protein levels of PDGFRβ, COL1A1, ANPEP, and IFITM1 were expressed in hPSC-CNC PCs and HBVPs. Interestingly, Vimentin protein expression in HBVPs was weaker than that in hPSC-CNC PCs, although both cell types had similar mRNA expression levels of this gene (Supplementary Fig. 5c). We also found that hPSC-CNC PCs proliferated faster than HBVPs as detected by CCK8 assay (*n* = 3; Fig. 3d). These results indicate that hPSC-CNC PCs share similar marker expression patterns with HBVPs. Moreover, pericyte-like cells expressing PDGFRβ, CD146, and CD13 could also be obtained from CNCs generated from H1 and H9 cell

lines following induction in PDGF-BB-containing conditions (Supplementary Fig. 6a, b).

It has been reported that marker expression is not significantly different between mesodermal- and neural crest-derived pericyte-like cells[18]. We then generated mesoderm progenitor-derived pericyte-like cells from hPSCs (hPSC-MP PCs) as described[18]. Indeed, we showed that pericyte-like cells of both origins highly expressed pericyte-related surface markers, including PDGFRβ, CD13, and NG2 (Supplementary Fig. 7a). Nonetheless, qPCR analysis revealed that the posterior HOX genes[29] were differentially expressed in these two types of pericytes and significantly higher mRNA levels of *HOX10–12* were detected in hPSC-MP PCs than in hPSC-CNC PCs (Supplementary Fig. 7b). The above evidence suggests that hPSC-MP PCs and hPSC-CNC PCs might represent distinct developmental origins despite similar phenotype marker expression patterns.

**Global gene expression profiling of hPSC-CNC PCs**. To further characterize the hPSC-CNC PCs, we performed RNA-seq to examine changes in the global expression profile during the pericyte differentiation of hPSCs (GSE132857). We established global lineage relationships by identifying genes that were differentially expressed among all populations and compared their transcripts per million (TPM) values, and we then used them to generate a two-way clustering heat map of differential gene expression profiles (451 genes; Supplementary Fig. 8a). Principal component analysis (PCA) of all mapped genes showed that undifferentiated pluripotent cells and CNCs in the intermediate differentiation stage each grouped together, while hPSC-CNC PCs from different cell lines together with control HBVPs exhibited distinct profiles (Supplementary Fig. 8b). Notably, we calculated coefficients of determination ($R^2$) for all expressed genes of hPSC-CNC PCs from hiPSCs or H9 and HBVPs, and the results revealed an extremely high level of similarity in gene expression profile between them ($R^2 > 0.90$) (Supplementary Fig. 8c). The results also demonstrated that the expression profile of undifferentiated hPSCs was enriched for genes related to pluripotency (*NANOG*, *SOX2*, *POU5F1*, *TDGF1*, *ZFP42*, and others). However, the freshly isolated hPSC-CNCs strongly expressed neural crest-specific transcription factors and cranial neural crest-related genes (for example, *SOX10*, *PAX3*, *PAX7*, *LHX5*, *TFAP2A*, and others). Significantly, the hPSC-CNC PCs and HBVPs were highly enriched in transcripts associated with pericytes or fibroblasts, including *PDGFRβ*, *CD248*, *ANGPT1*, *MMP1*, *Vitronectin* (*ITGAV*), *Laminin* (*LAMB1*), lipoprotein receptor (*LRP1*), *PDGFRA*, *COL1A1*, *COL1A2*, and *LUM*, which was in accordance with the results of qPCR or western blotting as described above (Supplementary Fig. 8d). Moreover, contractile proteins, including Vimentin, Caldesmon, tropomyosin (*TPM1*, *TPM2*, *TPM3*), and myosin (*MYH9*, *MYH10*) were also enriched in CNC PCs and HBVPs (Supplementary Fig. 8d).

**Functional characterization of CNC-derived pericyte-like cells**. To study the pericyte-specific functional properties of hPSC-CNC PCs, we evaluated their potential to stabilize vascular networks in vitro. We replated the cells on growth factor-reduced Matrigel and cocultured them with human brain microvascular endothelial cells (HBMECs) in vitro. Lavish cord networks were observed in both groups at day 1 (Fig. 3e). The resulting cord networks showed that cocultures containing different test cells have variable cord thickness, branching and cell clustering at branch points (Fig. 3e). The HBVPs could support the cord networks for ~72 h when cocultured with HBMECs (Fig. 3e and Supplementary Movie 1). Interestingly, the CNC-derived pericyte-like cells demonstrated a more profound effect on cord stability than

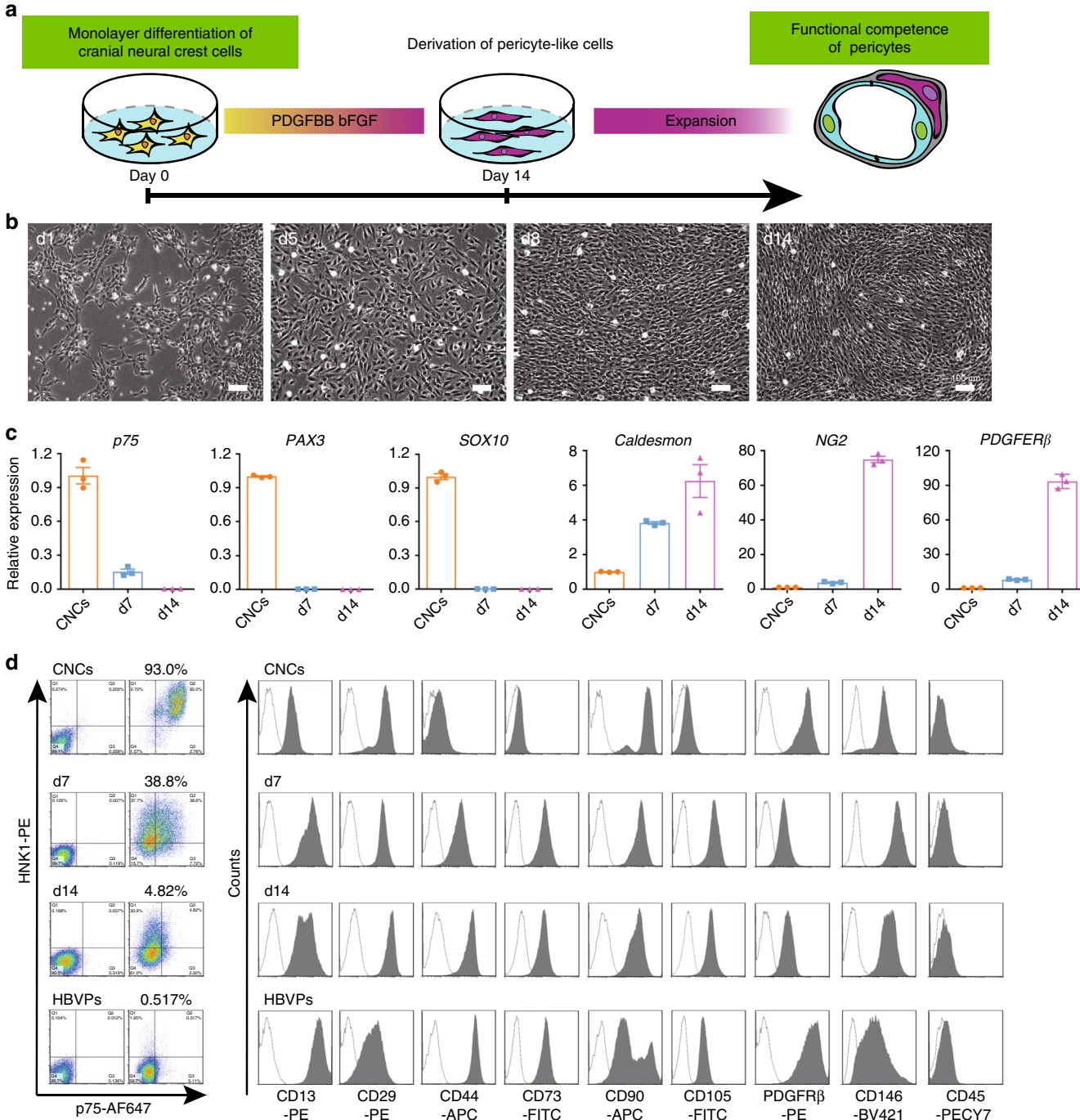

**Fig. 2 Differentiation of hiPSC-derived CNCs to pericyte-like cells. a** Strategy for deriving pericyte-like cells from hiPSC-derived CNCs. **b** The morphology change during pericyte differentiation from CNCs was detected under phase-contrast microscopy. Scale bar: 100 μm. **c** qPCR were used for analyze the expression of CNC-specific genes (*p75*, *SOX10*, *PAX3*) and pericyte markers (*Caldesmon*, *NG2*, *PDGFR*β) during differentiation of pericyte-like cells from CNCs. **d**. FACS analysis for the surface marker expression of hPSC-CNCs, hPSC-CNC PCs (day 7 and day 14), and HBVPs. Graphs represent the individual data points, the mean ± SEM of three independent experiments. FACS analysis and Bright field images are representative of n = 3 biological replicates. Source data are provided as a Source Data file.

HBVPs and supported cord networks for up to 7 days (Fig. 3e and Supplementary Movie 2). These observations were quantified by the evaluation of the total cord length and its retention capacity (n = 3; Fig. 3f, g).

To further evaluate the vasculogenic ability of hPSC-CNC PCs in vivo, we generated DsRedE2-expressing pericyte-like cells from DsRedE2-hiPSCs and DsRedE2-labeled HBVPs (Fig. 4a), embedded them in a Matrigel matrix with hrGFP-HBMECs,

respectively, and implanted them into subcutaneous tissues of NOD-SCID mice. When transplanted alone, HBMECs formed very few small vessels. In contrast, hPSC-CNC PCs and HBVPs strongly supported the formation of neovessels (Fig. 4b). At high magnification, we found that hPSC-CNC PCs and HBVPs aligned along and wrapped around the lumenized vascular tube structures (Fig. 4b) and maintained the expression of PDGFRβ (Fig. 4c). There was no statistically significant difference in the

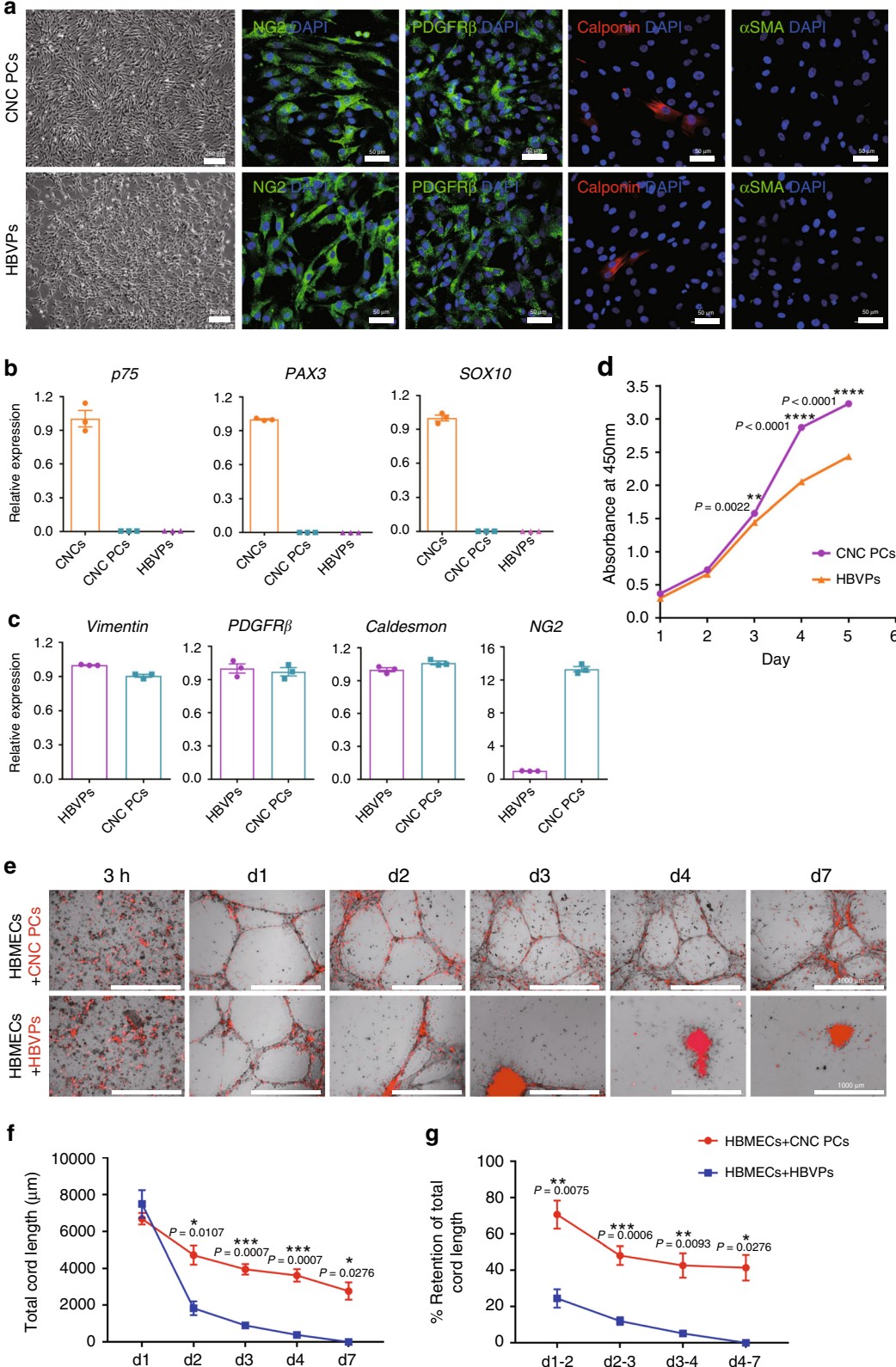

numbers of blood vessels between the hPSC-CNC PC group and HBVP group, but both helped to generate markedly higher numbers of vessels than the HBMEC group ($n = 3$; Fig. 4d). Significantly, the quantification of vessel diameter further demonstrated that HBVPs supported the formation of larger vessels, whereas smaller vessels were formed in the presence of hPSC-CNC PCs ($n = 3$; Fig. 4e). Thus, these in vivo studies

confirmed the strong capability of hPSC-CNC PCs to support the growth of the vasculature.

Pericytes are contractile cells surrounding the endothelial cells of small blood vessels and pericyte contraction may help to regulate blood flow in physiological and pathophysiological situations[30]. To determine the contractile properties of hPSC-CNC PCs, we performed a gel lattice contraction assay and

**Fig. 3 Characterization and cord formation potential of hPSC-CNC PCs. a** The cell morphology and expression of pericyte markers (Calponin, αSMA, NG2, and PDGFRβ) of hPSC-CNC PCs and HBVPs were revealed by phase-contrast microscopy and immunostaining, respectively. Scale bar: 250 μm (phase-contrast image) and 50 μm (immunostaining image). **b** qPCR assay for the expression of CNC-specific genes (*p75, SOX10, PAX3*) in hPSC-CNC PCs and HBVPs. **c** qPCR assay for the expression of pericyte markers (*Vimentin, NG2, Caldesmon*, and *PDGFRβ*) in hPSC-CNC PCs and HBVPs. **d** The proliferation activity of hPSC-CNC PCs and HBVPs was assessed by CCK8 assay. **e** A cord formation assay was used to monitor the vascular stabilization potential of hPSC-CNC PCs and HBVPs. Scale bar: 1000 μm. **f** Total cord length was measured and compared between the hPSC-CNC PC group and the HBVP group. **g** The retention of total cord length in the hPSC-CNC PC group and the HBVP group was calculated and compared. Graphs represent the individual data points, the mean ± SEM of three independent experiments. Confocal and cord images are representative of $n = 3$ biological replicates. *P*-value (*$P < 0.05$, **$P < 0.01$, ***$P < 0.001$, ****$P < 0.0001$) was calculated by two-tailed unpaired Student's *t*-test. Source data are provided as a Source Data file.

used human bone marrow-derived mesenchymal stem cells (HMSCs) as the negative control and human aortic smooth muscle cells (HASMCs) as the positive control[31], respectively (Fig. 5a). The results indicated that HBVPs and hPSC-CNC PCs exhibited similar basal contractile tone as HASMCs in this assay, with up to 60–80% reduction of the initial gel size, while HMSCs displayed minimal reduction in the initial gel size ($P < 0.01$; $n = 3$; Fig. 5b). A carbachol treatment assay was also performed to evaluate the contractile properties of hPSC-CNC PCs at the single-cell level. Time-lapse microscopy showed that hPSC-CNC PCs, HBVPs, and HASMCs substantially contracted in a tonic fashion and an ~20% change in surface area, whereas HMSCs presented very limited changes in surface area following carbachol treatment (Fig. 5c, d). These results indicate that hPSC-CNC PCs have contractile properties that are similar to those of HBVPs.

As pericytes are an important component of blood–brain barrier (BBB), we set up an in vitro model of the BBB that includes pericytes in coculture with endothelial cells to investigate whether hPSC-CNC PCs or HBVPs could improve the barrier function of HBMECs (Fig. 5e). We measured transendothelial electrical resistance (TEER), a hallmark of the BBB and a widely accepted quantitative technique to measure the integrity of TJ dynamics[30], for either HBMECs monolayers or cocultures with hPSC-CNC PCs/HBVPs. The results indicated that HBMECs cultured alone displayed lower TEER values than hPSC-CNC PCs/HBMECs or HBVPs/HBMECs group at different time points during the coculture process. A significantly increased TEER value of HBMECs was observed in both coculture groups from day 2 to day 4, and the maximal TEER value was detected 2 days following the initiation of coculture. TEER in the hPSC-CNC PC/HBMEC cocultures ($63.07 ± 2.77 \ \Omega \times cm^2$) was nearly one and a half times that observed in HBMEC monocultures ($45.83 ± 2.91 \ \Omega \times cm^2$), whereas TEER in the HBVP/HBMEC cocultures ($56.47 ± 0.97 \ \Omega \times cm^2$) was comparable to that in hPSC-CNC PC/HBMEC group 48 h after coculture ($n = 3$; Fig. 5f).

To further validate the formation of tight junctions formed by HBMECs after coculture with hPSC-CNC PCs or HBVPs, we analyzed the expression level of TJ proteins, including Occludin and ZO-1 by immunofluorescence staining. The data demonstrated that when HBMECs were grown in the absence of any pericytes, an irregular, "zipper-like" localization of the TJ proteins could be observed. In contrast, the localization of TJ proteins was more restricted to intercellular junctions delineating clear cell borders when HBMECs were cocultured with either hPSC-CNC PCs or HBVPs (Fig. 5g). Moreover, quantitative immunocytochemical evaluation showed that the number of frayed ZO-1 tight junctions was greatly reduced after coculture with hPSC-CNC PCs when compared to HBMEC monoculture (Fig. 5h, i). As one of the major functions of pericytes in the BBB is to regulate transcytosis within the brain endothelium[17], we then tried to determine the suppression ratio of hPSC-CNC PCs for transcytosis in HBMECs using 10-kDa dextran labeled with

Alexa 488. Compared to HBMEC monolayer group, coculture with hPSC-CNC PCs significantly decreased the Alexa 488-dextran fluorescence intensity in the medium (~50–60% decrease; $P < 0.0001$), which was similar to the effects of HBVPs (Fig. 5j). Altogether, these results revealed that hPSC-CNC PCs could strengthen the barrier integrity of cerebral endothelial monolayers as HBVPs.

**hPSC-CNC PC treatment promotes neurological functional recovery and reduces neuronal apoptosis.** To determine the potential therapeutic benefit for promoting the blood–brain barrier recovery, the tMCAO model of ischemic stroke was utilized and the changes in blood flow in the ischemic stage and reperfusion stage in tMCAO mice were evaluated by laser Doppler flowmetry (Supplementary Fig. 9a). To determine the potential therapeutic benefit for promoting the blood–brain barrier recovery of tMCAO mice, hPSC-CNC PCs, HBVPs, or human dermal fibroblasts were intravenously administered 6 h after tMCAO (Fig. 6a). Functional recovery was first assessed by the Menzies score as previously described[32,33], and greater improvement of the neurological deficits was detected in the hPSC-CNC PC and HBVP groups than in the human dermal fibroblast (HDF) and phosphate-buffered saline (PBS) groups at day 3 post stroke, and no significant difference was found between hPSC-CNC PC group and HBVP group (Fig. 6b). We then performed the rotarod test to evaluate locomotor coordination and motor balance after tMCAO. The results revealed that the latency time to fall from the rod was markedly prolonged on day 3 and day 7 after cell injection in the hPSC-CNC PC and HBVP groups compared to the PBS and HDF groups (~50 s) (Fig. 6c). The corner test was also used to measure the asymmetry in forelimb use in mice after tMCAO. The results indicated that mice treated with hPSC-CNC PCs and HBVPs exhibited similar enhancement but greater enhancement than HDF and PBS groups at days 3 and 7 post stroke (Fig. 6d). The adhesive removal test, which represents a sensitive approach to evaluate sensorimotor deficits in mice, revealed that mice in the hPSC-CNC PC and HBVP groups at days 3 and 7 after stroke took less time to remove the adhesives than mice in the PBS and HDF groups (Fig. 6e). We then evaluated the infarct size of the coronal sections of brains following tMCAO with and without cell infusion by 2,3,5-triphenyltetrazolium chloride (TTC) staining (Fig. 6f). Compared with the PBS group, the transplantation of HBVPs or hPSC-CNC PCs significantly reduced the infarct volumes, while only a mild reduction in infarct size in the HDF transplantation group (Fig. 6g). Moreover, much lower numbers of TUNEL$^+$/NeuN$^+$ cells in the cortex (CTX) and striatum (STR) of the hPSC-CNC PC (CTX: $276.5 ± 74.7$ cells/mm$^2$; STR: $214.5 ± 69.0$ cells/mm$^2$) and HBVP (CTX: $339.2 ± 87.9$ cells/mm$^2$; STR: $219.8 ± 71.6$ cells/mm$^2$) groups were detected when compared to that of the HDF group (CTX: $517.3 ± 42.6$ cells/mm$^2$; STR: $348.0 ± 94.4$ cells/mm$^2$) and the PBS group (CTX: $498.8 ± 81.6$ cells/mm$^2$; STR: $372.5 ± 84.9$ cells/mm$^2$) (Fig. 6h, i). These findings suggest that the infusion

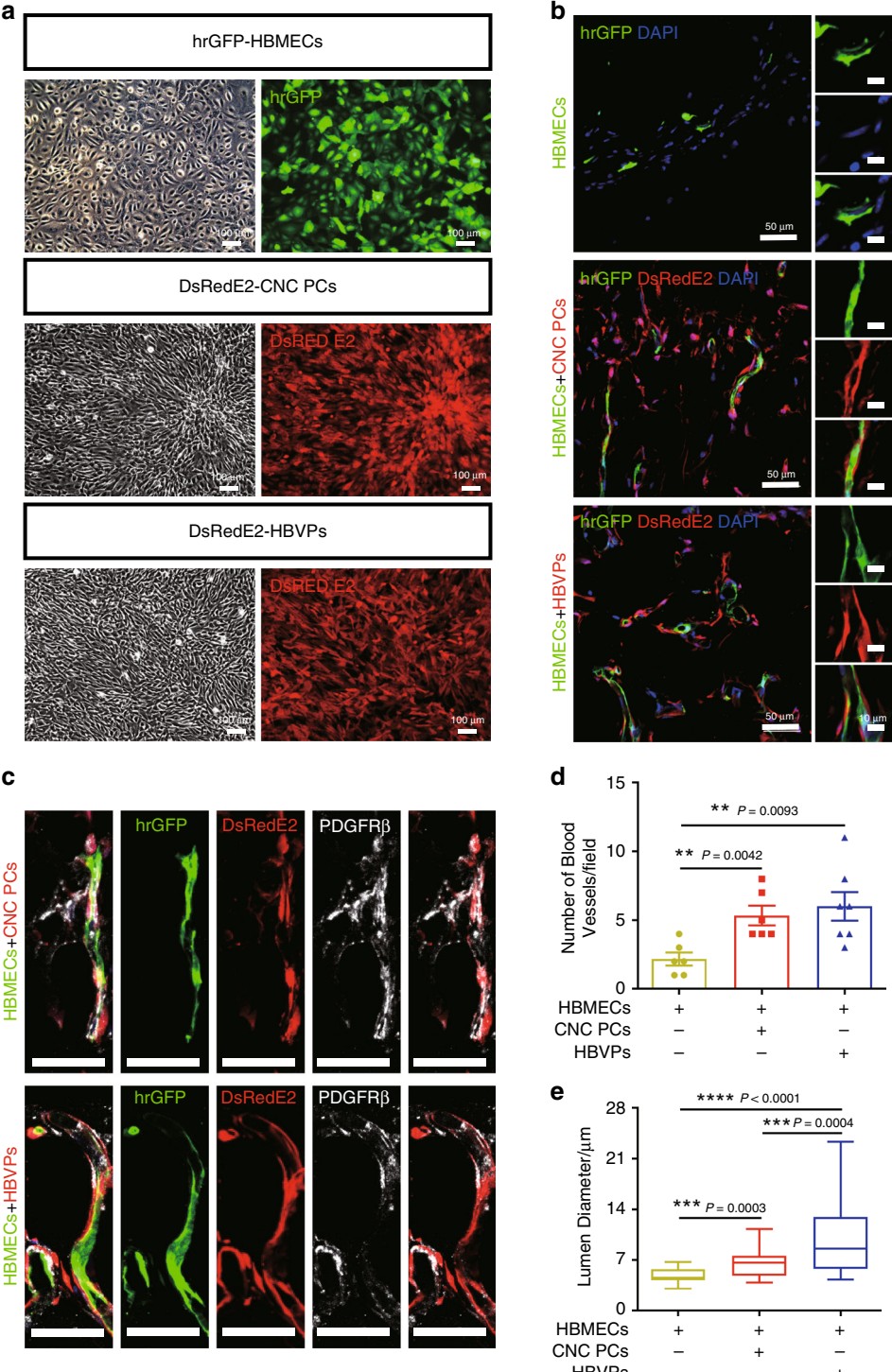

**Fig. 4 Matrigel plug assay shows the capacity of hPSC-CNC PCs to support the vasculature formation in vivo. a** DsRedE2-expressing hPSC-CNC PCs/ HBVPs and hrGFP-HBMECs were generated and used for the in vivo Matrigel plug assay. Scale bar: 100 μm. **b** The localization of DsRedE2-hPSC-CNC PCs and DsRedE2-HBVPs was observed under a fluorescence microscope. Scale bar: 50 μm (left panel) and 10 μm (right panel). **c** The expression of PDGFRβ in transplanted hPSC-CNC PCs and HBVPs was detected by immunofluorescence assay. Scale bar: 50 μm. **d** The numbers of blood vessels formed in different groups were counted. **e** The vessel diameter of different groups was quantified. Graphs represent the individual data points, the mean ± SEM of three independent experiments. Micrographs are representative of $n = 3$ biological replicates. $P$-value (*$P < 0.05$, **$P < 0.01$, ***$P < 0.001$, ****$P < 0.0001$) was calculated by two-tailed unpaired Student's $t$-test. Source data are provided as a Source Data file.

of hPSC-CNC PCs or HBVPs could rapidly improve neurological function at the early stage and protect neurons from death, while the infusion of HDFs displayed minimal therapeutic potential in the tMCAO model.

**hPSC-CNC PCs preserve BBB integrity in mice after ischemic stroke.** Previous studies have confirmed that damage to BBB integrity is closely related to cerebral edema after ischemia, and the repair of the BBB is considered an ideal therapeutic target to

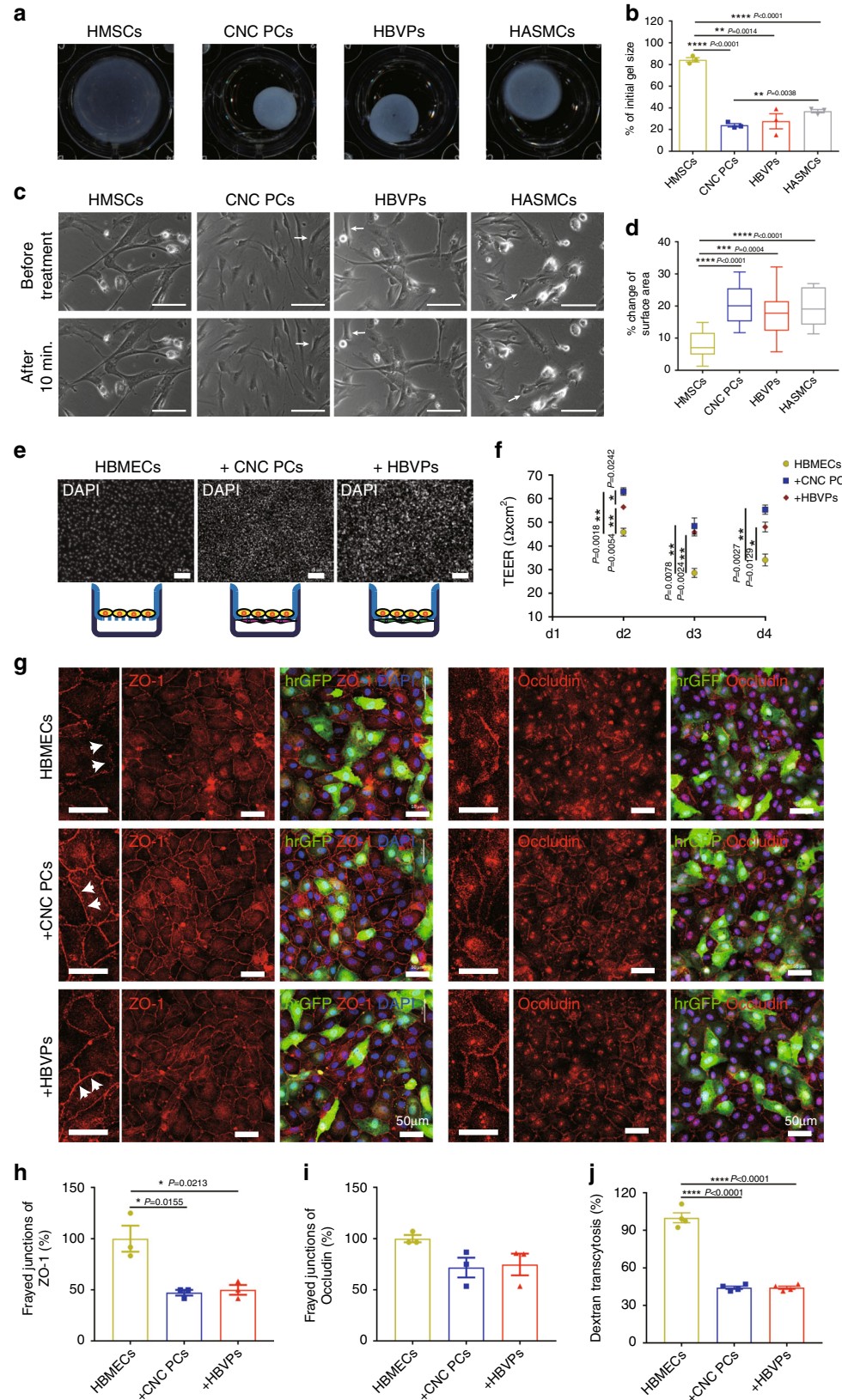

improve neurological function[2]. We sought to determine whether hPSC-CNC PCs intravenously administered could participate in BBB preservation. We observed apparent cerebral edema in brain samples from the PBS and HDF groups, whereas the transplantation of hPSC-CNC PCs and HBVPs noticeably reduced brain swelling (Fig. 7a), which was in accordance with the quantification analysis of brain water content in different groups (Fig. 7b). In addition, increased leakage of Evans blue dye was detected in the PBS and HDF groups, while treatment with HBVPs and hPSC-CNC PCs remarkably decreased Evans blue extravasation in the

**Fig. 5 Contractile and BBB properties of hPSC-CNC PCs in vitro. a** A gel lattice contraction assay was applied to test the contractile properties of hPSC-CNC PCs and HBVPs. **b** The gel size of different groups was calculated and compared. **c** Representative images of CNC PCs and HBVPs before and 15 min after treatment with carbachol (1 mM). Scale bar: 100 μm. **d** Changes in individual cell area following treatment with carbachol were calculated and compared among different groups. **e** An In vitro BBB model was established to investigate whether hPSC-CNC PCs or HBVPs could improve the barrier function of HBMECs. Scale bar: 75 μm. **f** The transendothelial electrical resistance (TEER) in different groups was measured and compared. **g** The expression of tight junction proteins, including Occludin and ZO-1 in HBMECs that cocultured with either hPSC-CNC PCs or HBVPs was revealed by immunofluorescence staining. Scale bar: 50 μm. **h** Quantification of frayed ZO-1 tight junctions. **i** Quantification of frayed Occludin tight junctions. **j** Transcytosis of Alexa 488–tagged 10-kDa dextran across HBMECs following 48 h of coculture with hPSC-CNC PCs or HBVPs. Graphs represent the individual data points, the mean ± SEM of three independent experiments. Micrographs are representative of $n = 3$ biological replicates. $P$-value (*$P < 0.05$, **$P < 0.01$, ***$P < 0.001$, ****$P < 0.0001$) was calculated by two-tailed unpaired Student's $t$-test. Source data are provided as a Source Data file.

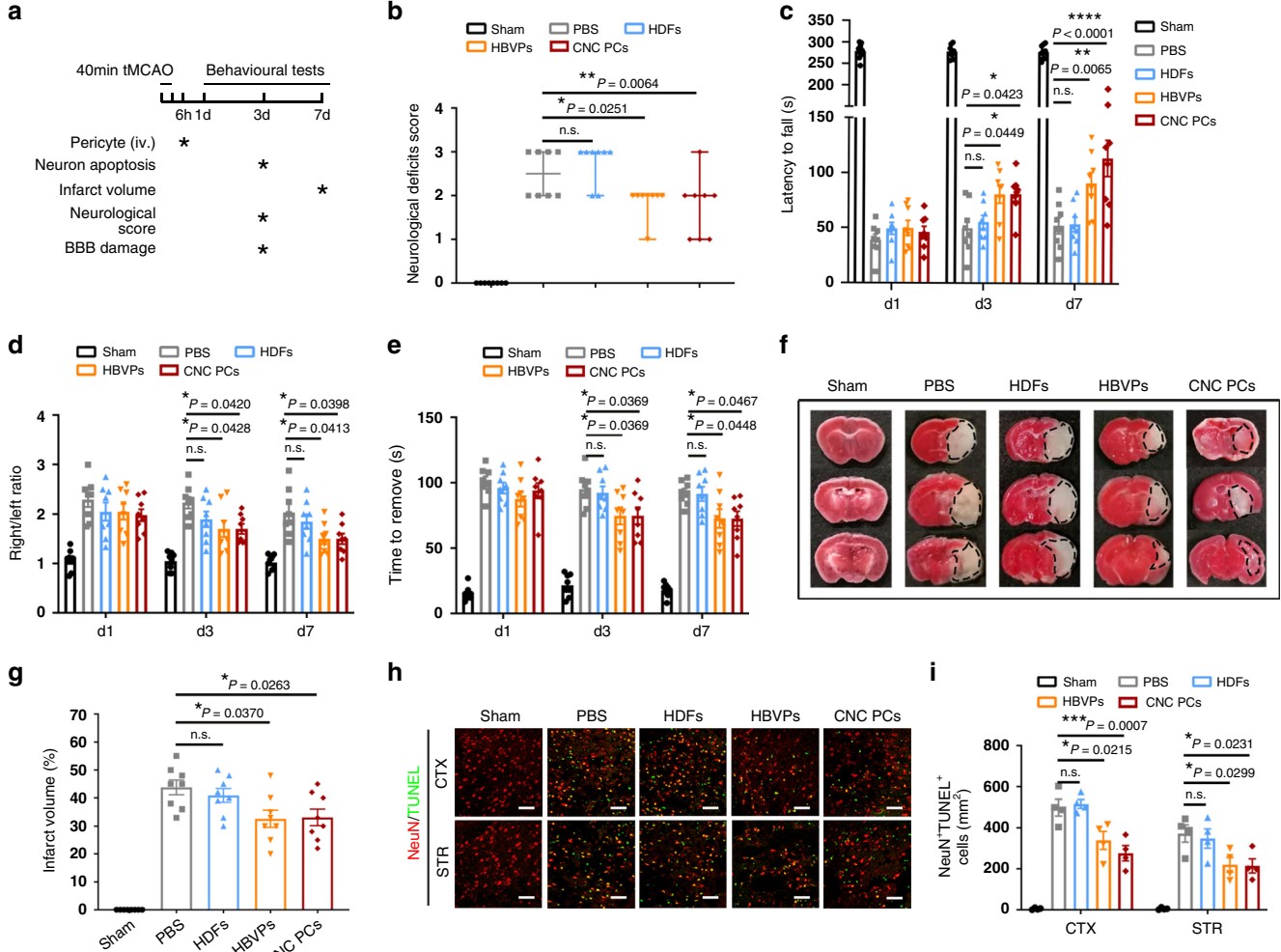

**Fig. 6 Transplantation of hPSC-CNC PCs ameliorates brain injury and improves neurological functional recovery after ischemic stroke. a** Scheme depicting the procedure of pericyte injection after stroke and the detection of neuronal apoptosis, neurological function and BBB damage. **b** Neurological deficit scores in different groups on day 3 were evaluated. **c** Motor coordination measurement from the rotarod test was rescued in tMCAO mice treated with HBVPs or CNC PCs. **d** The right/left ratio in the corner test at different time points was calculated. **e** The time to remove the sticker in the adhesive removal test at different time points was analyzed. **f** Representative TTC-stained coronal sections in different groups. **g** Infarct volumes of different groups were determined at 7 days post stroke and quantified on TTC (red)-stained coronal cerebral sections. **h** Fluorescent staining with NeuN (red) and TUNEL (green) showed neuronal apoptosis in the cortex and striatum at day 3 after transplantation. Scale bar: 100 μm. **i** The bar graph shows the quantification of the percentage of TUNEL+ neurons in different groups. Graphs represent the individual data points, and data are presented as median ± 95% CI (**b**) or mean ± SEM (**c, d, e, g, i**). Behavior tests and infarct volume analysis are derived from $n = 8$ (**b, c, d, e, f, g**) biologically independent animals and confocal images are representative of $n = 4$ (**h, i**) independent animals. $P$-value (*$P < 0.05$, **$P < 0.01$, ***$P < 0.001$, ****$P < 0.0001$) was calculated by one-way (**b, g**) or two-way ANOVA with Tukey post hoc test for multiple comparisons (**c, d, e, i**). Source data are provided as a Source Data file.

ipsilateral hemisphere (Fig. 7c). Similar results were obtained in the fluorescent-dextran assay (Fig. 7d) and the measurement of fibrinogen/albumin deposits (Fig. 7e). We also tried to examine whether the transplantation of HBVPs or hPSC-CNC PCs suppressed transcytosis within the brain endothelium following tMCAO by transmission electron microscopy (TEM)[34]. Compared to the sham group, the average number of caveolae in each TEM image was much higher in the PBS and HDF groups after tMCAO.

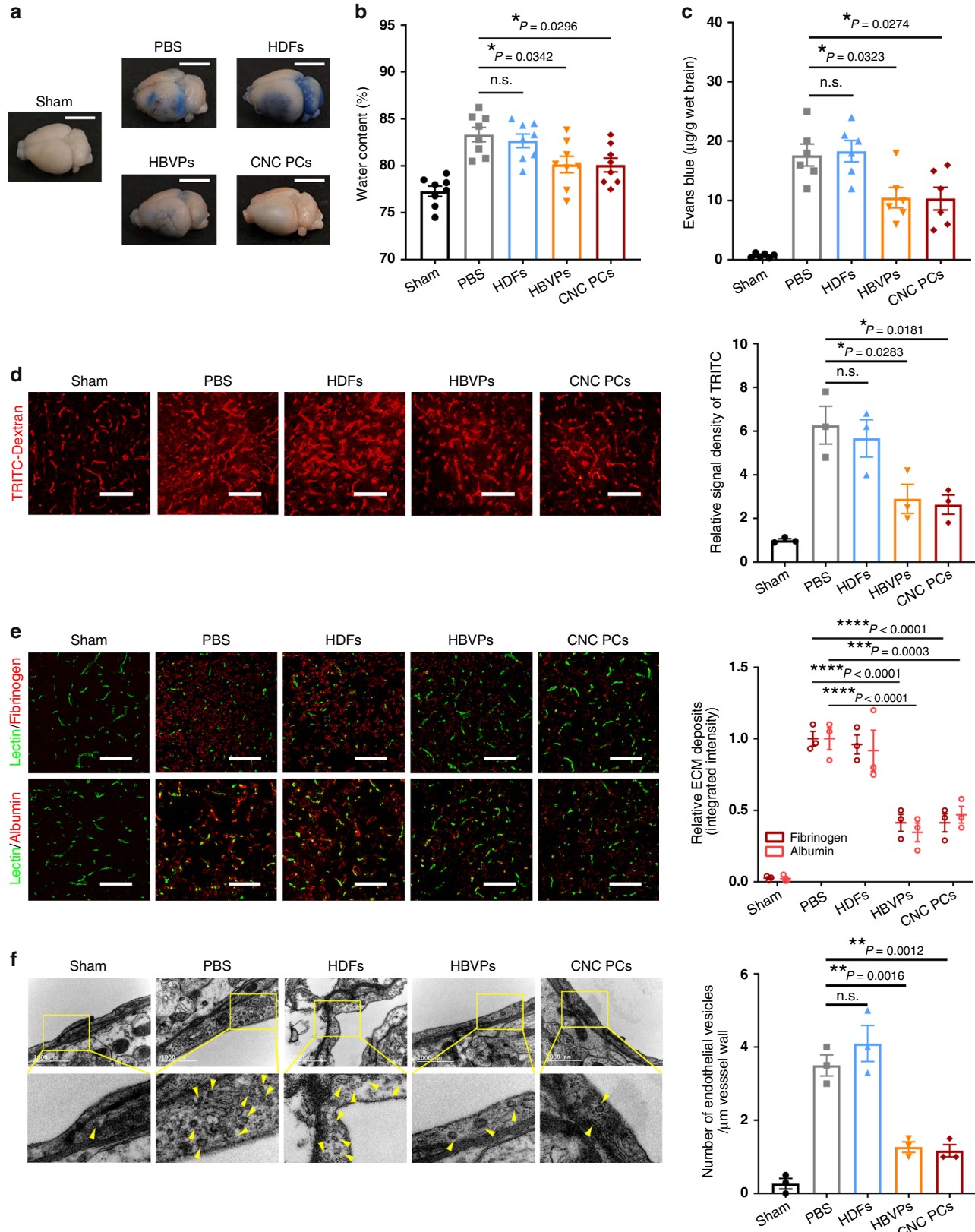

Intravenous infusion of HBVPs or hPSC-CNC PCs remarkably decreased the number of caveolae compared to the number in the PBS group ($P < 0.01$) (Fig. 7f).

As the barrier function of the BBB in tMCAO mice was effectively restored after treatment with pericyte-like cells, we then investigated the pericyte coverage of capillaries in the ischemic brain, which inversely correlates with BBB permeability[8], using dual immunostaining for NG2 and CD31. The results indicate that a loss of pericyte coverage was noted in the PBS group, while treatment with hPSC-CNC PCs and HBVPs substantially increased the pericyte coverage when compared with the PBS group. However, no significant difference in pericyte

**Fig. 7 hPSC-CNC PCs preserve BBB integrity in ischemic stroke mice. a** Representative images of the gross appearance of cerebral edema and Evans blue leakage. Scale bar: 1 cm. **b** The brain water contents in different groups were measured. **c** Statistical analysis of Evans blue extravasation using spectrofluorometry. **d** Representative images and quantification analysis of TRITC-dextran tracer extravasation assay. Scale bar: 100 μm. **e** Representative confocal microscopy and quantification analysis showing the extravascular fibrinogen (red) and albumin (red) leakage through lectin-labeled (green) capillaries. Scale bar: 100 μm. **f** Representative transmission electron microscopy (TEM) images and quantification analysis of vesicle number in brain endothelial cells. Scale bar: 1000 nm. Graphs represent the individual data points, and data are presented as mean ± SEM. Gross images of water content and Evans blue extravasation are representative of $n = 8$ (**a**, **b**, **c**) biologically independent animals. Confocal and TEM images are representative of $n = 3$ (**d**, **e**, **f**) independent animals. *P*-value (*$P < 0.05$, **$P < 0.01$, ***$P < 0.001$, ****$P < 0.0001$) was calculated by one-way (**b**, **c**, **d**, **f**) or two-way ANOVA with Tukey post hoc test for multiple comparisons (**e**). Source data are provided as a Source Data file.

coverage was observed between the HDF group and PBS group (Fig. 8a, b). Western blotting for the detection of TJ proteins showed that the protein levels of GLUT1, ZO-1, and Occludin were sharply downregulated in the PBS and HDF groups, while the injection of hPSC-CNC PCs or HBVPs significantly increased their expression (Fig. 8c). In addition, an immunofluorescence staining assay demonstrated that the disruption of TJ proteins was evident in the ischemic hemisphere in the PBS group and no obvious improvement was observed after HDF treatment, while infusion with hPSC-CNC PCs or HBVPs drastically alleviated the degradation of TJ proteins in the NVU (Fig. 8d). Quantitative analysis also proved that the administration of hPSC-CNC PCs or HBVPs effectively increased the TJs coverage area (normalized by CD31–positive endothelial area) after stroke (Fig. 8e). The above evidence suggests that BBB integrity was severely disrupted after damage and that BBB barrier function was effectively restored by treatment with HBVPs or hPSC-CNC PCs.

Further, to verify the distribution and characteristics of the hPSC-CNC PCs in vivo, the migration ability of DsRedE2+ transplanted cells toward the damaged sites was evaluated in tMCAO mice. We found that the number of the transplanted hPSC-CNC PCs (DsRedE2+ cells) increased gradually after cell transplantation (~1% of total cells on day 7; Fig. 9a) and most of the DsRedE2+ cells were located at the capillaries in the penumbra area (Fig. 9b). To identify the anatomical location of the transplanted hPSC-CNC PCs, we used fluorescent lectin-Dylight488 to label blood vessels and we revealed that the number of vessels (green) covered by the transplanted hPSC-CNC PCs (red) was ~10% 1 day post cell infusion, while ~30% of the capillaries in the penumbra area were surrounded by DsRedE2+ hPSC-CNC PCs 7 days post-treatment (Fig. 9c). Furthermore, immunofluorescence staining demonstrated that most of the transplanted DsRedE2+ hPSC-CNC PCs still expressed the pericyte-specific markers PDGFRβ and NG2 and localized near the lectin-Dylight488+ endothelial cells (Fig. 9d; and Supplementary Fig. 9b and Supplementary Movie 3) but were negative for the expression of NeuN or GFAP (Supplementary Fig. 9c). The above evidence confirms the perivascular localization of the transplanted hPSC-CNC PCs, which could migrate toward the site of injury in the BBB and directly participate in the reconstruction of the NVU of the BBB in vivo.

In this study, we use cyclosporine A as an immunosuppressant for the human pericyte treatment in the xenogeneic murine model of ischemic stroke. It was reported that cyclosporine A could promote the reconstruction of BBB and improve functional and structural neuronal changes through inhibiting CypA-NFκB-MMP9 pathway in brain pericytes in Apoe−/− and ApoE4 mice[35]. To rule out such a possibility, we performed additional experiments to evaluate the therapeutic effects of cyclosporine A in tMCAO model. Our results showed that cyclosporine A treatment alone displayed minimal therapeutic potential in tMACO mice (Supplementary Figs. 10 and 11). In addition, western blotting showed that the protein level of Cyclophilin A (CypA) did not change obviously among different groups, and

infusion of HBVPs+CSA or CNC PCs+CSA could strongly down-regulate the MMP9 protein level compared with PBS, CSA, and HDFs+CSA groups. We also found that NFκB nuclear translocation (indicating NFκB activation) happened in other cell types instead of NG2+ pericytes in penumbra area (Supplementary Fig. 11). The above evidence suggests that the CypA-NFκB-MMP9 pathway may not be involved in the pathogenesis of BBB disruption in tMCAO mice.

**MDK partially mediates the neuroprotective function of hPSC-CNC PCs in tMCAO mice**. To explore the underlying therapeutic mechanism of pericyte-like cells in the tMCAO model, we analyzed the RNA-seq data and found that hPSC-CNC PCs highly expressed midkine (*MDK*; TPM values > 350; Supplementary Fig. 12a, b), which is a retinoic acid-responsive gene that could enhance neurite outgrowth, neuronal cell survival and plasminogen activator activity[36]. To investigate whether hPSC-CNC PC-derived MDK participates in the neuroprotective function, we generated a knockdown of *MDK* in hPSC-CNC PCs by RNA interference. shMDK1 reduced *MDK* expression more efficiently than shMDK2 at the mRNA level (Supplementary Fig. 12c). MDK expression in shMDK1 hPSC-CNC PCs was successfully knocked down (Supplementary Fig. 12d–f) and these cells maintained their contractility and surface marker expression profile (Supplementary Fig. 12g, h). Compared to the NTC group, MDK knockdown significantly impaired the neuroprotective effects of hPSC-CNC PCs, as shown by neurological deficit score assessment (Supplementary Fig. 13a), neuron apoptosis rate (displayed by active caspase-3/NeuN coimmunostaining; Supplementary Fig. 13b), the infiltration of Evans blue dye (Supplementary Fig. 13c), and the decrease in coverage area (normalized by the CD31–positive endothelial area; assessed by the immunostaining) and the amount of TJs (assessed by western blotting of TJ proteins; Supplementary Fig. 13d–f). Taken together, MDK contributes significantly to CNC PC-mediated neuroprotection, which might provide clues for elucidating the potential mechanisms of hPSC-CNC PCs restoring BBB integrity in the tMCAO model.

## Discussion

In this study, we successfully derived pericyte-like cells with cranial neural crest origin (hPSC-CNC PCs) from human pluripotent stem cells. We showed that hPSC-CNC PCs possessed comparable contractile properties, vasculogenic capacity, and endothelial barrier function as primary HBVPs. Strikingly, our results demonstrated that the transplantation of hPSC-CNC PCs in a tMCAO murine model could promote neurological functional recovery by the reconstruction of the BBB, and reducing neuron apoptosis.

Forebrain pericytes are derived from cranial neural crest cells[15], while the neural crest is a population of multipotent stem cells that arise from the neural plate border during the gastrula stage. The neural crest can be divided into five populations,

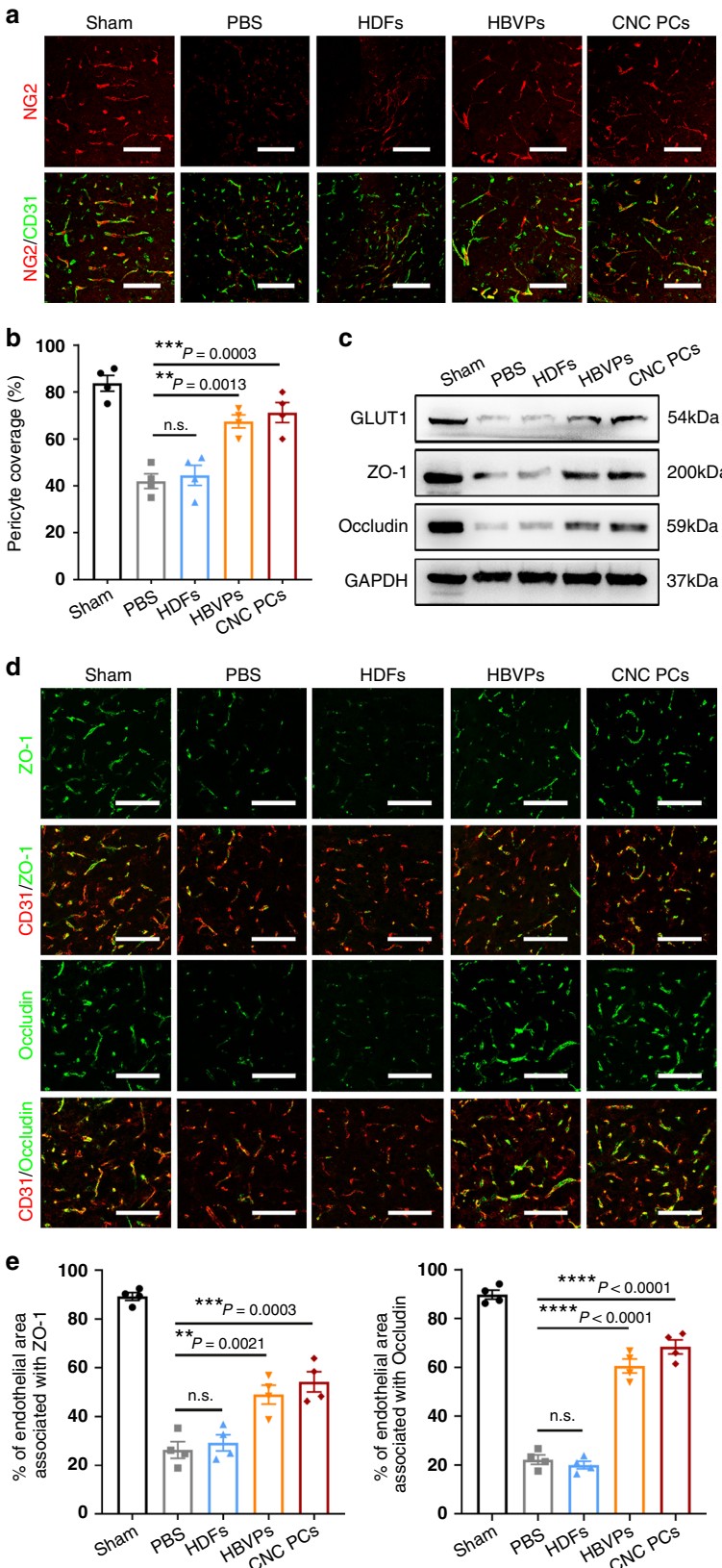

cranial, cardiac, vagal, trunk and sacral, in the human body. It has been reported that cranial, vagal and trunk neural crest stem cells could be successfully obtained from hPSCs[17,18,37]. In this study, we used defined medium supplemented with SB431542 (a potent and selective inhibitor of the transforming growth factor-β

signaling pathway) and CHIR99021 (a WNT signaling activator)[21] for differentiation of hPSCs into cranial neural crest cells. We found that a 6-day induction protocol could drive the commitment of hPSCs into p75[bright]HNK1[+] neural crest cells with an efficiency (>80%) comparable to that in previous studies[19,21].

**Fig. 8 hPSC-CNC PCs preserve BBB integrity in ischemic stroke mice. a** Representative confocal microscopy images of different experimental groups showing NG2[+] (red) pericytes extended over CD31[+] (green) capillaries. Scale bar: 100 μm. **b** Quantification of the percentage of CD31[+] capillaries covered by NG2[+] pericytes. **c** Expression of the BBB marker GLUT1 and TJ proteins including ZO-1 and Occludin was evaluated in the ipsilateral hemisphere using microvessel western blotting 3 days after cell transplantation. **d** Coimmunofluorescence staining for CD31 (red) and tight junction protein markers, including Occludin (green) and ZO-1 (green), in the peri-infarcted regions. Scale bar: 100 μm. **e** The quantification of TJ coverage area was presented as the percentage of CD31-labeled endothelial area associated with ZO-1/Occludin. Graphs represent the individual data points, and data are presented as mean ± SEM. Confocal images are representative of $n = 4$ (**a**, **b**, **d**, **e**) biologically independent animals and immunoblotting images are representative of $n = 3$ (**c**) biological replicates. *P*-value (*$P < 0.05$, **$P < 0.01$, ***$P < 0.001$, ****$P < 0.0001$) was calculated by one-way ANOVA with Tukey post hoc test for multiple comparisons (**b**, **e**). Source data are provided as a Source Data file.

CNCs isolated by FACS maintained the features of the neural crest during in vitro expansion, including the expression of neural crest-specific genes and multilineage differentiation potency. In particular, we discovered that the p75[bright]HNK1[+] CNCs also highly expressed cranial neural crest markers, including *HOXA1*, *LHX5*, *PAX3*, *PAX7*, and *SOX10*. The above evidence demonstrated that putative cranial neural crest cells were successfully generated from hPSCs.

Two elegant studies also demonstrated the protocol for differentiation of pericyte-like cells with cranial neural crest origin from hPSCs[17,18]. Stebbins et al.[17] found that treatment with E6 medium supplemented with several small molecules for 15 days yielded ~90% p75[+]/HNK1[+] NCSCs, which were further purified for pericyte differentiation. Compared to this study, our method could efficiently obtain putative cranial neural crest stem cells within a relatively short period (6 days). Faal et al.[18] reported that 5-day treatment with CHIR99021-containing serum-free medium could efficiently induce hPSCs to differentiate into p75[+]HNK1[+] cells (the quantitative data of the percentage of p75[+]HNK1[+] cells was not provided), and the resulting cells without FACS or MACS purification were then passaged and directly used for pericyte differentiation. Therefore, pericyte-like cells from other developmental origins (e.g., mesoderm) may exist in the heterogeneous cell population. Nonetheless, previous studies demonstrated there were at least two populations of p75[+] cells; only p75[bright]HNK1[+] cells with high levels of *AP2α* were authentic neural crest stem cells, while p75[dim]HNK1[+] cells expressing high levels of *PAX6* were actually neural stem cells[19,21]. We also observed similar gene expression patterns in p75[bright]HNK1[+] cells and p75[dim]HNK1[+] cells in our differentiation system (Supplementary Fig. 14). Thus, we isolated this subpopulation of p75[bright]HNK1[+] CNCs for pericyte differentiation. Taken together, we provide a fast and efficient method for the derivation of CNCs from hPSCs through distinctive developmental pathways.

Next, we showed that 14-day incubation with the commercial pericyte medium supplemented with PDGF-BB successfully induced p75[bright]HNK1[+] CNCs to be PDGFRβ and NG2 double-positive pericyte-like cells with a similar efficiency (>90%) as previously reported[17,18]. RNA-Seq, qPCR, immunostaining and western blotting analyses demonstrated that hPSC-CNC PCs shared similar gene expression patterns with HBVPs. Intriguingly, we found that hPSC-CNC PCs also expressed PDGFRA, which was also recognized as an oligodendrocyte and fibroblast marker. This data was similar with that of previous studies as revealed by their transcriptome analysis[17,38]. Meanwhile, our RNA-Seq data indicates that the CNC PCs and HBVPs did not express other oligodendrocyte lineage markers *SOX10* (the TPM value is about 0.3–1) or *OLIG2* (the TPM value is about 0–0.1). More importantly, we found that the vast majority of PDGFRA[+] CNC PCs and HBVPs (>90%) also coexpressed pericyte markers, including PDGFRβ, CD146, CD13, and NG2 as detected by FACS analysis (Supplementary Fig. 15). These results indicate that CNC PCs resemble the primary human brain pericytes in the expression of cell surface markers.

In addition, we demonstrated that hPSC-CNC PCs possessed various pericyte-specific functional properties. hPSC-CNC PCs displayed stronger ability of vascular stabilization in vitro than HBVPs, although their potential to support vessel growth and maturation in vivo did not differ significantly. Thus, the vasculogenic potential may allow hPSC-CNC PCs to be candidates for the treatment of peripheral vascular diseases. The hPSC-CNC PCs also closely resemble HBVPs in contractile ability, which may be attributed to the expression of contractile proteins, including Vimentin, Caldesmon, tropomyosin (TPM1, TPM2, TPM3), and myosin (MYH9, MYH10)[17,39], as illustrated by qPCR and/or RNA-Seq data, although other contractile proteins (αSMA and calponin) were absent or were rarely expressed in hPSC-CNC PCs. More importantly, when cocultured with BMECs, hPSC-CNC PCs behaved as primary HBVPs and significantly enhanced endothelial cell barrier function. This barrier function enhancement property of hPSC-CNC PCs would be of great value for modeling the human BBB in vitro, since the current understanding of the cellular and molecular mechanisms associated with the BBB during development or neurological disorders are greatly hampered by the limited availability of human cells and tissues. The above results demonstrated that hPSC-CNC PCs derived from our protocol were functional pericyte-like cells.

It was reported that brain pericytes had some distinct characteristics in contrast to their peripheral counterparts. First, brain pericytes help to seal the BBB endothelium by regulating the expression of endothelial TJ proteins, which results in low rate of bulk-flow transcytosis. However, the capillary endothelium in peripheral organs is leaky[1,40]. We discovered that our hPSC-CNC PCs could strengthen BBB integrity by increasing the expression level or inhibiting the disruption of TJ proteins and suppressing transcytosis in the BBB endothelium both in vitro and in vivo. Additionally, Vanlandewijck et al.[28] revealed that CD13 was not expressed by lung pericytes although these peripheral pericytes shared canonical pericyte markers Pdgfrb, Cspg4, and Des with brain pericytes. Moreover, they found that many members of different types of transporters were abundant in brain pericytes but were low or absent in lung pericytes. In our study, almost all of the hPSC-CNC PCs were positive for CD13 and highly expressed some of the solute carrier (SLC) transporters (*SLC1A5*, *SLC2A10*, *SLC16A1*, and others), ATP-binding cassette (ABC) transporters (*ATP1B3*, *ATP13A1*, *ATP2C1*, and others), and ATPase (ATP) family members (*ABCA1*, *ABCA2*, *ABCA7*, and others), as illustrated by RNA-Seq data (Supplementary Fig. 8d). We also demonstrated that posterior HOX genes were absent in hPSC-CNC PCs, while hPSC-MP PCs highly expressed *HOX10-12*. Moreover, the therapeutic effects of hPSC-CNC PCs were also superior to hPSC-MP PCs, as evidenced by the neurological rating score, ECM (fibrinogen and albumin) deposits, and TRITC-dextran leakage in tMCAO mice (Supplementary Fig. 16). These data indicate that the hPSC-CNC PCs exhibit the distinct characteristics and function of brain pericytes.

The dysfunction of brain pericytes has been implicated in various neurodegenerative diseases, including stroke, AD, Parkinson's

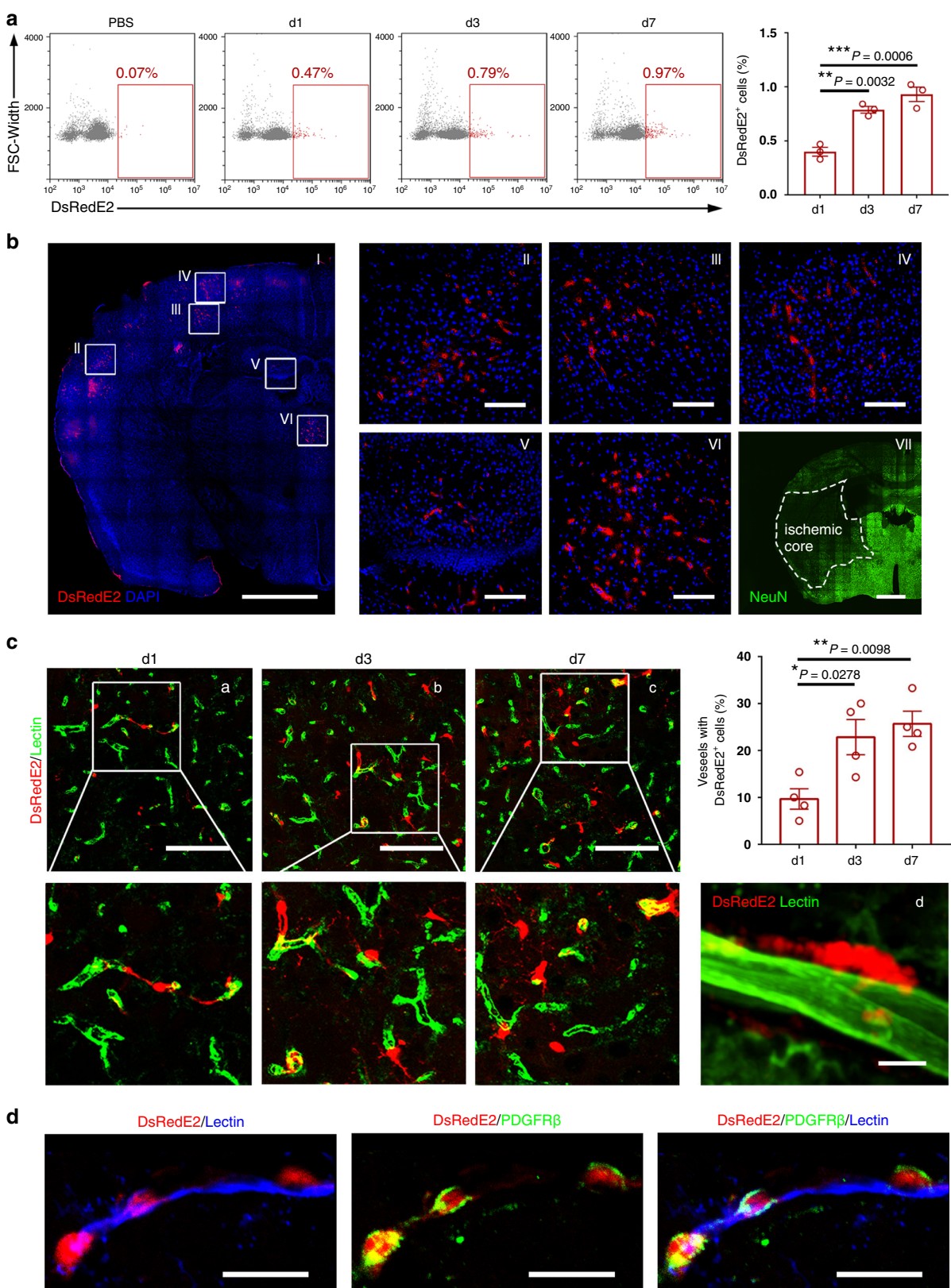

disease (PD), amyotrophic lateral sclerosis (ALS), and Huntington's disease[1,12–14]. These results suggest that engraftment of brain pericytes could achieve therapeutic effects in various BBB-related neurological disorders in vivo, but the limited sources of pericytes restrict their therapeutic applications. For example, ischemic stroke happens when the blood flow to the brain is blocked and represents one of the major causes of death and disability all over the worldwide[41]. Stroke causes the death of pericytes during the early phase, which in turn leads to increased BBB permeability and brain edema[42]. In the later recovery phase of stroke, pericytes also contribute to angiogenesis and neurogenesis and thereby promote neurological recovery[42]. Nonetheless, whether the transplantation

**Fig. 9 Temporal and spatial dynamics of infused CNC PC accumulation in tMCAO mice. a** Temporal and quantitative characterization of the infused CNC PCs by whole brain FACS for the PBS-treated control at 1, 3, and 7 days posttransplantation. **b** Representative images showing that DsRedE2-labeled CNC PCs were mainly distributed in the penumbra area of the ipsilateral hemisphere at 3 days posttransplantation (**a–f**). Immunostaining of the NeuN+ (green) neurons was applied to identify the area of the ischemic core (**g**). Scale bar: 100 μm (**b–f**); 1000 μm (**a, g**). **c** Temporal dynamics of DsRedE2+ CNC PC accumulation in the ipsilateral hemisphere. Representative confocal microscopy images showing the DsRedE2+ CNC PCs distributed and extended over the lectin-labeled capillaries. Scale bar: 100 μm (**a–c**); 10 μm (**d**). **d** Transplanted DsRedE2+ CNC PCs maintained the expression of the pericyte-specific marker PDGFRβ (violet). Scale bar: 10 μm. Graphs represent the individual data points, and data are presented as mean ± SEM. Confocal images are representative of $n = 4$ (**b, c, d**) biologically independent animals and FACS images are representative of $n = 3$ (**a**) biologically independent animals. P-value (*$P < 0.05$, **$P < 0.01$, ***$P < 0.001$, ****$P < 0.0001$) was calculated by one-way ANOVA with Tukey post hoc test for multiple comparisons (**a, c**). Source data are provided as a Source Data file.

of cranial neural crest-derived forebrain pericytes could offer a potential therapeutic benefit in the recovery from ischemic stroke remains to be elucidated.

Here, we focus on the functional assessment of hPSC-CNC PCs and demonstrate their in vivo therapeutic potential in a tMCAO mouse model. We provide the first evidence that transplanted hPSC-CNC PCs could significantly reduce the infarct size and promote neurological recovery by evaluating improvements in neurological deficits, locomotor coordination, asymmetry in forelimb use, and sensorimotor activity. Histological examination of brain samples showed that intravenously injected hPSC-CNC PCs were efficiently recruited to capillaries in the penumbra area, which may be attributed to blood reperfusion following vessel occlusion, BBB disruption after ischemia-reperfusion injury[34], and PDGF-B/PDGFRβ interaction[43]. BBB integrity was significantly restored by the recruited hPSC-CNC PCs, as revealed by reduced brain edema, decreased plasma protein deposits, and diminished leakage of TRITC-dextran and Evans blue dye compared to the control groups. Our results further showed that the apoptosis of neurons was markedly reduced after cell infusion, which may be due to the restoration of BBB integrity and the neuroprotective function exerted by injected hPSC-CNC PCs. A previous study suggests that pericyte ablation causes a rapid neurodegeneration cascade due to the loss of pericyte-derived pleiotrophin (PTN)[44]. Although *PTN* was expressed at a very low level, we found that *MDK*, the homolog of *PTN*, was highly expressed in hPSC-CNC PCs (Supplementary Fig. 10a, b). Further experiments demonstrate that MDK contributes significantly to CNC PC-mediated neuroprotection, which might provide clues to elucidate the potential mechanisms of hPSC-CNC PCs restoring BBB integrity in the tMCAO model.

The above results indicate that hPSC-CNC PCs may represent an ideal cell source for cell therapy in BBB-related neurological diseases, such as stroke and Alzheimer's disease. In this study, hPSC-CNC PCs were delivered by the intravenous (IV) route with a dose of $1 × 10^6$ cells/mouse, due to the relative ease of administration and limited risk. IV infusion certainly may result in limited numbers of cells reaching target tissues because of a large number of cells entrapment in the lungs[45,46]. We did observe low efficiency of engraftment in the present study and found that only ~1% of the cells were DsRedE2+ and ~30% of the capillaries were surrounded by DsRedE2+ CNC PCs 7 days posttreatment in the penumbra area. In addition, systemic injection of cells for stroke entails the potential risk about cells generating emboli or thrombi[45,46]. Therefore, the comparative studies for the optimal delivery routes and factors such as cell size, cell dosage and delivery speed for ensuring the efficacy and safety of pericyte transplantation need to be addressed in the future.

In conclusion, we demonstrate that cranial neural crest-derived pericyte-like cells could efficiently promote neurological recovery and rescue BBB barrier function in a tMCAO model, which may represent a promising cell source for the treatment of BBB dysfunction-related disorders and would help to establish an in vitro human BBB model for studies of the contribution of genetic and environmental factors associated with BBB pathophysiology.

## Methods

**Human ESC/iPSC culture.** The human-induced pluripotent stem cell (hiPSC) line was generated previously by the transduction of human embryonic fibroblasts with lentiviral vectors expressing OCT4, KLF4, SOX2, and c-MYC (OKSM)[47]. The human embryonic stem cell (hESC) lines H1 and H9[48] were obtained from WiCell Research Institute. Both hiPSCs and hESCs were maintained in feeder-free conditions using mTeSR1 medium (Stem Cell Technologies, Vancouver, Canada) on human embryonic stem cell (hESC)-qualified Matrigel (BD Bioscience, San Diego, CA). Cells were passaged every 4–5 days with 0.5 mM EDTA (Invitrogen, Carlsbad, CA, USA). All cells used had a normal diploid karyotype.

**Cranial neural crest differentiation of hPSCs.** A step-by-step protocol describing the differentiation process can be found at Protocol Exchange[49]. To initiate differentiation, hESCs/hiPSCs were harvested using Accutase (Invitrogen) and plated onto Matrigel (BD Bioscience) coated dishes at a density of $10^4$ cells/cm² in chemically defined N2B27 medium (N2B27-CDM) containing 95% DMEM/F12 medium, 1% N2 supplement, 2% B27 supplement, 1% Glutamax, 1% MEM-nonessential amino acids, 55 μM 2-mercaptoethanol (all from Invitrogen), 10 μM Y27632 (Stem Cell Technologies), and 20 ng/ml basic fibroblast growth factor (bFGF) (Peprotech, Rocky Hill, New Jersey, USA). Twenty-four hours later, the culture was switched to NCN2 medium containing N2 supplement, CHIR99021 (1.0 μM), and SB431542 (2.0 μM)[21] and cultured for 6 days. Then the cells were dissociated and labeled with antibodies against CD57 (also known as HNK1/B3GAT1) and low-affinity nerve growth factor receptor (NGFR, also known as p75), both from BD-Pharmingen (Palo Alto, CA, USA) for fluorescence-activated cell sorting (FACS) using a BD Influx cell sorter (BD-Pharmingen). p75bright HNK1+ cranial neural crest stem cells (CNCs) were sorted and replated onto poly-L-ornithine (15 mg/ml; Sigma-Aldrich, St. Louis, MO, USA) and fibronectin (10 mg/ml; Millipore, Temecula, CA, USA) (PO/FN)-coated dishes for adherent culture with neural crest culture medium (NCCM) containing 1% N2 supplement, 2% B27 supplement, 20 ng/ml bFGF, and 20 ng/ml epidermal growth factor (EGF; Peprotech). The detailed scheme of the differentiation protocol is presented in Supplementary Fig. 1.

**Multilineage differentiation of hPSC-derived CNCs.** For differentiation toward peripheral neurons, EGF/bFGF-expanded CNCs were cultured in NCCM lacking bFGF and EGF and supplemented with 10 ng/ml brain-derived neurotrophic factor (BDNF), 10 ng/ml glial cell line-derived neurotrophic factor (GDNF), 10 ng/ml nerve growth factor (NGF) and 10 ng/ml neurotrophin-3 (NT3) (all from Peprotech), as well as 200 μM ascorbic acid (AA) and 0.5 mM dibutyryl-cAMP (db-cAMP) (both from Sigma-Aldrich). Cells were induced to undergo neuronal differentiation for 4 weeks, and media were changed every 2–3 days. Differentiated cells were analyzed for the expression of neural markers by immunocytochemistry.

Schwann cell differentiation potential was assessed by first culturing CNCs in NCCM for at least 2 months[22]. Cells were then induced to differentiate into Schwann cells by culturing in NCCM lacking bFGF and EGF and supplemented with 10 ng/ml ciliary neurotrophic factor (CNTF; Peprotech), 20 ng/ml neuregulin (Sigma-Aldrich), and 0.5 mM db-cAMP for 30 days. The media were changed every 2–3 days. Cells were then examined for the expression of Schwann cell markers by immunostaining.

For mesenchymal differentiation, CNCs were cultured for >2 weeks on uncoated tissue culture dishes in low-glucose Dulbecco's modified Eagle's medium (L-DMEM; Gibco, Grand Island, NY, USA) containing 10% fetal bovine serum (FBS; Gibco). The phenotype and multipotency of CNC-derived mesenchymal stem cells (CNC-MSCs) were assessed by FACS analysis and subsequent assay of their ability to differentiate into cells of the mesenchymal lineage (osteoblasts, adipocytes, chondrocytes, and smooth muscle cells)[50]. For osteogenic differentiation, CNC-MSCs were cultured in low-glucose-DMEM containing 10 mM β-glycerol phosphate, 10 nM dexamethasone and 200 mM ascorbic acid (all from Sigma-Aldrich) for 14–21 days. For adipogenic differentiation, CNC-MSCs

were cultured in high-glucose-DMEM containing 1 mM Dex, 10 μg/ml insulin, and 0.5 mM 3-isobutyl-1-methylxanthine (all from Sigma-Aldrich) for 28 days. For chondrogenic differentiation, CNC-MSCs were were suspended in a 15-ml conical tube in containing medium containing 1% FBS, 10 ng/ml transforming growth factor-β3 (Peprotech) and 1% N2 (Life Technologies) for 4 weeks. For smooth muscle cell differentiation, CNC-MSCs were incubated in N2B27 medium containing 2.5 ng/ml TGF-β1 and 5 ng/ml human platelet-derived growth factor-BB (PDGF-BB) (both from Peprotech) for 7–14 days.

**Pericyte differentiation of hPSC-CNCs.** CNCs were dissociated using Accutase (Gibco) and plated onto PO/FN-coated dishes at a density of $10^5$ cells/cm$^2$ in NCCM supplemented with 10 μM Y27632. When differentiation was initiated, the culture was switched to commercial Pericyte Medium (ScienCell Research Laboratories, Carlsbad, CA, USA) containing 10 ng/ml bFGF and 50 ng/ml PDGF-BB for 7–14 days. The first confluent culture at this time point was denoted as passage 1. Cells were dissociated by Accutase solution, split at 1:4 in Pericyte Medium, and cultured on PO/FN-coated plates.

We also generated mesoderm progenitor-derived pericyte-like cells (MP PCs) from hPSCs. Confluent hPSCs were dissociated by EDTA and cultured on Matrigel-coated plates in mTeSR1 medium at a density of 40,000 cells/cm$^2$. 24 h later, cells were cultured in mesodermal induction media (MIM, Stem Cell Technologies) for 5 days to generate mesoderm progenitors (MPs), and MPs were further induced to differentiate into pericyte-like cells using pericyte medium (ScienCell Research Laboratories) for 1–2 weeks[18].

**In vitro cord formation assay.** Human brain vascular pericytes (HBVPs, ScienCell Research Laboratories; Catalog #1200; isolated from human embryonic brain tissue) served as a positive control of hPSC-CNC PCs in functional assays in this study. HBVPs were isolated from the human brain and cryopreserved at passage one after purification, according to the manufacturer's instructions. Human brain microvascular endothelial cells (HBMECs, $3 \times 10^4$ cells/well) (ScienCell Research Laboratories) were co-seeded with CNC-derived pericyte-like cells (hPSC-CNC PCs) expressing DsRed-Express2 (DsRedE2; $1.5 \times 10^4$ cells/well) or DsRedE2-HBVPs on pre-solidified Matrigel (BD Bioscience) in Endothelial Cell Medium (ScienCell Research Laboratories) containing 20 ng/ml vascular endothelial growth factor (VEGF). The cells were incubated for different time periods at 37 °C and 5% CO$_2$ in a humidified atmosphere. The vascular network was photographed at the indicated time points using a BioTek Lionheart FX Automated Live Cell Imager (BioTek Instruments, Inc., Winooski, VT, USA).

**Gel contraction assay and Carbachol treatment assay.** Cell contractility was assessed by the gel lattice contraction assay[51]. Briefly, 2 volumes of type I collagen solution (5 mg/ml; Upstate Biotechnology Inc., Lake Placid, NY, USA) were mixed with 1 volume of 5 x PBS and 2 volumes of 0.1 N NaOH on ice to yield 2 mg/ml of collagen solution at pH 7.4. A cell suspension was then made in the collagen solution on ice ($1 \times 10^6$ cells/ml). The cell-collagen mixture was added to a 24-well plate at 0.5 ml per well and incubated for 2 h at 37 °C for gelling, followed by the addition of medium over the gel. After allowing the cells to spread within the gel overnight, the gels were gently detached, lifted from the bottom of the well and photographed. Human bone marrow-derived mesenchymal stem cells (HMSCs; established in our laboratory) and human aortic smooth muscle cells (HASMCs; isolated from human aorta and cryopreserved at passage one; purchased from ScienCell Research Laboratories, Catalog # 6110) were utilized as negative control and positive control, respectively. The area of the gel lattices was determined with ImageJ software (NIMH, Bethesda, MD), and the relative lattice area was obtained by dividing the area at 48 h of culture by the initial area of the lattice and graphing.

For the carbachol treatment assay, 1 mM carbachol (Sigma) was added to the cell culture medium and time-lapse images were recorded for 15 min following carbachol addition using live cell imaging[31]. The percentage of contractile cells was measured by using ImageJ software (NIH). The percentage of contracting cells was determined from five different optical fields.

**Coculture of hPSC-CNC PCs with HBMECs.** Twelve-well Transwell inserts (Corning Life Sciences, MA, USA) were coated with collagen (1 μg/cm) and fibronectin (10 μg/ml) for at least 4 h at 37 °C. hPSC-CNC PCs or HBVPs ($3 \times 10^4$ cells/cm$^2$) were seeded on the bottom side of the inserts and incubated at 37 °C in 5% CO$_2$ for 2 h to adhere firmly. Then HBMECs were plated onto the upper side of the inserts at a density of $3 \times 10^5$ cells/cm$^2$.

For the quantification of frayed TJs in cultured cells, cells were defined as having frayed tight junctions if greater than 10% of the immunolabeled tight junction protrusions were not parallel to the cell–cell border[52,53]. The percentage of cells expressing frayed tight junctions was counted by a blinded observer and a minimum of three separate frames and 200 total cells were counted to obtain a percentage of frayed tight junctions.

TEER was measured as a function of time following the initiation of coculture. To improve measurement reproducibility and stability, it is important that the electrodes stay immersed in the culture media and to maintain a steady position. TEER measurements of inserts without cells should be subtracted from the TEER values obtained with inserts containing cells. TEER values are presented in Ωcm$^2$.

**Transcytosis assay.** To test the effect of hPSC-CNC PC coculture on HBMEC transcytosis, 10-kDa dextran labeled with Alexa 488 was used to quantify the level of transcytosis. Briefly, the level of transcytosis was quantified by detecting the fluorescence intensity in HBMEC medium on the basolateral side of the transwell after coculture for 48 h using a Tecan plate reader according to the manufacturer's instructions.

**Matrigel matrix implants.** A total of $2 \times 10^6$ hrGFP-expressing HBMECs (hrGFP-HBMECs) were mixed with $10^6$ DsRedE2-hPSC-CNC PCs or DsRedE2-HBVPs in 500 μl Matrigel (growth factor reduced; BD Biosciences) containing different growth factors (250 ng/ml each of VEGF and bFGF). The matrices were injected subcutaneously on each side lateral to the abdominal midline region of 8- to 10-week-old NOD-SCID mice. Two weeks later, mice were sacrificed to retrieve implants. The implants were fixed overnight in 4% paraformaldehyde (PFA), sectioned and stained with DAPI. Blood vessels were counted in 6 microscopic fields and averages were taken. All animal procedures were performed under protocols approved by the Sun Yat-Sen University Institutional Animal Care and Use Committee.

**Stroke model and pericyte-like cell transplantation.** All animal experimental procedures were approved by the Sun Yat-Sen University Animal Use and Care Committee. Male C57BL/6 mice aged 8 to 12 weeks (weight 18–25 g) were purchased from Guangdong Medical Laboratory Animal Center and housed under standard specific-pathogen-free (SPF) conditions in a temperature-, humidity- and light cycle-controlled facility (20 ± 2 °C; 50 ± 10%; 12-h light/dark cycle) and free access to food and water. A total of 243 male mice were used in this study, including 9 animals excluded from further experiments (due to unsuccessful surgery). The mortality rate was 22.2% (52/234) on day 3 and 30.4% (28/92) on day 7 postsurgery in the MCAO group.

To induce the focal ischemic stroke model[54], mice were anesthetized with 1.5% isoflurane in a 30% O2/69% N2O mixture. Prior to surgery, all surgical instruments and supplies were sterilized. A 10-mm incision was made on the right side of the neck and the carotid artery, external carotid artery (ECA), and internal carotid artery (ICA) was exposed. A silicone-coated filament was inserted into the ECA and then through the ICA to block the MCA[55]. The cerebral blood flow (CBF) was determined by laser Doppler flowmetry to confirm vascular occlusion. Mice with a CBF reduction less than 70% were excluded from further assessments. After 40 min, the filament was removed for reperfusion, and the incision was carefully sutured. Sham-operated mice underwent the same isoflurane anesthesia and surgical procedures as the MCAO group, except the insertion of an intraluminal filament. The body temperature of all experimental mice was controlled by heating pad at 37 ± 0.5 °C during the surgery and recovery period.

The inclusion criteria for tMCAO were as follows: 1) regional cerebral blood flow decreases >70% during occlusion as detected by laser Doppler flowmetry; and 2) neurological deficits 3 h after MCAO with a neurological score between ~2 and 3 points[56].

Included stroke mice were assigned randomly to different groups (Sham, PBS, HDFs, HBVPs, hPSC-CNC PCs). Before transplantation, confluent cells were trypsinized, triturated repeatedly and passed through a 40 μm cell strainer to enrich for single cells. Cells were counted and visualized on a hemocytometer. If cell clumps were observed, cells were passed through a 25-gauge needle 10 times. In consideration of ease of administration and minimal invasiveness, we performed the cell transplantation to evaluate the therapeutic effects via tail vein administration[46] and chose a commonly used cell dose of $1 \times 10^6$ cells/mouse referring to the dosages of IV-delivered MSCs for treating the mouse stroke models[57,58]. Then injection of PBS, human dermal fibroblasts (HDFs), mesoderm progenitor-derived pericyte-like cells from hPSCs[18], HBVPs or hPSC-CNC PCs were injected 6 h after tMCAO via the caudal vein. The animals were injected subcutaneously with cyclosporin A (CSA, 10 mg/kg/day body weight, Sandoz, Switzerland) to suppress the immune rejection of transplanted cells according to previous protocols[59]. All animal experiments were approved by the Ethical Committee of Sun Yat-Sen University.

The infarct volume was determined by staining with 2,3,5-triphenyltetrazolium chloride (TTC, Sigma-Aldrich) and measured by ImageJ analysis software[55].

**Determination of BBB integrity.** The BBB integrity was determined by Evans blue staining, fluorescent-dextran assay, and the measurement of fibrinogen/albumin deposits. Briefly, 2% Evans blue dye (EBD, Sigma-Aldrich; 6 ml/kg in saline) or 70 kDa dextran conjugated to tetramethylrhodamine isothiocyanate (dextran-TRITC; 0.1 ml per mouse, 10 mg/ml in saline) was intravenously injected and allowed to circulate 3 h before the scarification. The leakage of EBD or dextran-TRITC in the ischemic brain tissue was analyzed 4 days after tMCAO (Sham and PBS group) or 3 days after cell transplantation. The concentration of Evans blue extracted from the ipsilateral hemispheres was quantified by spectrophotometry at 610 nm according to a standard curve[60]. The extravasation of plasma fibrinogen and albumin was detected by immunofluorescence staining and analyzed by ImageJ software.

**NG2$^+$ pericyte coverage and TJ formation analysis**. For pericyte coverage, NG2 and CD31 signals from microvessels (diameter < 10 μm) were selected and subjected to threshold processing. We assessed the areas occupied by the fluorescent signals by the area measurement tool of ImageJ. The pericyte coverage was calculated as a percentage (%) of NG2$^+$ pericyte surface area covering CD31$^+$ capillary surface area per observation field. In each experimental group, non-adjacent coronal sections of four mice were selected for the statistical analysis.

To evaluate the TJ formation after pericyte-like cell transplantation, ZO-1 or Occludin coverage area was calculated and normalized by the length of CD31/lectin–positive endothelial cells[61,62]. The quantification of TJ coverage area was presented as the percentage of CD31/lectin-labeled endothelial area associated with ZO-1/Occludin.

**Transmission electron microscopy**. Brain samples from the penumbra area of stroke cortex in different groups were dissected and fixed in 5% glutaraldehyde/4% PFA/0.1 M sodium-cacodylate for 1 h at room temperature (RT) followed by overnight fixation at 4 °C. Coronal vibratome free-floating sections of 50 μm were then postfixed in 1% osmium tetroxide and 1.5% potassium ferrocyanide, dehydrated, and embedded in epoxy resin. Ultrathin sections of 70 nm were then cut from the block surface, collected on copper grids, and counterstained with Reynold's lead citrate.

**Behavioral tests**. The evaluation of neurological deficit was performed on day 1, day 3, and day 7 after tMCAO: 0 = no deficit; 1 = forelimb weakness; 2 = circling to affected side; 3 = partial paralysis on affected side; and 4 = no spontaneous motor activity[63]. Moreover, corner test and adhesive removal test were performed to evaluate motor functional recovery at 1 day, 3 days and 7 days post stroke[32,33]. All the behavioral tests were carried out in a blinded fashion. For the corner test, the mouse was placed between two boards that were attached and formed at a 30° angle with a small opening. When both sides of the vibrissae were stimulated by the two boards, the mouse reared forward and upward, then turned back to face the open end. The turns in one versus the other direction were recorded from twenty trials for each test and repeated for three tests. Only turns involving full rearing along either board were counted. For the adhesive removal test, the mouse was placed into a transparent Perspex box for a 60 s habituation period. Two adhesive tapes were applied with an equal pressure on each animals paw so that they covered the hairless part of the forepaws. The order of placing the adhesive (right or left) was alternated between each animal and each session. The mouse was then replaced in the Perspex box and the times to contact and to remove each adhesive tape were collected with a maximum of 120 s. For the rotarod test, experimental mice were placed on an accelerating (4–40 r.p.m.) rotarod apparatus (Med Associates). Each animal was tested for three times at 1 day, 3 days and 7 days after stroke, according to the manufacturer's instructions. Latency to fall was manually determined by observers blinded to the experimental groups and indicated the quality of motor coordination. Statistical analysis was conducted using two-way analysis of variance (ANOVA).

**Brain water content**. The brain samples were collected 3 days after tMCAO and dried in an oven at 95 °C for 24 h. The weights of brain samples were measured before and after dehydration, and the brain water content was calculated as [(wet tissue weight − dry tissue weight)/wet tissue weight] × 100%[32].

**Lentiviral-mediated short-hairpin RNA (shRNA) interference of MDK**. Short-hairpin RNAs (shMDK1 and shMDK2) designed to knock down CX43 synthesis were annealed and cloned into the pLL3.7 lentiviral vector containing the EGFP reporter gene. The lentiviral vectors were transfected into 293FT cells in the presence of packaging plasmids using Lipofectamine 3000 (Invitrogen) for lentivirus packaging. 293FT cells were infected, and the knockdown efficacy of each shRNA-containing lentivirus was assessed after 2 days by qPCR. The shRNA lentivirus with higher knockdown efficiency was selected for hPSC-CNC PC transduction and EGFP$^+$ cells were enriched for the detection of MDK protein levels and the subsequent in vivo transplantation study.

**Quantitative real-time PCR**. Total RNA was extracted using TRIzol Reagent (Invitrogen) according to the manufacturer's instructions. After digestion with DNase I (Fermentas, Glen Burnie, MD, USA), RNA yield was determined by using the NanoDrop ND-1000 spectrophotometer (NanoDrop Technologies). Total RNA (1 μg) was converted to cDNA using a Quantitect Reverse Transcription kit (Qiagen, Valencia, CA, USA). Quantitative real-time PCR (qPCR) analysis was performed using a DyNAmo ColorFlash SYBR Green qPCR kit (Thermo Fisher Scientific, Rutherford, NJ, USA) and the LightCycler 480 Detection System (Roche Diagnostics, Branchburg, NJ, USA). The expression levels were normalized to those of glyceraldehyde-3-phosphate dehydrogenase (GAPDH), and changes in gene expression were calculated as fold changes using the ΔΔCt method. Primer details are provided in Supplementary Table 1.

**Fluorescence-activated cell sorting (FACS) analysis**. For flow cytometry analysis, cells were dissociated with Accutase and single-cell suspensions were prepared in FACS buffer. Cell surface staining was completed using the antibodies outlined in Supplementary Table 2. 7-Aminoactinomycin D (7AAD) was used for dead cell exclusion. Cells were analyzed with a FACS Calibur flow cytometer (BD Biosciences). Control staining with the appropriate isotype-matched mouse monoclonal antibody controls was included to establish a threshold for positive staining and subset gating (Gating strategies are shown in Supplementary Fig. 17).

**Immunofluorescence staining and immunohistochemical (IHC) staining**. Cells were fixed with 4% PFA at room temperature for 20 min and rinsed three times with PBS. Cells were permeabilized with 0.3% Triton X-100 in PBS (Sigma-Aldrich), and were incubated overnight at 4 °C with primary antibody or isotype diluted in PBS containing 10% BSA or goat serum. Secondary antibodies (1:1000 dilution) were incubated for 2 h at room temperature. Samples were counterstained with 4′,6-diamidino-2-phenylindole (DAPI) (Sigma-Aldrich) and mounted with mounting medium (DAKO). The antibodies used are listed in Supplementary Table 3. Images were taken by using a confocal laser-scanning microscope (LSM 780; Carl Zeiss, Jena, Germany).

The brain samples located at the penumbra area were collected, sliced into 100-μm-thick sections (showing the longitudinal orientation of capillaries) and then used for histochemistry and immunostaining. Immunofluorescent images were acquired using an LSM800 confocal microscope (Zeiss) and Dragonfly high-speed confocal microscopy (ANDOR, Oxford Instruments). Quantitative image analysis, including NG2$^+$ pericyte coverage, fluorescent density analysis of extravascular dextran-TRITC, fibrinogen and albumin deposits, was performed from maximum projections of 10-μm-thick Z-stack images and conducted by the investigators blinded to the treatment groups using ImageJ software (NIH).

Terminal deoxynucleotidyl transferase dUTP nick end labeling (TUNEL) using the in situ cell death detection kit (Roche) or fluorescently labeled inhibitor of caspases (FLICA) probe active caspase-3 (ImmunoChemistry Technologies)/NeuN coimmunostaining was applied to monitor the general level of apoptosis in the ischemic hemisphere according to the manufacturer's instructions.

**Fluorescent intensity analysis of extravascular deposits**. TRITC-conjugated dextran (700,000 Da, Sigma, 0.1 ml of 10 mg/ml) was injected intravenously via the femoral vein 3 days after stem cell treatment. Experimental mice were transcardially perfused with ice-cold PBS and the ipsilateral hemispheres were removed. For the quantification of extravascular deposition of fibrinogen/albumin/TRITC-dextran, 10-μm maximum project Z-stacks were reconstructed using ZEN software (black edition, ZEISS), and the positive signals were subjected to the threshold processing and analyzed using the Area Integrated Density measurement tool of ImageJ. We analyzed six randomly selected fields from three mice per group.

**RNA sequencing**. Total mRNA was isolated from undifferentiated hPSCs, hPSC-CNCs, hPSC-CNC PCs on day 7 and day 14, and HBVPs. RNA sequencing libraries were constructed using the Illumina mRNA-seq Prep Kit (Illumina, San Diego, CA, USA) as recommended by the manufacturer. The fragmented and randomly primed 150 bp paired-end libraries were sequenced using Illumina HiSeq 2000. Sequencing data were processed using Consensus Assessment of Sequence and Variation (CASAVA, version 1.8.2; Illumina) using the default settings. The TPM values were used to evaluate the expression levels of genes. Pearson's correlation coefficients were calculated ($R^2$) to measure the similarities of the global gene expression profiles between pericyte-like cells from different hPSCs and control HBVPs. Finally, the RNA-Seq data were analyzed using Ingenuity Pathways Analysis (IPA) software (Ingenuity Systems Inc., Redwood City, CA, USA) to categorize the differentially regulated genes. RNA-seq data have been deposited in the Gene Expression Omnibus (GEO) under accession number GSE132857.

**Western blotting**. For western blot assay, samples were washed with cold PBS, lysed in 1× RIPA buffer and centrifuged at 15,000 × $g$ for 10 min at 4 °C. Equal amounts of protein were separated on sodium dodecyl sulfate polyacrylamide gel electrophoresis and then electrotransferred to 0.45 μm pore-sized polyvinylidene difluoride membranes (Millipore). Each membrane was blocked by a solution of 0.1% (v/v) Tween 20/TBS (TBS/T) containing 5% (w/v) nonfat milk powder for 1 h at room temperature and then incubated with appropriate primary antibodies (listed in Supplementary Table 4) overnight at 4 °C. Specifically bound primary antibodies were detected using peroxidase-coupled secondary antibodies and enhanced chemiluminescence signaling (Cell Signaling Technologies).

**Statistical analysis**. GraphPad Prism 7 Software was used for statistical analysis. Image J 1.52a Software was used for analyzing the contractile experiment and immunostaining images. CytExpert 2.0 and Flow Jo V 10.0 were used to analyzed FACS data. Ingenuity Pathways Analysis (IPA; Version: 52912811) software (Ingenuity Systems Inc.) was used for analyzing the RNA-Seq data. Data obtained from multiple experiments are reported as the mean ± SEM. The significance of the difference between the mean values was determined by two-tailed unpaired Student's $t$-test, one-way analysis of variance (ANOVA) or two-way ANOVA. Differences were considered significant when $P < 0.05$.

**Reporting summary**. Further information on research design is available in the Nature Research Reporting Summary linked to this article.

## Data availability

The authors declare that all data supporting the results in this study are available within the paper and its Supplementary Information. Raw data are available from the corresponding author upon reasonable request. The RNA-Seq data have been deposited in the GEO database, under accession number GSE132857. Source data are provided with this paper.

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

## Acknowledgements

This work was supported by the National Key Research and Development Program of China (2017YFA0103802, 2018YFA0107200, 2017YFA0103403, 2019YFA0110303); the Strategic Priority Research Program of the Chinese Academy of Sciences (XDA16020701); the National Natural Science Foundation of China (U1501245, 81970474, 81970222, 81730005, 31771616); the Key Research and Development Program of Guangdong Province (2019B020234001, 2019B020236002, 2019B020235002); Frontier and Innovation of Key Technology Project in Science and Technology Department of Guangdong Province (2016B030229002, 2016B090918040); Key Scientific and Technological Program of Guangzhou City (201704020223, 201802020023); Guangzhou Regenerative medicine and health Guangdong laboratory (2018GZR0301003); The Natural Science Foundation of Guangdong Province (2017A030313799, 2017A030313786, 2018A030313570, 2016A030310158); Critical disease stem cell therapy innovation team (2018KCXTD017); and the Fundamental Research Funds for the Central Universities (19ykpy158).

## Author contributions

J.S., Y.H., J.G., J.W., and W.L.: collection and/or assembly of data, data analysis, and interpretation, manuscript writing. Y.F., J.C., Y.W., Y.Q., Y. W., C.X., J.C., B.W., Y.M., L.H., X.C., S.Z., and W.H.: collection and/or assembly of data. Q.K., T.W., X.L., and W.Z.: provision of study material. A.P.X., W.L.: conception and design, final approval of manuscript.

## Competing interests

The authors declare no competing interests.
