## [Peer Review File · Nature Communications]

Reviewers' comments:

Reviewer #1 (Remarks to the Author):

In the manuscript entitled "Transplantation of cranial neural crest-derived pericytes from human pluripotent stem cells promotes functional recovery in a murine model of ischemic stroke" by Sun, Gong, Huang et al., explored the protective potential of cranial neural crest-derived pericytes (from human iPSCs) following ischemic stroke (MCAO). This study shows that their protocol of differentiating iPSCs into cranial neural crest (CNC) stem cells into is robust. They show that they can further differentiate them into pericyte-like cells (expressing the canonical pericyte markers *Pdgfrb* and *NG2*) using PDGFBB and bFGF. They show that the iPSC-CNC-derived pericyte-like cells participate in vascular growth and promote BBB properties in human brain microvascular endothelial cells (HBMECs). They show that I.V. infusion of iPSC-CNC-derived pericyte-like cells improve neurological/behavioral defects, neuronal death, and brain edema following MCAO-induced ischemic stroke. The protective effects elicited by the iPSC-CNC-derived pericyte-like cells may be due to improvements in the BBB properties in the brain vasculature.

While this paper does present a novel therapeutic idea in using iPSC-CNC derived pericyte-like cells to improve the pathological events of ischemic stroke, a very similar method in producing pericyte-like cells from iPSCs through a CNC differentiation protocol this has been previously published (Stebbins and Gastfriend et al., Mar 2019). However if more thorough investigations can be provided to show how the iPSC-CNC derived pericyte like cells improve the BBB and attenuate neuronal damage then the novelty of these studies may be heightened.

Comments:

- More thorough characterization is required to define the "pericyte-like" cells in addition to *Pdgfrb* and *NG2*. *Vimentin* and *Caldesmon* are also highly expressed by vSMCs and fibroblasts. Expression of *Anep*, *Abcc9*, *kcnj8* would be ideal to see as well. Expressional analysis of other neural-crest derived and fibrocyte markers would be ideal such as *Pdgfr-alpha*, *Col1a1/1a2*, *Tbx18* to understand how related these pericyte-like cells are to pericytes. It would be best to change phrasing in the title and throughout the text to refer to these cells as pericyte-like.

- The rationale for using human vascular brain pericytes (HVBPs) is not clear – especially when the pericyte transplantation experiments are ultimately performed in mice. More info is needed on how the HVBPs were isolated and whether they are freshly isolated or primary cell line.

- Please rephrase the "tube assays" to cord assay (Stebbins and Gastfriend et al.,). This phrasing presumes that the HBMECs co-cultured with HBVPs or iPSC-CNC pericyte-like cells are forming tubes however they are not known to lumenize under these conditions. Alternatively, data confirming lumenization could be provided.

- Fig. 6A,B. The contractility assay is not typical and is testing tension and adhesion. A cell type that does not exhibit these behaviors would be needed to show this is unique to the HBVPs and iPSC-CNC pericyte-like cells. Perhaps there is a better test for this and the authors can include a positive control/agonist to stimulate pericyte contraction in addition to adding a negative control cell type. In the absence of alpha-SMA expression, information on the mechanism of contractility would be appreciated.

-The TEER and improvement in TJ organization is convincing that the iPSC-CNC derived pericyte-like cells evoke BBB properties in HBMECs. However, earlier studies showing BBB-inducing properties of

pericytes came to the major conclusion that pericytes suppress transcytosis within the brain endothelium (Daneman et al 2010., Armulik et al., 2010). Further studies have substantiated these findings (Ben-Zvi Nature 2014). Assays to test the ability of the iPSC-CNC derived pericyte-like cells in suppressing transcytosis in HBMECs would be preferred. Such assays were performed by Stebbins and Gastfriend et al. In addition, quantification of frayed TJs is needed.

- Fig. 7. Please include gross coronal sections of brains with quantification of infarct zone following MCAO with and without the I.V. infusion of HBVPs and iPSC-CNC derived pericyte-like cells. While the improvement in neurological deficits and neuronal death are convincing it would be ideal to see the gross pathological features following these experiments.

- NCSC-PCs have no added benefit beyond HBVP. The injection of another cell types that fails to provide protection would be useful to show something unique about pericytes. In general, there lacks mechanistic detail on how either of these cells contribute to increased BBB properties. Is it conveyed by release of a molecule, or do pericytes need to integrate into the vascular wall in order for these properties to be conferred? It is possible that pericytes are supplying trophic factors such as pleiotropin (Nikolakopoulou et al., 2019), this could be considered as a protective mechanism.

-Functional tests of the BBB are needed in the MCAO experiments, brain edema/water content is not sufficient. Typically, release of endogenous blood-borne molecules are quantified (i.e. albumin, fibrinogen), and exogenous dyes of varying molecular weight are used (i.e. fluorescent-dextran).

-Figure 8B,C: TJ protein expression and organization is difficult to discern in cross sections of vessels. Images showing the longitudinal orientation of vessels will provide more information in expression and organization and the consistency along the vasculature. IHC for TJ proteins are generally clean where localization is distinctly between endothelial membranes. The quality of TJ staining in the figure is poor. Perhaps better antibodies are needed to provide a clearer depiction of TJ protein expression and localization. It seems that the focus was on larger diameter vessels where the BBB properties are not as profound, therefore a capillary focus is ideal. Anatomical locations are also needed for all images taken within the brain and their relation to the penumbra or infarct core. Please provide imaging approach in methods, the thickness of sections, maximum projections etc.

-Does I.V. infusion of HBVP and iPSC-CNC derived pericyte-like cells suppress transcytosis within the brain endothelium following MCAO?

-Fig. S8. Data showing perivascular localization of iPSC-CNC derived pericyte-like cells following I.V. infusion is not convincing. A couple incidental images are shown. Pdgfrb staining could be improved and images similar to ones found in Supplement Fig 9 would be great to see with endothelial-staining as well. Please quantify the amount of vessels with homed-pericytes. Where is this happening in relation to the infarct core? What kind of vessels? These are important factors to these studies and therefore needs to be included in a main figure.

Reviewer #2 (Remarks to the Author):

This manuscript by Li and colleagues describes the differentiation of human pluripotent stem cells to pericyte-like cells through a neural crest intermediate. The authors use a variety of in vitro tests to demonstrate the pericyte character of their cells, all of which are reasonably well done and convincing. However, these results have been previously demonstrated in other studies (which the authors

appropriately note). The authors then use a middle cerebral artery occlusion model and intravenous injection of the hPSC-derived pericytes to study therapeutic potential of these cells. These results are somewhat less convincing due to a lack of proper controls and a limited battery of tests. Overall, these limitations mute the reviewer's enthusiasm for this work.

Major concerns:

1) None of the in vitro work is novel per se. The authors correctly cite two papers that came out earlier this year demonstrating pericyte derivation from hPSCs through a neural crest intermediate. I can appreciate that the authors' work was likely in process when these other papers were published. However, given the profile of Nature Communications, I would be surprised to see a publication that confirms past results. This would be a decision for the editor, in my opinion.

2) The in vivo work lacks a proper cohort of experiments, but more importantly, lacks proper controls. Many previous reports have examined the beneficial effects of mesenchymal stem cells on stroke recovery, and as the authors properly note, pericytes have mesenchymal stem cell-like activity. However, all of their in vivo experiments use a vehicle control as the negative control, rather than a proper control cell type – such as hPSC-derived mesenchymal cells or hPSC-derived pericytes that are not "brain-specific". I suspect the effects that they observe are just general benefits of a cell that can suppress inflammation, rather than effects that are specific to brain pericytes. This outcome would significantly dampen the novelty of this work.

Minor concerns related to experiments:

1) In vitro characterization of BBB properties – the authors solely look at how pericytes improve tight junction-related properties in BMECs. Pericytes are known to serve other purposes, for example suppression of nonspecific transcytosis. The authors should demonstrate these properties as well. It is also unclear how the authors quantified "frayed" tight junctions. Their observations are very subjective. There are automated ImageJ and Matlab plugins that can be used to assess tight junction continuity quantitatively.

2) Figure 7B-D – a more comprehensive battery of assays for stroke recovery would include at least a rotarod test.

3) Fig 7E – the authors should show an example image of water content for each condition.

4) Fig 8 – these images again seem subjective. The images in panel A for PBS could just be a slice where fewer vessels are in plane. It is also difficult to see differences in panels B and C. It is unclear why the authors didn't do a vessel isolation and a more quantitative assay like western blot.

Reviewer #3 (Remarks to the Author):

Comments on NCOMMS-19-24655

Pericytes are perivascular cells that grow adjacent to capillary vessels and play important roles in blood brain barrier integrity and cerebrovascular function. Here the authors describe a protocol for differentiating pericytes from human pluripotent stem cells through a neural crest intermediate stage. The differentiated pericytes expressed markers for *Pdgfrb*, *NG2*, *CD146*, caldesmon, and vimentin, determined by immunostaining. An in vitro tube formation assay with endothelial cells indicated that

derived pericytes could form tube-like structures, and a gel contraction assay indicated that the derived pericytes could contract the gel in which they were imbedded. The authors then briefly present experiments using their derived pericytes as a treatment after transient middle cerebral artery occlusion (tMCAO). Derived pericytes injected intravenously into mice after tMCAO improved neurological and behavior scores and reduced neuronal death, and improved tight junction protein fluorescence profiles in large vessels. In several assays and experiments in vitro and in vivo, commercially available human embryonic brain vascular pericytes were used for comparison with the derived pericytes.

Overall it is not clear what novelty the authors pericyte differentiation method brings to the field. There are now two published methods detailing pericyte differentiation from stem cell sources. Characterization of resulting pericytes described in this manuscript is minimal and entirely in vitro. Furthermore, characterization and mechanism of effects of iPSC-derived pericytes on stroke outcome is inadequate. Many concerns exist:

Major

1. It is unclear what the novelty is of the present pericyte differentiation method. Two previous studies have described frontal lobe (cranial) pericyte differentiation from neural crest cells (Sci Adv. 2019 Mar 13;5(3):eaau7375, Stem Cell Reports. 2019 Mar 5;12(3):451-460). What are the advantages of this method over the previously described methods for differentiating pericytes? Also, Stem Cell Reports. 2019 Mar 5;12(3):451-460 showed that marker expression is not significantly different for mesodermal vs. neural crest derived pericytes.
2. The characterization of NCSC-PCs is minimal in vitro staining. It would be beneficial to perform western blots to compare levels of known pericyte markers between NCSC-PCs and primary pericytes.
3. How are the pericytes injected into the blood stream crossing the BBB? What is the mechanism? Cells do not normally spontaneously cross the BBB in general.
4. Related to above, how was clumping of the infused cells controlled? How was the infusion of NCSC-PCs and HBVPs performed? What is the estimated survival-percentage of the pericyte cells inside the mouse body relative to the number injected?
5. Some pericytes express SMA. Detection of SMA expression in pericytes may be dependent on staining techniques (PMID:29561727).
6. Did the authors assess levels of interferon-induced transmembrane protein 1, a pericyte-specific marker (Sci Rep. 2016 Oct 11;6:35108. doi: 10.1038/srep35108), in hiPSCs?
7. In addition to the gel assay in Figure 6 A-B, it would be more convincing to demonstrate hiPSC-derived pericyte contractility at the single cell level.
8. The authors investigate the pluripotent capacity of NCSC-PCs and HBVPs and found that their cells could be induced to differentiate to osteocytes, adipocytes, or chondrocytes. This seems more typical of mesodermal derived pericytes to differentiate into peripheral cells. However, brain pericytes have been demonstrated to differentiate into other brain cell types which is perhaps more interesting and relevant to NCSC-PCs (CNS pericytes) (Curr Pharm Des. 2008;14(16):1581-93, Stem Cells. 2015 Jun;33(6):1962-74. doi: 10.1002/stem.1977, Exp Gerontol. 2018 Oct 2;112:30-37. doi: 10.1016/j.exger.2018.08.003)

9. For the in vitro BBB models, the TEER values are very low for endothelial monolayers. Have the authors considered using primary endothelial cultures?

10. Figure 8 shows staining on large vessels (over 25 um diameter). How are pericytes, which are normally present on capillaries, contribute to such large vessels? The authors should focus on studying capillaries where the pericytes normally exist.

11. In the stroke experiments, were there differences between the stroke groups in terms of lesion volumes?

12. Since the iPSC-derived pericytes are injected into the vasculature after stroke, how are they getting to the infarct region, penumbra area, and other peri-infarct areas, where blood flow likely does not exist, is low or greatly reduced?

13. The authors do not show adequate evidence of BBB leakage reduction after pericyte injection. BBB leakiness should be measured by staining for extravasated molecules. Reduced water content might just reflect changes in brain edema, and does not necessarily reflect vasogenic edema or BBB changes.

14. It is unclear whether changes in Glut1 after stroke with pericyte treatment are due to differences in microvascular density or Glut1 levels in the vasculature. This should be evaluated more carefully, using microvessel western blotting for example.

15. How was the rejection controlled in C57 mice? Immunosuppression?

16. Figure 8 Should start with a staining and quantification for pericyte coverage. Supplementary figure S8 stainings are not convincing, it is difficult to see cell morphology for example.

17. How certain are the authors that the glial cells observed in their differentiated cultures are "schwann" cells? The markers used (S100b and GFAP) are typically used as astrocyte markers, and astrocytes are commonly observed in neuronal iPSC cultures.

18. Are all HBVP cells used for comparison studies sourced from ScienCell? Pericytes from this source are human embryonic brain pericytes, not adult pericytes. This should be made clear in the text and considered carefully when interpreting results.

19. The authors spend a significant part of their discussion (most of page 23) describing pericyte models and treatments in non-brain tissue. Brain pericytes are well known to have somewhat different properties than peripheral tissues. See for example Nature 2018 Feb 22;554(7693):475-480, Nat Neurosci. 2016 May 26;19(6):771-83, Physiol Rev. 2019 Jan 1;99(1):21-78

20. Important stroke methodology details are missing. What is the mortality rate of stroke surgery, exclusion criteria for these experiments, and how many mice were excluded? Was laser Doppler or a similar method used to confirm successful occlusion?

21. The title of the manuscript does not match very well the data presented. The stroke data presented is minimal.

Minor:

22. The text would benefit from proofreading or editing by a native English speaker.

23. Figure 7 does not have letter 'E'.

24. Please define "PO/FN" on first use.

Response to reviewers

The reviewers raised a number of constructive criticisms and suggestions. To fully address them, we have performed additional experiments as well as implementing considerable changes to the manuscript. As a result, we believe the manuscript is much stronger. We wish to take this opportunity to thank the reviewers for their valuable input. Below, we summarize the reviewers' comments, and describe point-by-point how we have addressed them.

Reviewer #1

In the manuscript entitled "Transplantation of cranial neural crest-derived pericytes from human pluripotent stem cells promotes functional recovery in a murine model of ischemic stroke" by Sun, Gong, Huang et al., explored the protective potential of cranial neural crest-derived pericytes (from human iPSCs) following ischemic stroke (MCAO). This study shows that their protocol of differentiating iPSCs into cranial neural crest (CNC) stem cells into is robust. They show that they can further differentiate them into pericyte-like cells (expressing the canonical pericyte markers *Pdgfrb* and *NG2*) using PDGFBB and bFGF. They show that the iPSC-CNC-derived pericyte-like cells participate in vascular growth and promote BBB properties in human brain microvascular endothelial cells (HBMECs). They show that I.V. infusion of iPSC-CNC-derived pericyte-like cells improve neurological/behavioral defects, neuronal death,

and brain edema following MCAO-induced ischemic stroke. The protective effects elicited by the iPSC-CNC-derived pericyte-like cells may be due to improvements in the BBB properties in the brain vasculature.

While this paper does present a novel therapeutic idea in using iPSC-CNC derived pericyte-like cells to improve the pathological events of ischemic stroke, a very similar method in producing pericyte-like cells from iPSCs through a CNC differentiation protocol this has been previously published (Stebbins and Gastfriend et al., Mar 2019). However if more thorough investigations can be provided to show how the iPSC-CNC derived pericyte like cells improve the BBB and attenuate neuronal damage then the novelty of these studies may be heightened.

Major comments:

Point 1: More thorough characterization is required to define the “pericyte-like” cells in addition to *Pdgfrb* and *NG2*. *Vimentin* and *Caldesmon* are also highly expressed by vSMCs and fibroblasts. Expression of *Anpep*, *Abcc9*, *kcnj8* would be ideal to see as well. Expressional analysis of other neural-crest derived and fibrocyte markers would be ideal such as *Pdgfr-alpha*, *Col1a1/1a2*, *Tbx18* to understand how related these pericyte-like cells are to pericytes. It would be best to change phrasing in the title and throughout the text to refer to these cells as pericyte-like.

Response: As suggested, we refer to these cells as pericyte-like cells throughout the text and performed additional assays to thoroughly characterize

the properties of human pluripotent stem cells-derived cranial neural crest pericyte-like cells (hPSC-CNC PCs). Using fluorescence activated cell sorting (FACS) analysis, we identified that PDGFR α , CD248, and NG2 were homogeneously expressed in hPSC-CNC PCs and HBVPs. NOTCH3, an important regulators of brain vascular integrity and pericyte expansion (Development. 2014,141:307-17), could also be detected in at least 50% of hPSC-CNC PCs and HBVPs. This data was added as Fig. S5a.

Accordingly, we compared the expression of pericyte or fibroblast-related genes at transcriptional level by qRT-PCR and found hPSC-CNC PCs and HBVPs expressed ABCC9, DLK1, KCNJ8, ANPEP (CD13), IFITM1, TBX18, PDGFRA, COL1A1, COL1A2, and LUM at similar or different levels, which was in accordance with our RNA-Seq data (Fig. S8d). This data was added as Fig. S5b. In addition, we also identified hPSC-CNC PCs expressed PDGFR β , Vimentin COL1A1, CD13, and IFITM1 by western blotting. Both of these data thoroughly define the characteristics of hPSC-CNC PCs cells. The corresponding data was presented as Fig. S5c.

Point 2: The rationale for using human vascular brain pericytes (HVBPs) is not clear – especially when the pericyte transplantation experiments are ultimately performed in mice. More info is needed on how the HVBPs were isolated and whether they are freshly isolated or primary cell line.

Response: We are sorry for the unclear description. In this study, human

vascular brain pericytes (HBVPs) are served as a positive control of iPSC-CNC PCs. HBVPs were purchased from ScienCell Research Laboratories (Carlsbad, CA, USA; Catalog #1200), which were isolated from human brain, cryopreserved at passage one after purification, and could be expanded for 15 population doublings, according to the manufacturer's instructions. These cells have been widely used to study the formation and functionality of the blood-brain barrier (J Cell Mol Med. 2016,20:980-986; Acta Neuropathol.2016, 131:753–773; Int J Biol Sci. 2016,12:87-99). The corresponding information was added in Methods section.

Point 3: Please rephrase the “tube assays” to cord assay (Stebbins and Gastfriend et al.). This phrasing presumes that the HBMECs co-cultured with HBVPs or iPSC-CNC pericyte-like cells are forming tubes however they are not known to lumenize under these conditions. Alternatively, data confirming lumenization could be provided.

Response: As suggested, we rephrase the “tube assays” to “cord assay” both in Results and Methods sections.

Point 4: Fig. 6A,B. The contractility assay is not typical and is testing tension and adhesion. A cell type that does not exhibit these behaviors would be needed to show this is unique to the HBVPs and iPSC-CNC pericyte-like cells. Perhaps there is a better test for this and the authors can include a positive

control/agonist to stimulate pericyte contraction in addition to adding a negative control cell type. In the absence of alpha-SMA expression, information on the mechanism of contractility would be appreciated.

Response: This is a good point. To address this concern, we performed additional experiments and used human bone marrow-derived mesenchymal stem cells (HMSCs; established in our lab; *Leukemia*. 2015,29:636-46) as the negative control and human aortic smooth muscle cells (HASMCs; isolated from human aorta and cryopreserved at passage one; purchased from ScienCell Research Laboratories, Catalog # 6110) as the positive control, respectively. Both the gel lattice contraction assay and the carbachol treatment assay indicate that hPSC-CNC PCs and HBVPs possess similar basal contractile tone compared to HASMCs, whereas HMSCs display minimal reduction in the initial gel size or surface area. These results further confirm that hPSC-CNC PCs and HBVPs have comparable contractile properties. The corresponding data was added as Fig. 5a-d.

Indeed, our results indicate that hPSC-CNC PCs and HBVPs did not express α SMA protein, which is consistent with the results of other studies (*Stem Cell Reports*. 2019,12:451-460. *Nature*. 2018,554:475-480). Thus, the contractility property of hPSC-CNC PCs and HBVPs may be attributed to the constitutive expression of other contractile proteins, including Vimentin, Caldesmon, tropomyosin (TPM1, TPM2, TPM3), and myosin (MYH9, MYH10) according to the results of qRT-PCR, immunofluorescence staining, or

RNA-Seq (Fig. S8d). The corresponding information was described in Discussion section.

Point 5: The TEER and improvement in TJ organization is convincing that the iPSC-CNC derived pericyte-like cells evoke BBB properties in HBMECs. However, earlier studies showing BBB-inducing properties of pericytes came to the major conclusion that pericytes suppress transcytosis within the brain endothelium (Daneman et al 2010., Armulik et al., 2010). Further studies have substantiated these findings (Ben-Zvi Nature 2014). Assays to test the ability of the iPSC-CNC derived pericyte-like cells in suppressing transcytosis in HBMECs would be preferred. Such assays were performed by Stebbins and Gastfriend et al. In addition, quantification of frayed TJs is needed.

Response: As suggested, we performed additional experiments to determine the suppression ratio of hPSC-CNC PCs for transcytosis in HBMECs. Compared to HBMECs monolayer group, coculture with HBVPs or hPSC-CNC PCs significantly decreased the Alexa 488-dextran fluorescence intensity in the medium (about 50-60% decrease; $p < 0.0001$), indicating that hPSC-CNC PCs have the potential to suppress transcytosis in HBMECs. The corresponding data was described in Results section and added as Fig. 5j.

Moreover, we performed additional experiments to quantify the frayed TJs in HBMECs with or without pericyte-like cells according the previous study (Sci Adv. 2019, 5(3): eaau7375.), and found that the frayed ZO-1 tight junctions

were greatly reduced after coculture with hPSC-CNC PCs when compared to HBMECs monoculture. The corresponding data was described in Results section and added as Fig. 5h.

Point 6: Fig. 7. Please include gross coronal sections of brains with quantification of infarct zone following MCAO with and without the I.V. infusion of HBVPs and iPSC-CNC derived pericyte-like cells. While the improvement in neurological deficits and neuronal death are convincing it would be ideal to see the gross pathological features following these experiments.

Response: To quantify the therapeutic effects of cell treatment, a series of evaluations assessing the neurological outcomes were performed. As suggested, we performed additional experiments to evaluate the infarct size of the coronal sections of brains following tMCAO with and without cell infusion by 2,3,5-triphenyltetrazolium chloride (TTC) staining. Compared with the PBS group, transplantation of HBVPs or hPSC-CNC PCs significantly reduced the infarct volumes, while only mild reduction in infarct size in the human dermal fibroblasts (HDFs) transplantation group. The corresponding data were added as Fig. 6f, g.

Point 7: NCSC-PCs have no added benefit beyond HBVP. The injection of another cell types that fails to provide protection would be useful to show something unique about pericytes. In general, there lacks mechanistic detail

on how either of these cells contribute to increased BBB properties. Is it conveyed by release of a molecule, or do pericytes need to integrate into the vascular wall in order for these properties to be conferred? It is possible that pericytes are supplying trophic factors such as pleiotropin (Nikolakopoulou et al., 2019), this could be considered as a protective mechanism.

Response: As suggested, we performed additional experiments to evaluate the therapeutic effects of the different transplanted cell types, including hPSC-CNC PCs, HBVPs and human dermal fibroblasts (HDFs). Both hPSC-CNC PCs and HBVPs treatment markedly reduced brain infarct volume, prevented neuron apoptosis and promoted neurological functional recovery in tMCAO mice in comparison to PBS group. However, infusion of HDFs displayed minimal therapeutic potential and the recovery of the neurological function in HDFs group was not statistically different from PBS treated group. The corresponding data was added as Fig. 6a-i.

As the reviewer mentioned, pericyte ablation would cause a rapid neurodegeneration cascade, which is partially due to the loss of pericyte-derived pleiotrophin (PTN) (Nat Neurosci. 2019,22:1089-1098). Therefore, we detected the transcripts of PTN in hPSC-CNC PCs by qPCR and found that these cells expressed very low level of PTN mRNA, which was consistent with the RNA-seq data (Fig. S10a). Intriguingly, based on the RNA-seq data and qPCR, hPSC-CNC PCs and HBVPs highly expressed midkine (MDK; Fig. S10b), the homologues of PTN. Both MDK and PTN

belong to a group of heparin-binding growth factors. To investigate whether MDK participate the protective mechanism, hPSC-CNC PCs were transduced with lentivirus containing short hairpin RNAs (shRNAs) against MDK (shMDK-1) for *in vivo* transplantation study. Compared to NTC group, MDK knockdown significantly impaired the neuroprotective effects of hPSC-CNC PCs, as shown by the increase of frayed TJ junction, infiltration of Evans blue dye, neuron apoptosis rate, and the neurological deficits score. Taken together, MDK contributes significantly to CNC PCs-mediated neuroprotection, which might provide new clues for elucidating the potential mechanisms of hPSC-CNC PCs restoring BBB integrity in tMCAO model. The corresponding data was added as Fig. S11a-f.

Point 8: Functional tests of the BBB are needed in the MCAO experiments, brain edema/water content is not sufficient. Typically, release of endogenous blood-borne molecules are quantified (i.e. albumin, fibrinogen), and exogenous dyes of varying molecular weight are used (i.e. fluorescent-dextran).

Response: This is a good suggestion. As suggested, we performed additional experiments to determine the BBB integrity *in vivo* through Evans blue assay, measurement of fibrinogen/albumin deposits and fluorescent-dextran (70 kDa dextran-TRITC) assay. Compared to the PBS group, infusion of hPSC-CNC PCs or HBVPs significantly decreased the leakage of Evans blue dye ($p <$

0.05). However, treatment with HDFs did not improve the blood-brain barrier function efficiently. Similar results were obtained in fluorescent-dextran assay and measurement of fibrinogen/albumin deposits. Increased leakage of the endogenous plasma protein fibrinogen and albumin can be detected in PBS and HDFs groups, while treatment with HBVPs and hPSC-CNC PCs remarkably decreased the fibrinogen and albumin deposits in the brain ($p < 0.001$). Moreover, we discovered that more dextran-TRITC passed through the BBB in PBS and HDFs groups than that in HBVPs and hPSC-CNC PCs groups ($p < 0.05$). These data suggest that transplanted HBVPs or hPSC-CNC PCs could efficiently restore the integrity of BBB. The corresponding data was added as Fig. 7d, e.

Point 9: Figure 8B,C: TJ protein expression and organization is difficult to discern in cross sections of vessels. Images showing the longitudinal orientation of vessels will provide more information in expression and organization and the consistency along the vasculature. IHC for TJ proteins are generally clean where localization is distinctly between endothelial membranes. The quality of TJ staining in the figure is poor. Perhaps better antibodies are needed to provide a clearer depiction of TJ protein expression and localization. It seems that the focus was on larger diameter vessels where the BBB properties are not as profound, therefore a capillary focus is ideal. Anatomical locations are also needed for all images taken within the brain and

their relation to the penumbra or infarct core. Please provide imaging approach in methods, the thickness of sections, maximum projections etc.

Response: We are sorry for the unrepresentative data of TJ protein expression and organization. As suggested, we performed additional experiments and provided images showing the longitudinal orientation of capillaries. We used double immunofluorescence staining for TJ proteins (Occludin, ZO-1) with the endothelial marker CD31 3 days post-stroke. The results demonstrated that disruption of TJ proteins was evident in ischemic hemisphere in PBS group and no obvious improvement was observed after HDFs treatment, whereas infusion with hPSC-CNC PCs or HBVPs drastically alleviated degradation of TJ proteins in the NVU. Quantitative analysis also confirmed that hPSC-CNC PCs or HBVPs transplantation effectively prevented the loss of TJ proteins after stroke. The corresponding data was added as Fig. 8d, e.

Accordingly, we added more detailed description of the anatomical locations and imaging approach of samples. All the samples were collected from the penumbra area, sliced into 100 μ m-thick sections and then used for histochemical staining. Immunofluorescent images were acquired using the LSM800 confocal microscope (Zeiss) and Dragonfly high speed confocal microscopy (ANDOR, Oxford Instruments). Quantitative image analysis, including NG2⁺ pericyte coverage, fluorescent density analysis of extravascular dextran-TRITC, fibrinogen and albumin deposits, were

performed from maximum projections of 10- μ m-thick Z-stack images and conducted by the investigators blinded to the treatment groups using ImageJ software (NIH). We provide a detailed description of the experimental protocol in Methods section.

Point 10: Does I.V. infusion of HBVP and iPSC-CNC derived pericyte-like cells suppress transcytosis within the brain endothelium following MCAO?

Response: This is a good point. As suggested, we performed additional experiments to examine whether transplantation of HBVPs or hPSC-CNC PCs suppress transcytosis within the brain endothelium following tMCAO by transmission electron microscopy (TEM) (Neuron. 2014,82:603–617). Compared to the sham group, the average number of caveolae in each TEM image was much higher in PBS and HDFs groups after tMCAO. Intravenous infusion of HBVPs or hPSC-CNC PCs remarkably decreased the number of caveolae compared to PBS group ($p < 0.01$). Such evidence indicates the upregulation of endothelial transcytosis caused by the damage of barrier function after tMCAO was efficiently alleviated by the treatment of HBVPs or hPSC-CNC PCs. The corresponding data was added as Fig. 7f.

Point 11: Fig. S8. Data showing perivascular localization of iPSC-CNC derived pericyte-like cells following I.V. infusion is not convincing. A couple incidental images are shown. Pdgfrb staining could be improved and images similar to

ones found in Supplement Fig 9 would be great to see with endothelial-staining as well. Please quantify the amount of vessels with homed-pericytes. Where is this happening in relation to the infarct core? What kind of vessels? These are important factors to these studies and therefore needs to be included in a main figure.

Response: We are sorry for the inadequate evidence to identify the perivascular localization of the transplanted hPSC-CNC PCs. To address the concern of reviewer, we performed additional experiments to determine the number and location of homed-hPSC-CNC PCs and the amount of vessels with these homed-pericytes. FACS analysis was applied to test the homing ability of transplanted hPSC-CNC PCs to the BBB, and we found that the number of the homed-hPSC-CNC PCs (DsRedE2⁺ cells) increased gradually after cell transplantation (~1% of total cells on day 7). Fluorescence microscopy indicated that most of the DsRedE2⁺ cells located at the capillaries in penumbra area. We also used fluorescent lectin-Dylight488 to label blood vessels to identify the anatomical relationship between endothelial cells and the transplanted hPSC-CNC PCs. The results revealed that the amount of vessels (green) covered by homed-hPSC-CNC PCs (red) was about 10% 1 day post cell infusion, while about 30% of the capillaries in penumbra area were surrounded by DsRedE2⁺ hPSC-CNC PCs 7 days post-treatment. Furthermore, immunofluorescence staining demonstrated that most of the DsRedE2⁺ homed-hPSC-CNC PCs still expressed the pericyte-specific

markers, PDGFR β and NG2, and localized near the lectin-Dylight488⁺ endothelial cells. These results confirmed the perivascular localization of the transplanted hPSC-CNC PCs. The corresponding data was added as Fig. 9a-d; S9b.

Reviewer #2

This manuscript by Li and colleagues describes the differentiation of human pluripotent stem cells to pericyte-like cells through a neural crest intermediate. The authors use a variety of in vitro tests to demonstrate the pericyte character of their cells, all of which are reasonably well done and convincing. However, these results have been previously demonstrated in other studies (which the authors appropriately note). The authors then use a middle cerebral artery occlusion model and intravenous injection of the hPSC-derived pericytes to study therapeutic potential of these cells. These results are somewhat less convincing due to a lack of proper controls and a limited battery of tests. Overall, these limitations mute the reviewer's enthusiasm for this work.

Major comments:

Point 1: None of the in vitro work is novel per se. The authors correctly cite two papers that came out earlier this year demonstrating pericyte derivation from hPSCs through a neural crest intermediate. I can appreciate that the authors' work was likely in process when these other papers were published.

However, given the profile of Nature Communications, I would be surprised to see a publication that confirms past results. This would be a decision for the editor, in my opinion.

Response: Indeed, two elegant studies demonstrated the differentiation protocol for pericyte-like cells with cranial neural crest origin from hPSCs. Stebbins MJ et al. found that treatment with E6 medium supplemented with several small molecules for 15 days yielded ~90% p75⁺/HNK1⁺ NCSCs, which were further purified for pericyte differentiation (Sci Adv. 2019, 5: eaau7375.). Compared to this study, our method could efficiently obtain putative cranial neural crest stem cells within a relatively short period (6 days). Faal T et al. also reported that 5-day treatment with CHIR99021-containing serum-free medium could efficiently induce hPSCs differentiate into p75⁺HNK1⁺ NCSCs (the quantitative data of the percentage of p75⁺HNK1⁺ cells was not provided), and the resulting cells without purification were then passaged and directly used for pericyte differentiation (Stem cell reports 2019,12:451-460). Therefore, pericyte-like cells from other developmental origin (e.g., mesoderm) may exist in this study.

Both the above two studies used the p75⁺HNK1⁺ NCSCs for deriving pericyte-like cells. Nonetheless, previous studies demonstrated there were two populations in p75⁺ cells derived from human pluripotent stem cells, only p75^{bright}HNK1⁺ cells with high levels of AP2 α were authentic neural crest stem cells, while p75^{dim}HNK1⁺ cells expressing high levels of PAX6 were actually

neural stem cells (Proc Natl Acad Sci USA. 2011,108:19240-5. Stem Cell Rep 2017,9:1043-1052). We also observed similar gene expression pattern in $p75^{\text{bright}}\text{HNK1}^+$ ($\text{PAX6}^{\text{low}}\text{AP2}\alpha^{\text{high}}$) cells and $p75^{\text{dim}}\text{HNK1}^+$ ($\text{PAX6}^{\text{high}}\text{AP2}\alpha^{\text{low}}$) cells in our differentiation system (see the following figure). Then we isolated this subpopulation of $p75^{\text{bright}}\text{HNK1}^+$ cells that expressed a set of cranial markers (HOXA1, HOXB1, LHX5 and OTX2; Fig. 1d) for pericyte differentiation. Moreover, we showed that hPSC-CNC PCs in our protocol did not express the posterior HOX genes (HOX10-12), which were highly expressed by mesoderm progenitor-derived pericyte-like cells from hPSCs (hPSC-MP PCs), although hPSC-CNC PCs and hPSC-MP PCs shared similar expression pattern of typical pericyte markers (Fig. S7a, b). Taken together, our protocol represents a new, fast, and efficient method for derivation of cranial pericyte-like cells from hPSCs through distinctive developmental pathway, which may provide new clues for the derivation and application of iPSC-CNC derived pericyte-like cells.

Figure legend: Detection of mRNA levels of neural crest- (AP2 α , SOX10) or neural stem cell-specific genes (PAX6, Nestin) in $p75^{\text{dim}}\text{HNK1}^+$ cells and

p75^{bright}HNK1⁺ by qPCR.

More importantly, we would like to clarify that this study focuses on the functional assessment of cranial neural crest-derived pericyte-like cells from hPSCs (hPSC-CNC PCs) and demonstrates their *in vivo* therapeutic potential in tMCAO mice model, for the first time. Our results indicate that transplantation of hPSC-CNC PCs could promote neurological functional recovery by restoration of the BBB integrity and reducing neuron apoptosis in ischemic stroke model, which may be partially attributed to the secretion of MDK by pericyte-like cells. Our results may not only supply a powerful tool for studying the pathogenesis of BBB dysfunction and associated diseases, but also represent a potentially ideal cell source for brain pericyte transplantation treatment for pericyte-related neurological diseases, such as stroke, Alzheimer's disease, and so on. We have revised the manuscript to place a clear emphasis on the novelties of our study in Discussion section.

Point 2: The *in vivo* work lacks a proper cohort of experiments, but more importantly, lacks proper controls. Many previous reports have examined the beneficial effects of mesenchymal stem cells on stroke recovery, and as the authors properly note, pericytes have mesenchymal stem cell-like activity. However, all of their *in vivo* experiments use a vehicle control as the negative control, rather than a proper control cell type – such as hPSC-derived mesenchymal cells or hPSC-derived pericytes that are not “brain-specific”. I

suspect the effects that they observe are just general benefits of a cell that can suppress inflammation, rather than effects that are specific to brain pericytes. This outcome would significantly dampen the novelty of this work.

Response: We agree that this is a good idea. Indeed, transplantation of mesenchymal stem cells (MSCs) derived from different tissues could reduce infarct size and improve functional outcome in rodent ischemic stroke models in our and other studies (Theranostics. 2018,8: 5929-5944; Trends Mol Med. 2012,18: 292-7). However, it was reported that tissue-resident MSCs have multiple developmental origins including neural crest and mesoderm (Elife. 2014,3:e03696), which may result in functional heterogeneity of MSCs and inconsistent outcomes in MSC-based therapies (Cytotherapy. 2013,15:2-8). To avoid the ambiguity caused by the heterogeneity of MSCs, we performed additional experiments to generate the mesoderm progenitor-derived pericyte-like cells from hPSCs (hPSC-MP PCs; not “brain-specific” pericytes) as the control cells for *in vivo* study. The neurological recovery and BBB integrity after cell transplantation in different groups were assessed and compared by multiple assays. The results showed that neurological deficit score, ECM (fibrinogen and albumin) deposits, and TRITC-Dextran leakage increased rapidly in PBS group. Interestingly, treatment with hPSC-MP PCs displayed minimal therapeutic potential and was not statistically different from PBS treated group. These results imply that hPSC-CNC PCs may possess the brain-specific therapeutic potential and will be an ideal cell source for the

treatment of BBB dysfunction-related disorders. The corresponding data was added in Results section and added as Fig. S12a-e.

Minor comments:

Point 1: In vitro characterization of BBB properties – the authors solely look at how pericytes improve tight junction-related properties in BMECs. Pericytes are known to serve other purposes, for example suppression of nonspecific transcytosis. The authors should demonstrate these properties as well. It is also unclear how the authors quantified “frayed” tight junctions. Their observations are very subjective. There are automated ImageJ and Matlab plugins that can be used to assess tight junction continuity quantitatively.

Response: This point echoes a similar point raised by reviewer #1. Accordingly, we performed additional experiments to determine whether hPSC-CNC PCs could efficiently suppress transcytosis in HBMECs. The results suggest that there is an appreciable decrease in 10-kDa dextran transcytosis in HBMECs upon HBVPs and hPSC-CNC PCs coculture (about 40-50% decrease in both groups; $p < 0.0001$) when compared to HBMECs monolayer control group, respectively. The corresponding data was described in Results section and added as Fig. 5j.

Moreover, we performed additional experiments to quantify the frayed TJs in HBMECs with or without pericyte-like cells according to the previous study (Sci Adv. 2019,5:eaau7375.), and found that the frayed ZO-1 tight junctions

were greatly reduced after pericyte-like cell coculture when compared to HBMECs monoculture. The corresponding data was added as Fig. 5h.

Point 2: Figure 7B-D – a more comprehensive battery of assays for stroke recovery would include at least a rotarod test.

Response: As suggested, we performed rotarod test to evaluate locomotor coordination and motor balance after tMCAO, besides the previous corner test and adhesive removal test. Latency to fall from the rod was recorded for each group. The results revealed that the rotarod latency (about 50 s) did not differ between all the groups on day 1 post cell transplantation. However, we found that the latency time was markedly prolonged on day 3 and day 7 after cell injection in hPSC-CNC PCs (up to 150 sec on day 7; $p < 0.001$) and HBVPs groups (up to 100 s on day 7; $p < 0.01$) when compared to the PBS and HDFs groups (about 50 s). The corresponding data was added as Fig. 6c.

Point 3: Fig 7E – the authors should show an example image of water content for each condition.

Response: As suggested, we provided representative images of water content (together with Evans blue staining) for each group. We observed that cerebral edema was apparent in the PBS and HDFs groups, whereas transplantation of hPSC-CNC PCs and HBVPs noticeably reduced the cerebral edema. Quantification of water content also showed markedly reduction of cerebral

edema in hPSC-CNC PCs and HBVPs groups when compared to PBS and HDFs groups ($p < 0.05$), which was in accordance with the corresponding brain swelling phenotype in different groups. The corresponding data was added as Fig. 7a.

Point 4: Fig 8 – these images again seem subjective. The images in panel A for PBS could just be a slice where fewer vessels are in plane. It is also difficult to see differences in panels B and C. It is unclear why the authors didn't do a vessel isolation and a more quantitative assay like western blot.

Response: This point echoes a similar point raised by reviewer #1. Accordingly, we performed additional experiments and provide images from each group showing similar density of capillaries in longitudinal orientation. We used double immunofluorescence staining for TJ proteins (Occludin, ZO-1) with the endothelial marker CD31 3 days post-stroke. The results demonstrated that disruption of TJ proteins was evident in ischemic hemisphere in PBS group and no obvious improvement was observed after HDFs treatment, while infusion with hPSC-CNC PCs or HBVPs drastically alleviated degradation of TJ proteins in the NVU. Quantitative analysis also proved that administration with hPSC-CNC PCs or HBVPs effectively prevented the loss of TJ proteins after stroke. The corresponding data was added as Fig. 8d, e.

In addition, we performed vessel isolation from brain samples for the

detection TJ proteins using western blotting as suggested. The results exhibited that the protein levels of GLUT1, ZO-1, and Occludin were sharply downregulated in PBS group and were significantly elevated after transplantation of hPSC-CNC PCs or HBVPs, whereas HDFs treatment did not affect the expression of TJ proteins. These results were in line with quantitative data from immunostaining of TJ proteins. The corresponding data was added as Fig. 8c.

Reviewer #3

Pericytes are perivascular cells that grow adjacent to capillary vessels and play important roles in blood brain barrier integrity and cerebrovascular function. Here the authors describe a protocol for differentiating pericytes from human pluripotent stem cells through a neural crest intermediate stage. The differentiated pericytes expressed markers for *Pdgfrb*, *NG2*, *CD146*, *caldesmon*, and *vimentin*, determined by immunostaining. An in vitro tube formation assay with endothelial cells indicated that derived pericytes could form tube-like structures, and a gel contraction assay indicated that the derived pericytes could contract the gel in which they were imbedded. The authors then briefly present experiments using their derived pericytes as a treatment after transient middle cerebral artery occlusion (tMCAO). Derived pericytes injected intravenously into mice after tMCAO improved neurological and

behavior scores and reduced neuronal death, and improved tight junction protein fluorescence profiles in large vessels. In several assays and experiments in vitro and in vivo, commercially available human embryonic brain vascular pericytes were used for comparison with the derived pericytes.

Overall it is not clear what novelty the authors pericyte differentiation method brings to the field. There are now two published methods detailing pericyte differentiation from stem cell sources. Characterization of resulting pericytes described in this manuscript is minimal and entirely in vitro. Furthermore, characterization and mechanism of effects of iPSC-derived pericytes on stroke outcome is inadequate. Many concerns exist:

Major comments:

Point 1: It is unclear what the novelty is of the present pericyte differentiation method. Two previous studies have described frontal lobe (cranial) pericyte differentiation from neural crest cells (Sci Adv. 2019 Mar 13;5(3):eaau7375, Stem Cell Reports. 2019 Mar 5;12(3):451-460). What are the advantages of this method over the previously described methods for differentiating pericytes? Also, Stem Cell Reports. 2019 Mar 5;12(3):451-460 showed that marker expression is not significantly different for mesodermal vs. neural crest derived pericytes.

Response: This point echoes a similar point raised by reviewer #2. Two elegant studies demonstrated the differentiation protocol for pericyte-like cells with cranial neural crest origin from hPSCs. Stebbins MJ et al. found that

treatment with E6 medium supplemented with several small molecules for 15 days yielded ~90% p75⁺/HNK1⁺ NCSCs, which were further purified for pericyte differentiation (Sci Adv. 2019,5:eaau7375.). Compared to this study, our method could efficiently obtain putative cranial neural crest stem cells within a relatively short period (6 days). Faal T et al. also reported that 5-day treatment with CHIR99021-containing serum-free medium could efficiently induce hPSCs to differentiate into p75⁺HNK1⁺ NCSCs (the quantitative data of the percentage of p75⁺HNK1⁺ cells was not provided), and the resulting cells without purification were then passaged and directly used for pericyte differentiation (Stem cell reports 2019,12:451-460). Therefore, pericyte-like cells from other developmental origin (e.g., mesoderm) may exist in this study.

Both the above two studies used the p75⁺HNK1⁺ NCSCs for deriving the pericyte-like cells. Nonetheless, previous studies demonstrated there were two populations in p75⁺ cells, only p75^{bright}HNK1⁺ cells with high levels of AP2 α were authentic neural crest stem cells, while p75^{dim}HNK1⁺ cells expressing high levels of PAX6 were actually neural stem cells (Proc Natl Acad Sci USA. 2011,108:19240-5. Stem Cell Rep. 2017,9:1043-1052). We also observed similar gene expression pattern in p75^{bright}HNK1⁺ (PAX6^{low}AP2 α ^{high}) cells and p75^{dim}HNK1⁺ (PAX6^{high}AP2 α ^{low}) cells in our differentiation system (see the following figure). Then we isolated this subpopulation of p75^{bright}HNK1⁺ cells that expressed a set of cranial markers (HOXA1, HOXB1, LHX5 and OTX2; Fig. 1d) for pericyte differentiation. Taken together, our protocol represents a

new, fast, and efficient method for derivation of cranial pericyte-like cells from hPSCs through distinctive developmental pathway, which may provide new clues for the derivation and application of iPSC-CNC derived pericyte-like cells.

Figure legend: Detection of mRNA levels of neural crest- (AP2α, SOX10) or neural stem cell-specific genes (PAX6, Nestin) in p75^{dim}HNK1⁺ cells and p75^{bright}HNK1⁺ by qPCR.

More importantly, we would like to clarify that this study focuses on the functional assessment of cranial neural crest-derived pericyte-like cells from hPSCs and demonstrates their *in vivo* therapeutic potential in tMCAO mice model, for the first time. Our results indicate that transplantation of hPSC-CNC PCs could promote neurological functional recovery by restoration of the BBB integrity and reducing neuron apoptosis in ischemic stroke model, which may be partially attributed to the secretion of MDK by pericyte-like cells. Our results may not only supply a powerful tool for studying the pathogenesis of BBB dysfunction and associated diseases, but also represent a potentially ideal cell source for brain pericyte transplantation treatment for pericyte-related

neurological diseases, such as stroke, Alzheimer's disease, and so on. We have revised the manuscript to place a clear emphasis on the novelties of our study in Discussion section.

For the reviewer's second concern, Faal T et al. (Stem cell reports 2019, 12(3): 451-460.) reported that marker expression is not significantly different between mesodermal- and neural crest-derived pericyte-like cells. Indeed, we showed that pericyte-like cells of both origins highly expressed CD13, NG2, CD146, CD29, CD44, CD73, and CD105. To discriminate the difference between the mesodermal- and neural crest-derived pericyte-like cells, we used qPCR to detect homeodomain transcription factors (HOX genes) that determining the anterior-posterior identity of body segments (Development. 2005,132:2931-42). Interestingly, the results unveiled a different HOX gene expression pattern between these two types of pericytes. We found that mRNA level of some of the posterior HOX genes (HOX10-12) was highly up-regulated in human mesodermal progenitor cell-derived pericyte-like cells (hPSC-MP PCs), rather than those of hPSC-CNC PCs, which was in line with the RNA-Seq data (Fig. S7b). The above evidence suggests that hPSC-MP PCs and hPSC-CNC PCs represent distinct developmental origins in spite of similar phenotype marker expression pattern, which may supply valuable insights to uncover the underlying mechanism why they possess different therapeutic capacity in tMCAO model. The corresponding information was added in Results and Discussion section.

Point 2: The characterization of NCSC-PCs is minimal in vitro staining. It would be beneficial to perform western blots to compare levels of known pericyte markers between NCSC-PCs and primary pericytes.

Response: This point echoes a similar point raised by reviewer #1. As suggested, we performed additional assays to thoroughly characterize the properties of neural crest stem cell-derived pericyte-like cells (hPSC-CNC PCs). Using fluorescence activated cell sorting (FACS) analysis, we identified that PDGFR α , CD248, and NG2 were homogeneously expressed in hPSC-CNC PCs and HBVPs. NOTCH3, an important regulators of brain vascular integrity and pericyte expansion (Development. 2014,141:307-17), could also be detected in at least 50% of hPSC-CNC PCs and HBVPs. This data was added as Fig. S5a.

Accordingly, we compared the expression of pericyte or fibroblast-related genes at transcriptional level by qRT-PCR and found hPSC-CNC PCs and HBVPs expressed ABCC9, DLK1, KCNJ8, ANPEP (CD13), IFITM1, TBX18, PDGFRA, COL1A1, COL1A2, and LUM at similar or different levels, which was in accordance with our RNA-Seq data (Fig. S8d). This data was added as Fig. S5b. In addition, we also identified hPSC-CNC PCs expressed PDGFR β , Vimentin, COL1A1, CD13, and IFITM1 by western blotting. Both of these data thoroughly define the characteristics of neural crest stem cell-derived pericyte-like cells. We added these points to the Results section. The

corresponding data was presented as Fig. S5c.

Point 3: How are the pericytes injected into the blood stream crossing the BBB?

What is the mechanism? Cells do not normally spontaneously cross the BBB in general.

Response: It is true that cells do not spontaneously cross the BBB in normal condition. Here, the stroke model was induced by occlusion of one-side middle cerebral artery with silicone-coated sutures for 40 minutes (transient MCAO, tMCAO). The blood flow was greatly reduced during occlusion stage. However, there was a reperfusion stage that could partially restore the blood flow for these model mice. The changes of blood flow in ischemic stage and reperfusion stage were further demonstrated by Laser Doppler flowmetry (Fig. S9a). We speculate that there may be at least three approaches contributing to the homing of hPSC-CNC PCs to the damaged BBB. First, in tMCAO mice, the reperfusion stage after vessel occlusion may allow the hPSC-CNC PCs to reach around the damaged site in the brain through the blood flow. Second, the ischemia-reperfusion injury would lead to the disruption of BBB structure and increase the BBB permeability as early as 6 hrs post-stroke (Nat Commun. 2016,7:10523; Neuron. 2014,82:603-17), which could result in pericyte-like cells crossing the BBB. Furthermore, the PDGF-B/PDGFR β interaction may also play a pivotal role during the homing of pericyte-like cells. It was reported that PDGF-B and PDGFR β were essential factors for the recruitment of

pericytes to brain capillaries during embryonic development (Science. 1997,277:242-5; Development. 1999,126:3047-55), and the expression of PDGF-B was found to be conspicuously elevated from 3 h to 1 week in MCAO model (Brain Res Mol Brain Res. 2003,113:44-51). In addition, PDGFR β was highly expressed in hPSC-CNC PCs as detected by qPCR, immunostaining and western blotting (Fig. 2c, d; S5c). Therefore, the injected hPSC-CNC PCs could be recruited to the penumbra area and help to reconstruct the BBB and prevent the apoptosis of neurons.

Actually, our results indicated that about ~1% of injected cells (on day 7) were recruited and most of the homed hPSC-CNC PCs (DsRedE2⁺ cells) located at the capillaries in penumbra area. We also identified about 30% of the capillaries in penumbra area were surrounded by DsRedE2⁺ hPSC-CNC PCs 7 days post-treatment. Furthermore, immunofluorescence staining demonstrated that most of the DsRedE2⁺ homed-hPSC-CNC PCs still expressed PDGFR β and NG2, and localized near the lectin-Dylight488⁺ endothelial cells (Fig. 9a-d; S9b). We revised the discussion to make this point more clear.

Point 4: Related to above, how was clumping of the infused cells controlled? How was the infusion of NCSC-PCs and HBVPs performed? What is the estimated survival-percentage of the pericyte cells inside the mouse body relative to the number injected?

Response: We added description of the cell infusion assay to the Methods section. In brief, confluent cells were trypsinized, triturated repeatedly and passed through a 40 μ m cell strainer to enrich for single cells. Cells were counted and visualized on a hemacytometer. If cell clumps were observed, cells were passed through a 25-gauge needle 10 times. Then 1×10^6 single cell suspension for each mouse were injected via caudal vein. The corresponding information was described in Methods section.

To address the reviewer's second concern, we performed additional experiments to detect the survival-percentage of transplanted cells homing to the damaged sites by FACS analysis, and discovered that the percentage of the homed-hPSC-CNC PCs (DsRedE2⁺ cells) increased gradually after cell transplantation (~1% of total cells in the ipsilateral hemisphere on day 7). The corresponding data was added as Fig. 9a.

Point 5: Some pericytes express SMA. Detection of SMA expression in pericytes may be dependent on staining techniques (PMID:29561727).

Response: As suggested, we performed additional experiments to determine the protein expression of α SMA in HDFs, HBVPs and hPSC-CNC PCs using the immunofluorescence staining protocol as described (Elife. 2018 Mar 21;7. pii: e34861. PMID: 29561727). We found that some of the HDFs were positive for α SMA with strong staining intensity, while only a small population of HBVPs weakly expressed α SMA. However, no α SMA⁺ cells were detected in

hPSC-CNC PCs (see the following figure), which was consistent with the results of previous studies (Stem Cell Rep. 2019,12:451-460; Nature. 2018,554:475-480).

Figure legend: Representative images for anti- α SMA immunofluorescence staining in different cell types. Scale bar: 50 μ m.

Point 6: Did the authors assess levels of interferon-induced transmembrane protein 1, a pericyte-specific marker (Sci Rep. 2016 Oct 11;6:35108. doi: 10.1038/srep35108), in hiPSCs?

Response: As suggested, we performed qPCR and western blotting to detect the expression of interferon-induced transmembrane protein 1 (IFITM1) in pericyte-like cells. The result showed that mRNA and protein of IFITM1 was strongly expressed in hiPSC-CNC PCs (Fig. S5b, c). Previous study also

reported that IFITM1 was expressed in human embryonic stem cells (hESCs) (FEBS Open Bio. 2017,7:1102-1110), which is in accordance with our RNA-Seq data (the TPM value of IFITM1 in hiPSC is 662). Then qPCR was also carried to compare the mRNA level of IFITM1 between hiPSCs and hPSC-CNC PCs, and we found that hPSC-CNC PCs had significantly higher level (3~9 folds) of IFITM1 transcripts than that in hiPSCs (see the following figure).

Figure legend: qPCR analysis of IFITM mRNA levels during pericyte differentiation process from hiPSCs (* $p < 0.05$, ** $p < 0.01$, *** $p < 0.001$, **** $p < 0.0001$ and n.s. is non-significant.).

Point 7: In addition to the gel assay in Figure 6 A-B, it would be more convincing to demonstrate hiPSC-derived pericyte contractility at the single cell level.

Response: This point echoes a similar point raised by reviewer #1. To address this concern, we performed additional experiments and used human bone marrow-derived mesenchymal stem cells (HMSCs; established in our lab; Leukemia. 2015 Mar;29(3):636-46) as the negative control and human aortic smooth muscle cells (HASMCs; isolated from human aorta and cryopreserved at passage one; purchased from ScienCell Research Laboratories, Catalog # 6110) as the positive control, respectively. Both the gel lattice contraction assay and the carbachol treatment assay (at single cell level) indicate that hPSC-CNC PCs and HBVPs possess similar basal contractile tone compared to HASMCs, whereas HMSCs display minimal reduction in the initial gel size or surface area. These results further confirm that hPSC-CNC PCs and HBVPs have comparable contractile properties. The corresponding data was added as Fig. 5a-d.

Point 8: The authors investigate the pluripotent capacity of NCSC-PCs and HBVPs and found that their cells could be induced to differentiate to osteocytes, adipocytes, or chondrocytes. This seems more typical of mesodermal derived pericytes to differentiate into peripheral cells. However, brain pericytes have been demonstrated to differentiate into other brain cell types which is perhaps more interesting and relevant to NCSC-PCs (CNS pericytes) (Curr Pharm Des. 2008;14(16):1581-93, Stem Cells. 2015 Jun;33(6):1962-74. doi: 10.1002/stem.1977, Exp Gerontol. 2018 Oct

2;112:30-37. doi: 10.1016/j.exger.2018.08.003)

Response: In this study, we demonstrated a multi-stage differentiation protocol to derive pericyte-like cells from cranial neural crest stem cells. It is known that neural crest stem cells (NCSCs) are multipotent cells that generate a wide variety of cell types, including neural lineage cells, mesenchymal cells, and melanocytes, as demonstrated by our and other studies (Mol Psychiatry. 2018,23:499-508; Nat Biotechnol. 2007,25:1468-75; Sci Adv. 2019,5:eaau 7375). Cranial neural crest stem cells were enriched by FACS and then induced to differentiate into pericyte-like cells (hPSC-CNC PCs) with serum-containing medium for 2 weeks. We infer that most, if not all, the pericyte-like cells (with mesenchymal phenotype) might lose the multipotency of NCSCs. Therefore, as suggested by the reviewer, we perform additional experiments to investigate whether hPSC-CNC PCs and commercial HBVPs have similar multilineage-differentiation characteristics under protocols described in the above reports. We found that less than 1% hPSC-CNC PCs could be differentiated into Tuj1⁺ neurons and S100B⁺ astrocytes. HBVPs also showed the similar tendency (see the following figure). These results indicate that a very small population of the hPSC-CNC PCs retained the neural lineages differentiation potential *in vitro*, although such behavior was not found in transplanted cells in our study.

Figure legend: Detection of multilineage differentiation potential of hPSC-CNC PCs and HBVPs by immunofluorescence staining. Scale bar: 20µm.

Point 9: For the in vitro BBB models, the TEER values are very low for endothelial monolayers. Have the authors considered using primary endothelial cultures?

Response: In our study, the HBMECs were primary cells and obtained from

ScienCell Research Laboratories (which were isolated from human brain and cryopreserved at passage one; Catalog #1000) and used for TEER assay at passage 3-4. Here, the TEER values for HBMECs monolayers are about 20-50 Ωcm^2 , which is similar to the results of previous reports using the same cell source or other primary endothelial cell lines (Front Cell Neurosci. 2019,13:230; MethodsX. 2015,3:25-34; Handbook of Neurochemistry and Molecular Neurobiology, 3rd ed., 2007, 29–55). However, some studies also reported significant higher TEER values when using hPSC-derived BMECs (Stem Cell Reports. 2019,12:451-460; Sci Adv. 2019,5: eaau7375; Stem Cell Rep. 2017,8:894-906) or primary HBMECs from ScienCell Research Laboratories (Cell Rep. 2019,26:1598-1613; J Neurosci Methods. 2013,212:173-9; BMC Neurol. 2019,19:289). Therefore, we speculate the discrepancy in TEER values may be due to the difference in cell sources/batches, TEER assay protocol, or measurement equipment used in these studies. Furthermore, when co-cultured with CNC PCs or HBVPs, significantly increased TEER value in HBMECs was observed from day 2 to day 4. Also, the immunostaining revealed that HBMECs co-cultured with CNC PCs or HBVPs had notably fewer frayed ZO-1 tight junctions and decreased the transcytosis in HBMECs compared to HBMECs monoculture (Fig. 5e-j). The above evidence demonstrates that the CNC PCs could efficiently improve the barrier function of HBMECs and we believe that our findings are convincing.

Point 10: Figure 8 shows staining on large vessels (over 25 μm diameter). How are pericytes, which are normally present on capillaries, contribute to such large vessels? The authors should focus on studying capillaries where the pericytes normally exist.

Response: We would like to clarify that Figure 8 displayed the co-staining of TJ proteins with endothelial marker CD31 or astrocyte marker GFAP in brain samples from different groups, which showed that the expression of TJ proteins was significantly restored in these large vessels, rather than showing the localization of the transplanted hPSC-CNC PCs.

As suggested, we perform additional experiments and provide images showing the longitudinal orientation of capillaries using double immunofluorescence staining for TJ proteins (Occludin, ZO-1) with the endothelial marker CD31 3 days post-stroke. The results demonstrated that disruption of TJ proteins was evident in ischemic hemisphere in PBS group and no obvious improvement was observed after HDFs treatment, whereas infusion with hPSC-CNC PCs or HBVPs drastically alleviated degradation of TJ proteins in the NVU. Quantitative analysis also confirmed that hPSC-CNC PCs or HBVPs transplantation effectively prevented the loss of TJ proteins after stroke. The corresponding data was added as Fig. 8d, e.

Point 11: In the stroke experiments, were there differences between the stroke groups in terms of lesion volumes?

Response: This point echoes a similar point raised by reviewer #1. To quantify the therapeutic effects of cell treatment, a series of evaluations assessing the neurological outcomes were performed. As suggested, we performed additional experiments to evaluate the infarct size of the coronal sections of brains following tMCAO with and without cell infusion by 2,3,5-triphenyltetrazolium chloride (TTC) staining. Compared with the PBS group, transplantation of HBVPs or hPSC-CNC PCs significantly reduced the infarct volumes, while only mild reduction in infarct size in the human dermal fibroblasts (HDFs) transplantation group. The corresponding data were added as Fig. 6f, g.

Point 12: Since the iPSC-derived pericytes are injected into the vasculature after stroke, how are they getting to the infarct region, penumbra area, and other peri-infract areas, where blood flow likely does not exist, is low or greatly reduced?

Response: In this study, the stroke model was induced by occlusion of one-side middle cerebral artery with silicone-coated sutures for 40 minutes (transient MCAO, tMCAO). We realized that the blood flow was greatly reduced during occlusion stage. However, there was a reperfusion stage that could partially restore the blood flow for these model mice. The ischemic stage and reperfusion stage were further demonstrated by Laser Doppler flowmetry (Fig. S9a). We speculate that there may be three important factors contributing

to the recruitment of hPSC-CNC PCs to the damaged BBB. First, in tMCAO mice, the reperfusion stage after vessel occlusion may allow the hPSC-CNC PCs to reach around the damaged site in the brain with the blood flow. Second, the Ischemia-reperfusion injury would lead to the disruption of BBB structure and increase the BBB permeability as early as 6 hrs post-stroke (Nat Commun. 2016,7:10523; Neuron. 2014,82:603-17), which could result in pericyte-like cells crossing the BBB. Furthermore, the PDGF-B/PDGFR β interaction may also play a pivotal role during the homing of pericyte-like cells. It was reported that PDGF-B and PDGFR β are essential factors for the recruitment of pericytes to brain capillaries during embryonic development (Science. 1997,277:242-5; Development. 1999,126:3047-55), and the expression of PDGF-B was found to be conspicuously elevated from 3 h to 1 week in MCAO model (Brain Res Mol Brain Res. 2003,113:44-51). In addition, PDGFR β was highly expressed in hPSC-CNC PCs as detected by qPCR, immunostaining and western blotting (Fig. 2c, d; S5c). Therefore, the injected hPSC-CNC PCs could be recruited to the penumbra area and help to reconstruct the BBB and prevent the apoptosis of neurons.

Actually, our results indicated that about ~1% of injected cells (on day 7) were recruited and most of the homed DsRedE2⁺ cells located at the capillaries in penumbra area. We also identified about 30% of the capillaries in penumbra area were surrounded by DsRedE2⁺ hPSC-CNC PCs 7 days post-treatment. Furthermore, immunofluorescence staining demonstrated that

most of the DsRedE2⁺ homed-hPSC-CNC PCs still expressed PDGFR β and NG2, and localized near the lectin-Dylight488⁺ endothelial cells (Fig. 9a-d; S9b). We revised the discussion to make this point more clear.

Point 13: The authors do not show adequate evidence of BBB leakage reduction after pericyte injection. BBB leakiness should be measured by staining for extravasated molecules. Reduced water content might just reflect changes in brain edema, and does not necessarily reflect vasogenic edema or BBB changes.

Response: This point echoes a similar point raised by reviewer #1. As suggested, we performed additional experiments to determine the BBB integrity *in vivo* through Evans blue assay, measurement of fibrinogen/albumin deposits and fluorescent-dextran (70 kDa dextran-TRITC) assay. Compared to the PBS group, infusion of hPSC-CNC PCs or HBVPs significantly decreased the leakage of Evans blue dye ($p < 0.05$). However, treatment with HDFs did not change the BBB permeability efficiently. Similar results were obtained in fluorescent-dextran assay and measurement of fibrinogen/albumin deposits. Increased leakage of the endogenous plasma protein fibrinogen and albumin can be detected in PBS and HDFs groups, while treatment with HBVPs and hPSC-CNC PCs remarkably decreased the fibrinogen and albumin deposits in the brain ($p < 0.001$). Moreover, we discovered that more dextran-TRITC passed through the BBB in PBS and HDFs groups than that in HBVPs and

hPSC-CNC PCs groups ($p < 0.05$). These data suggest that transplanted HBVPs or hPSC-CNC PCs could efficiently restore the integrity of BBB. The corresponding data were added as Fig. 7a-7e.

Point 14: It is unclear whether changes in Glut1 after stroke with pericyte treatment are due to differences in microvascular density or Glut1 levels in the vasculature. This should be evaluated more carefully, using microvessel western blotting for example.

Response: This point echoes a similar point raised by reviewer #2. Accordingly, we performed additional experiments and provide images from each group showing similar density of capillaries in longitudinal orientation. We used double immunofluorescence staining for TJ proteins (Occludin, ZO-1) with the endothelial marker CD31 3 days post-stroke. The results demonstrated that disruption of TJ proteins was evident in ischemic hemisphere in PBS group and no obvious improvement was observed after HDFs treatment, while infusion with hPSC-CNC PCs or HBVPs drastically alleviated degradation of TJ proteins in the NVU. Quantitative analysis also proved that administration with hPSC-CNC PCs or HBVPs effectively prevented the loss of TJ proteins after stroke. The corresponding data was added as Fig. 8d, e.

In addition, we performed vessel isolation from brain samples for the detection TJ proteins using western blotting as suggested. The results

exhibited that the protein levels of GLUT1, ZO-1, and Occludin were sharply downregulated in PBS group and were significantly elevated after transplantation of hPSC-CNC PCs and HBVPs groups, whereas HDFs treatment did not affect the expression of TJ proteins. These results were in line with quantitative data from immunostaining of TJ proteins. The corresponding data was added as Fig. 8c.

Point 15: How was the rejection controlled in C57 mice? Immunosuppression?

Response: We are sorry for the insufficient description about the immunosuppression protocol. Indeed, the animals were injected subcutaneously with Cyclosporin A (CSA, 10 mg/kg/day body weight, Sandoz, Switzerland) to suppress the immune rejection of transplanted cells according to the previous protocols (Brain Res. 2002,958:70-82; Stem Cells. 2013,31: 2354-63). We added this information to the Methods section.

Point 16: Figure 8 Should start with a staining and quantification for pericyte coverage. Supplementary figure S8 stainings are not convincing, it is difficult to see cell morphology for example.

Response: This is a good point. As suggested, we performed additional experiments to ascertain the pericyte coverage in different groups. NG2 and CD31 signals from micro-vessels (diameter < 10 μ m) were selected and subjected to the threshold processing and the areas occupied by the

fluorescent signals respectively were assessed by the Area measurement tool of ImageJ. The pericyte coverage was then calculated as a percentage (%) of NG2⁺ pericyte surface area covering CD31⁺ capillary surface area per observing field. The results indicate that loss of pericyte coverage was noted in PBS group, while treatment with hPSC-CNC PCs and HBVPs substantially increased the pericyte coverage when compared with PBS group. However, no significant difference in pericyte coverage was observed between HDFs group and PBS group. The corresponding data was added as Fig. 8a, b.

Moreover, to address the concern of reviewer, we performed additional experiments to determine the morphology, number and location of homed-hPSC-CNC PCs and the amount of vessels with these homed-pericytes. FACS analysis was applied to test the homing ability of transplanted hPSC-CNC PCs to the BBB, and we found that the number of the homed-hPSC-CNC PCs (DsRedE2⁺ cells) increased gradually after cell transplantation (~1% of total cells on day 7). Fluorescence microscopy indicated that most of the DsRedE2⁺ cells located at the capillaries in penumbra area. We also used fluorescent lectin-Dylight488 to label blood vessels to identify the anatomical relationship between endothelial cells and the transplanted hPSC-CNC PCs. The results revealed that the amount of vessels (green) covered by homed-hPSC-CNC PCs (red) was about 10% 1 day post cell infusion, while about 30% of the capillaries in penumbra area were surrounded by DsRedE2⁺ hPSC-CNC PCs 7 days post-treatment.

Furthermore, immunofluorescence staining demonstrated that most of the DsRedE2⁺ homed-hPSC-CNC PCs still expressed PDGFR β and NG2, and localized near the lectin-Dylight488⁺ endothelial cells. These results confirmed the perivascular localization of transplanted hPSC-CNC PCs. The corresponding data was described in Results section and added as Fig. 9a-d; S9b.

Point 17: How certain are the authors that the glial cells observed in their differentiated cultures are “schwann” cells? The markers used (S100b and GFAP) are typically used as astrocyte markers, and astrocytes are commonly observed in neuronal iPSC cultures.

Response: This is a good point. It is known that S100b and GFAP are typical markers of astrocytes in central nervous system (*Acta Neuropathol.* 2010,119: 7-35), while they are also widely used for the characterization of schwann cells (*Nat Biotechnol.* 2007,25:1468-75; *Am J Stem Cell* 2013,2:119-131; *Stem Cells Transl Med.* 2012,1:266-78.). In our study, however, these S100b and GFAP double positive cells are derived from p75^{high}HNK1⁺ neural crest stem cells, which usually generate cells of peripheral nervous system (*Nat Biotechnol.* 2009,27:275-80; *Nat Biotechnol.* 2007,25:1468-75). Moreover, we performed additional experiments and found that most of the GFAP⁺ glial cells co-expressed another Schwann cell marker, myelin basic protein (MBP), which is not present in astrocytes (*J Neurosci.* 1986,6:1925–1933). The above data

indicate that the glial cells observed in our differentiated cultures are actually Schwann cells. The corresponding data was added as Fig. S4c.

Point 18: Are all HBVP cells used for comparison studies sourced from ScienCell? Pericytes from this source are human embryonic brain pericytes, not adult pericytes. This should be made clear in the text and considered carefully when interpreting results.

Response: In this study, human vascular brain pericytes (HVBPs) are served as a positive control of hPSC-CNC PCs. HBVPs were purchased from ScienCell Research Laboratories (Carlsbad, CA, USA; Catalog #1200), which were isolated from human brain, cryopreserved at passage one after purification, and could be expanded for 15 population doublings, according to the manufacturer's instructions. These cells have been widely used to study the formation and functionality of the blood-brain barrier (J Cell Mol Med. 2016,20:980-986; Acta Neuropathol.2016, 131:753–773; Int J Biol Sci. 2016,12:87-99). The corresponding information was added in Methods section.

Point 19: The authors spend a significant part of their discussion (most of page 23) describing pericyte models and treatments in non-brain tissue. Brain pericytes are well known to have somewhat different properties than peripheral tissues. See for example Nature 2018 Feb 22;554(7693):475-480, Nat

Neurosci. 2016 May 26;19(6):771-83, Physiol Rev. 2019 Jan 1;99(1):21-78

Response: This is a good suggestion. It was reported that brain pericytes had some different characteristics in contrast to the peripheral counterparts. First, brain pericytes help to seal the BBB endothelium by regulating the expression of endothelial TJ proteins, which results in low rate of bulk-flow transcytosis. However, capillary endothelium in peripheral organs is leaky (Nat Neurosci. 2016,19:771-83; Physiol Rev. 2019,99:21-78.). We did discover that our hPSC-CNC PCs could strengthen the BBB integrity by increasing the expression level or inhibiting the disruption of TJ proteins and suppressing transcytosis in BBB endothelium both *in vitro* and *in vivo*. Also, Vanlandewijck M et al. (Nature. 2018,554:475-480.) revealed that CD13 was not expressed by lung pericytes, although these peripheral pericytes shared canonical pericyte markers *Pdgfrb*, *Cspg4*, and *Des* with brain pericytes. Moreover, they found that many members of different types of transporters were abundant in brain pericytes but low or absent in lung pericytes. In our study, we demonstrated that almost all of the hPSC-CNC PCs and HBVPs were positive for CD13 by FACS analysis. We further discovered that hPSC-CNC PCs highly expressed some of the solute carrier (SLC) transporters (*SLC1A5*, *SLC2A10*, *SLC16A1*, and others), ATP-binding cassette (ABC) transporters (*ATP1B3*, *ATP13A1*, *ATP2C1*, and others), and ATPase (ATP) family members (*ABCA1*, *ABCA2*, *ABCA7*, and others), as illustrated by RNA-Seq data (Fig. S8d). The above evidence suggests that the hPSC-CNC PCs closely resemble their *in*

vivo counterparts in the phenotype and functional properties.

More importantly, the dysfunction of brain pericytes has been implicated in various neurodegenerative diseases. Nikolakopoulou AM et al. showed that pericyte loss led to a rapid neuron loss in the brain due to loss of pericyte-derived pleiotrophin (PTN) (Nat Neurosci. 2019,22:1089-1098). Nortley R et al. revealed that the reduced cerebral blood flow in Alzheimer's disease (AD) was attributed to the constriction of brain capillaries at pericyte locations by amyloid β oligomers via ET1-ETA receptors pathway (Science. 2019,365. pii: eaav9518). It is also reported that degeneration of pericytes and BBB breakdown could be detected in samples of AD, Parkinson disease (PD), Amyotrophic lateral sclerosis (ALS), or Huntington's disease (Physiol Rev. 2019,99:21-78; Nat Neurosci. 2016,19:771-83). Consequently, the pericyte-like cells derived from cranial neural crest stem cells may supply an ideal cell source for the research of the pathogenesis and treatment of neurological disorders that are associated with pericyte loss and/or dysfunction.

As suggested, we repackaged this part and the above information was added in Discussion section.

Point 20: Important stroke methodology details are missing. What is the mortality rate of stroke surgery, exclusion criteria for these experiments, and how many mice were excluded? Was laser Doppler or a similar method used

to confirm successful occlusion?

Response: As suggested, we provide a detailed description of the experimental protocol in Methods section. The mortality rate of stroke surgery in our study was about 22.2% (52/234) on day 3 and 30.4% (28/92) on day 7. The inclusion criteria were: 1) regional cerebral blood flow decreases $>70\%$ during occlusion as detected by laser Doppler flowmetry; 2) neurological deficits 3 hour after tMCAO with neurological score between 2~3 points (determined as described in Stem cells 2018, 36: 1295-1310.). Overall there were 58 mice were excluded (49 for death and 9 for neurological scores). Laser Doppler flowmetry was used to confirm the ischemic stage and reperfusion stage. The corresponding data was added as Fig. S9a.

Point 21: The title of the manuscript does not match very well the data presented. The stroke data presented is minimal.

Response: As suggested, in the revised manuscript, we pay particular attention to the *in vivo* function and therapeutic potential of cranial neural crest-derived pericyte-like cells from human pluripotent stem cells for treating the murine model of ischemic stroke. We believe it will match the title quite well.

Minor Points

Point 1: The text would benefit from proofreading or editing by a native English speaker.

Response: As suggested, the revised manuscript was proofread by native English professionals from American Journal Experts (Certificate Verification Key: B0EE-CF87-5A38-5322-5A1A).

Point 2: Figure 7 does not have letter 'E'.

Response: We apologize for the errors that were due to our oversight during the preparation of the original manuscript. As suggested, the letter 'E' was added in Fig. 7.

Point 3: Please define "PO/FN" on first use.

Response: We apologize for the errors that were due to our oversight during the preparation of the original manuscript. As suggested, "PO/FN" was defined as "poly-L-ornithine/ fibronectin" on first use.

REVIEWER COMMENTS

Reviewer #1 (Remarks to the Author):

The authors have done a great job addressing the original comments. They have included: 1) more expressional data on their CNC-pericyte like cell line. 2) Convincing functional assays for the CNC-pericytes and necessary quantification. 3) Gross images of coronal MCAO brains with/without the infused CNC-pericytes. 4) Better BBB analysis (including transcytosis via EM) and functionality tests. 5) Very convincing and intriguing images showing the infused CNC-pericytes “home” to the capillaries in the penumbra. 6) Mechanism for how CNC-pericytes provide neuroprotection – through midkine a homolog of pleiotrophin that is necessary to provide partial protection from MCAO injury. Overall, this is a very interesting finding and impressive body of work.

The minor comments below help tie up loose ends, but do not detract from the overall enthusiasm for the manuscript.

(1) The authors did not directly address how “frayed” TJs was quantified. There is no section in the Methods that describes this in detail. While, I see that a criterion for this can be established in cultured cells, where the junctional boundaries can be clearly observed in 2D, I don’t see how the same criterion could be applied to the more complex brain histology images. The authors should describe more clearly how a capillary in histology was categorized as “frayed” vs no frayed. Also, it should be made clear whether this analysis was done in a blinded fashion.

(2) Figure 9d may not be the ideal example. The Dsred does not seem to be in the pericyte-like cell, but rather in the lumen of the capillary. There is a discontinuity in the dsred fluorescence, as one would see with a such red blood cell. The soma of the pericyte-like cell is not labeled, which means that the PDGFRb staining of the soma could not overlap, which is the point of the figure. Finally, there is blue staining, presumably for dapi, and this is not labeled as such. Overall, I feel that there must be better images to show considering the numerous homed pericytes shown in panel 9c.

(3) Fig. 2a. Cranial neural crest is misspelled.

Reviewer #2 (Remarks to the Author):

All my comments have been addressed sufficiently.

Reviewer #3 (Remarks to the Author):

The authors have generated a substantial amount of new data and analyses to address this reviewer’s comments. Overall the manuscript is greatly improved and most of the comments have been addressed. However, the new information provided brings up additional concerns in regards to characterization of the derived “pericyte-like” cells, and the immunosuppression method used for “pericyte-like” cell in vivo experiments, detailed below.

1. In regards to original point 2 (adequate characterization of derived hPSC-CSC-PCs), The authors state that in their FACS experiments that both hPSC-CSC-PCs and HBVPs expressed PDGFR-alpha. This is an oligodendrocyte and fibroblast marker, and not typically expressed by pericytes. This indicates that the NCSC-PCs are either a mixed population of multiple cell types, or a different cell type all together, such as perivascular fibroblast. Further characterization of this derived cell

population is needed.

2. In response to original point 15 (suppression pericyte and pericyte-like cell immune rejection), the authors indicate that cyclosporine A was used as an immunosuppressant during pericyte and pericyte-like cell treatment in the in vivo stroke experiments. It has been shown that cyclosporine A can also inhibit CypA-NFkB-MMP9 pathway in pericytes and improve blood brain barrier integrity in ApoE4 mice (Nature. 2012 May 16;485(7399):512-6). The authors should check whether cyclosporine A treatment itself had an effect in their model.

Reviewer #1

The authors have done a great job addressing the original comments. They have included: 1) more expressional data on their CNC-pericyte like cell line. 2) Convincing functional assays for the CNC-pericytes and necessary quantification. 3) Gross images of coronal MCAO brains with/without the infused CNC-pericytes. 4) Better BBB analysis (including transcytosis via EM) and functionality tests. 5) Very convincing and intriguing images showing the infused CNC-pericytes “home” to the capillaries in the penumbra. 6) Mechanism for how CNC-pericytes provide neuroprotection – through midkine a homolog of pleiotrophin that is necessary to provide partial protection from MCAO injury. Overall, this is a very interesting finding and impressive body of work.

The minor comments below help tie up loose ends, but do not detract from the overall enthusiasm for the manuscript.

Minor comments:

Point 1: The authors did not directly address how “frayed” TJs was quantified. There is no section in the Methods that describes this in detail. While, I see that a criterion for this can be established in cultured cells, where the junctional boundaries can be cleanly observed in 2D, I don’t see how the same criterion could be applied to the more complex brain histology images. The authors should describe more clearly how a capillary in histology was categorized as

“frayed” vs no frayed. Also, it should be made clear whether this analysis was done in a blinded fashion.

Response: We are sorry for the unclear description about the quantification of frayed TJs.

For the quantification of “frayed” TJs in the cultured cells, cells were defined as having frayed tight junctions if greater than 10% of the immunolabeled tight junction protrusions were not parallel to the cell–cell border (J Neurochem. 2011;119:507-20; J Neurochem. 2006;97:922-33). The percentage of cells expressing frayed tight junctions was counted by a blinded observer and a minimum of three separate frames and 200 total cells were counted to obtain a percentage of frayed tight junctions. We add the description in the Methods section.

For the histology images, quantitative analyses of murine brain tight junctions were performed from maximum projections of 10- μ m-thick Z-stack confocal images. 2D images were reconstructed by software ZEN (Zeiss) and subjected to further analyses by Image J (NIH). In the previous version of this manuscript, we borrow the criterion adapted from the method of *in vitro*-cultured endothelial cells that the capillary was defined as “frayed”, if over 10% of ZO-1/Occludin immunolabeled tight junction was discontinuous along the CD31/lectin-labeled blood vessel. We agree with the reviewer’s comment that this criterion might not be suitable for the complicated *in vivo* model. Therefore, we re-analyzed the TJ formation after pericyte-like cell

transplantation according to the method of ZO-1 or Occludin coverage area normalized by CD31/lectin-positive endothelial area (Stroke. 2016;47:1068-77; J Neuroinflammation. 2018;15:16). The quantification of TJ coverage area (displayed as the percentage of CD31-labeled endothelial area associated with ZO-1/Occludin) was added in the Methods and Results section (Fig. 8e; S13d, e).

Point 2: Figure 9d may not be the ideal example. The Dsred does not seem to be in the pericyte-like cell, but rather in the lumen of the capillary. There is a discontinuity in the dsred fluorescence, as one would see with a such red blood cell. The soma of the pericyte-like cell is not labeled, which means that the PDGFRb staining of the soma could not overlap, which is the point of the figure. Finally, there is blue staining, presumably for dapi, and this is not labeled as such. Overall, I feel that there must be better images to show considering the numerous homed pericytes shown in panel 9c.

Response: We apologize for providing atypical data for co-staining of PDGFR β and DsRedE2. As suggested, we replaced them with the more representative data showing PDGFR β /DsRedE2/lectin expression and co-localization. The corresponding data was added as Fig. 9d.

Point 3: Fig. 2a. Cranial neural crest is misspelled.

Response: We apologize for the errors that were due to our oversight during

the preparation of the original manuscript. Accordingly, the word “Cranial eural crest” was replaced by “Cranial neural crest” in Fig. 2a.

Reviewer #2

All my comments have been addressed sufficiently.

Reviewer #3

The authors have generated a substantial amount of new data and analyses to address this reviewer’s comments. Overall the manuscript is greatly improved and most of the comments have been addressed. However, the new information provided brings up additional concerns in regards to characterization of the derived “pericyte-like” cells, and the immunosuppression method used for “pericyte-like” cell *in vivo* experiments, detailed below.

Major comments:

Point 1: In regards to original point 2 (adequate characterization of derived hPSC-CSC-PCs), The authors state that in their FACS experiments that both hPSC-CSC-PCs and HBVPs expressed PDGFR-alpha. This is an oligodendrocyte and fibroblast marker, and not typically expressed by pericytes. This indicates that the NCSC-PCs are either a mixed population of multiple cell types, or a different cell type all together, such as perivascular fibroblast. Further characterization of this derived cell population is needed.

Response: This is a good suggestion. To address this concern, we reanalyzed the RNA-Seq data and observed that the CNC PCs and HBVPs did not express other oligodendrocyte lineage markers, such as SOX10 (the TPM value is about 0.3~1) or OLIG2 (the TPM value is about 0~0.1). Moreover, we detected the multi-lineage differentiation potential of CNC PCs and HBVPs and found that less than 1‰ hPSC-CNC PCs/HBVPs could be differentiated into Tuj1⁺ neurons and S100B⁺ astrocytes, while no O4⁺ oligodendrocytes were generated (see the following figure). These results indicate that CNC PCs might not contain the oligodendrocyte lineage cells.

Figure legend: Detection of multilineage differentiation potential of hPSC-CNC PCs and HBVPs by immunofluorescence staining. Scale bar: 20µm.

Although previous studies showed that *Pdgfra* (the TPM/FPKM value is about 1~13) was expressed at significantly lower levels than *Pdgfrb* (the TPM/FPKM value is about 388~470) in mouse brain pericyte samples (GSE75668, *Sci Rep.* 2016; 6: 35108; GSE100355, *Sci Rep.* 2018; 8: 12272), the other two studies showed hPSC-derived brain pericyte-like cells or

primary human brain pericytes expressed higher mRNA transcripts of PDGFRA than PDGFRB (GSE125869, Nat Med. 2020;26:952-963; GSE104141, SSRN. 2018; 10.2139/ssrn.3189103). In our study, we found that the mRNA expression level of PDGFRA (the TPM value is about 14~20) was lower than that of PDGFRB (the TPM value is about 38-66) in CNC PCs and HBVPs (GSE132857), and these results were consistent with RNA-Seq data of the recent publication (GSE124579, Sci. Adv. 2019; 5: eaau7375). To further confirm this data, we detected the expression of PDGFRA in CNC PCs and HBVPs by FACS. The result demonstrated that most of CNC PCs and HBVPs expressed PDGFRA (Fig. S5A). More importantly, the vast majority of PDGFRA⁺ CNC PCs and HBVPs also co-expressed pericyte markers including PDGFRB, CD146, CD13, and NG2 (>90%). These results indicate that CNC PCs resemble the primary human brain pericytes in the expression of cell surface markers and may not be a mixed population of multiple cell types. The corresponding data was added as Fig.S14.

Figure legend: FACS analysis of surface marker expression of CNC PCs and HBVPs.

Point 2: In response to original point 15 (suppression pericyte and pericyte-like cell immune rejection), the authors indicate that cyclosporine A was used as an immunosuppressant during pericyte and pericyte-like cell treatment in the *in vivo* stroke experiments. It has been shown that cyclosporine A can also inhibit CypA-NFκB-MMP9 pathway in pericytes and improve blood brain barrier integrity in ApoE4 mice (Nature. 2012 May 16;485(7399):512-6). The authors should check whether cyclosporine A treatment itself had an effect in their model.

Response: This is a good point. It was reported that cyclosporine A could promote the reconstruction of BBB and improve functional and structural neuronal changes through inhibiting CypA-NFκB-MMP9 pathway in brain pericytes in ApoE^{-/-} and ApoE4 mice (Nature. 2012;485:512-6.). To address this concern, we performed additional experiments to evaluate the therapeutic effects of cyclosporine A. The results showed that cyclosporine A treatment did not significantly improve the recovery of the neurological function, the infarct volumes, neuron apoptosis rate, TRITC-Dextran leakage, and degradation of TJ proteins, compared with the PBS- or HDFs+CSA group. While treatment with HBVPs+CSA or CNC PCs+CSA markedly promoted neurological functional recovery, reduced brain infarct volume, prevented neuron apoptosis, and restored BBB integrity in tMCAO mice in comparison to PBS, CSA, and HDFs+CSA groups. In addition, western blotting showed that the protein expression level of Cyclophilin A (CypA) did not change obviously among

different groups. The MMP9 expression was significantly increased in PBS, CSA, and HDFs+CSA groups, while infusion of HBVPs+CSA or CNC PCs+CSA strongly down-regulated the MMP9 protein level. We also found that NF κ B was activated in cells of penumbra area in stroke mice. However, NF κ B nuclear translocation (indicating NF κ B activation) was not detected in NG2⁺ pericytes after tMCAO. Moreover, cyclosporine A treatment or pericyte transplantation did not significantly influence the nuclear translocation of NF κ B in these cells (see the following figures). The above evidence suggests that treatment with cyclosporine A alone displayed minimal therapeutic potential and the CypA-NF κ B-MMP9 pathway may not be involved in the pathogenesis of BBB disruption in tMCAO mice. The corresponding data was added as Fig.S10, S11.

Figure legend: The assessment of neurological function and neuronal damage after treatment with PBS, CSA, HDFs+CSA, HBVPs+CSA and CNC PCs+CSA in stroke mice. a. Neurological deficit scores in different groups on day 3 were

evaluated (n=9; one-way ANOVA); b. Motor coordination measurement from the rotarod test was rescued in tMCAO mice treated with HBVPs or CNC PCs (n=9; two-way ANOVA); c. The time to remove the sticker in the adhesive removal test was analyzed at 3 days after MCAO (n=9; two-way ANOVA); d. The right/left ratio in the corner test on day 3 was calculated (n=9; one-way ANOVA); e. Infarct volumes of different groups were determined at 7days poststroke and quantified on TTC (red)-stained coronal cerebral sections (n=9; one-way ANOVA); f. Fluorescent staining with NeuN (red) and active caspase-3 (green) showed neuronal apoptosis in the cortex (CTX) and striatum (STR) at day 3 after transplantation. Scale bar: 50 μ m; g. The bar graph shows the quantification of the percentage of active caspase-3⁺ neurons in different groups (n=4; two-way ANOVA).

Figure legend: The recovery of the BBB function after treatment in stroke mice.

a. Localization of ZO-1, Occludin, and lectin in the cerebral tissue of different groups was analyzed. Scale bar: 100 μ m; b. The quantification of cerebral tight junctions as the percentage of lectin-labeled endothelial area associated with ZO-1/Occludin (n=4; two-way ANOVA); c. Expression of ZO-1, Occludin, CypA, and MMP9 was evaluated in the ipsilateral hemisphere using microvessel western blotting 3 days after cell transplantation; d. Representative images of

TRITC-dextran tracer extravasation assay. Scale bar: 100 μm ; e. Quantification analysis of TRITC-dextran tracer extravasation assay (n=4; one-way ANOVA); f. Detection of NF κ B nuclear translocation by immunofluorescence assay. Scale bar: 100 μm ;

REVIEWERS' COMMENTS:

Reviewer #3 (Remarks to the Author):

The authors have adequately addressed the comments. No further comments.